# Implicit Riemannian Optimism with Applications to Min-Max Problems

**Christophe Roux** [* 1 2]  **David Martínez-Rubio** [* 1 2 3]  **Sebastian Pokutta** [1 2]

## Abstract

We introduce a Riemannian optimistic online learning algorithm for Hadamard manifolds based on inexact implicit updates. Unlike prior work, our method can handle in-manifold constraints, and matches the best known regret bounds in the Euclidean setting, removing the dependence on geometric constants, like the minimum curvature. Building on this method, we develop multiple algorithms for g-convex, g-concave smooth min-max problems on Hadamard manifolds. Notably, one method nearly matches the gradient oracle complexity of the lower bound for Euclidean problems, for the first time.

## 1. Introduction

Riemannian optimization refers to the optimization functions defined over Riemannian manifolds. Such problems arise when the constraints of Euclidean optimization problems can be viewed as Riemannian manifolds, such as the symmetric positive-definite cone, the sphere, or the set of orthogonal linear layers for a neural network. This Riemannian formulation enables us to leverage the geometric structure of such problems by viewing them as unconstrained problems on a manifold.

Further, some non-convex Euclidean problems, such as operator scaling or optimistic likelihood estimation become geodesically convex, i.e., convex along all geodesics, when viewed as Riemannian optimization problems under the right metric (Allen-Zhu et al., 2018; Nguyen et al., 2019).

---

[*]Equal contribution (arbitrary order). [1]Zuse Institute Berlin, Germany [2]Technische Universität Berlin, Germany [3]Carlos III University of Madrid, Spain. Correspondence to: Christophe Roux <roux@zib.de>, David Martínez-Rubio <martinez-rubio.zib.de>.

*Proceedings of the $42^{nd}$ International Conference on Machine Learning*, Vancouver, Canada. PMLR 267, 2025. Copyright 2025 by the author(s).

Most of the non-local notations in this work have a link to their definitions, using **this code**, such as $\mathrm{Exp}_x$, which links to where this notation is defined as the exponential map of a manifold from a point $x$.

Riemannian optimization methods have many application in machine learning such as hyperbolic embeddings (Sala et al., 2018), hyperbolic neural networks (Ganea et al., 2018; Chami et al., 2019), Gaussian mixture models (Hosseini & Sra, 2015), the Karcher mean (Karcher, 1977), dictionary learning (Cherian & Sra, 2017; Sun et al., 2017), low-rank matrix completion (Vandereycken, 2013; Mishra & Sepulchre, 2014; Tan et al., 2014; Cambier & Absil, 2016; Heidel & Schulz, 2018), and optimization under orthogonality constraints (Edelman et al., 1998; Lezcano-Casado & Martínez-Rubio, 2019).

In this work, we analyze Riemannian optimization in the online setting where an agent receives an arbitrary, possibly adversarial, sequence of Riemannian loss functions and selects actions before knowing the loss functions. Its goal is to minimize the cumulative values of the losses associated to the actions it chooses, i.e., its *regret*. In particular, we are interested in *optimistic* methods, which allow the agent to improve its regret whenever the environment is not fully adversarial by predicting the next loss based on a *hint*.

Online optimistic Riemannian algorithms were analyzed by Hu et al. (2023a); Wang et al. (2023b). Both works face issues with circular arguments related to the geometry of the manifold. Hu et al. (2023a) consider the constrained setting but can only guarantee the actions to lie in a *neighborhood* of the constraint set. The size of this neighborhood depends on the step size, which in turn depends on a geometric factor dependent on the neighborhood's size. Wang et al. (2023b) consider the unconstrained setting but their step sizes depend on the diameter of the set in which the iterates stay set via geometric constants, which again influence the size of this set. They address this issue by *assuming* that the method's iterates stay in a compact set whose diameter is known *a-priori*.

Such circular relationships between step sizes and other problem parameters, which depend on the size of the set in which the iterates lie via geometric terms occur in many Riemannian optimization settings and lead to unfinished analyses, cf. Martínez-Rubio et al. (2024).

Chen & Orabona (2023); Choi et al. (2023); Ajalloeian et al. (2020); Dixit et al. (2019) study implicit online algorithms in the Euclidean setting. However, implicit optimistic algorithms have not been developed, even in the Euclidean case.

Our optimistic algorithm RIOD is based on a two-step implicit update rule, i.e., on minimizing a regularized version of the full loss or hint function, instead of a linearized version thereof. This is important since the linear approximations of geodesically convex (g-convex) are neither g-convex nor g-concave. Further, we chose a two-step variant even though there are optimistic methods requiring just one step in the Euclidean setting, since subtracting a full g-convex hint function loss would yield a potentially hard non g-convex subproblem.

In contrast to previous works, RIOD can enforce in-manifold constraints without relying on strong assumptions and has regret guarantees independent of geometric terms, matching the regret guarantees of Euclidean algorithms. Further, the regret guarantees allow for inexact updates under a certain precision criterion, which can be implemented cheaply for smooth losses. Given that our assumption covers the Euclidean space as a special case, this algorithm might be of independent interest for the online optimization community.

Application for online Riemannian optimization include online formulations of the Karcher mean, covariance estimation and robust subspace recovery (Karcher, 1977; Wiesel, 2012; Zhang, 2015). Another notable application of online optimistic methods is to solve $L$-smooth and g-convex, g-concave min-max problems. To that end, the problem is interpreted as a two-player, zero-sum game and the strategies of both variables are updated based on online optimistic algorithms. This means that both players use gradient based methods and hence adapt their strategy in an incremental fashion. This knowledge can be leveraged using optimism, which leads to faster rates, see Orabona (2019, Chapter 11.5) for an introduction to this approach in the Euclidean setting.

Applications of g-convex, g-concave and $L$-smooth min-max problems in machine learning include the robust Karcher mean, distributionally robust linear quadratic control and more generally the distributionally robust version of any finite-sum, g-convex optimization problem (Zhang et al., 2023; Jordan et al., 2022; Taskesen et al., 2023). Further, we believe that Riemannian min-max algorithms represent a promising method to solve distributionally robust optimization problems with ambiguity sets based on parametric probability distributions as their parameter space can often be seen as a Riemannian manifold, see (Brigant et al., 2023). Other Riemannian min-max problems, which do not satisfy the g-convex, g-concave and $L$-smooth conditions include geometry-aware robust PCA and projection-robust optimal transport (Horev et al., 2017; Jiang & Liu, 2023).

Building on our online algorithm RIOD, we introduce RIODA, an inexact and implicit min-max algorithm for possibly constrained problems based on updating both variables based on the RIOD update rule. Implementing the update rules using Composite Riemannian Gradient Descent (CRGD) introduced in Martínez-Rubio et al. (2024), we achieve near-optimal gradient oracle complexity of $O\left(LR^2/\varepsilon\right)$ and $\widetilde{O}(L/\mu)$ for $\mu = 0$ and $\mu > 0$, respectively, for an $\varepsilon$ duality gap, matching Euclidean algorithms up to logarithmic factors. In particular, the complexity only has a logarithmic dependence on $\zeta$, a geometric term arising from the curvature of the manifold, defined in the next section, for the first time. Note that implementing the CRGD update steps might be a hard computational problem. Nevertheless, this is a surprising result given that in the g-convex and $L$-smooth minimization setting the best known rates of $\widetilde{O}(\zeta + \sqrt{\frac{\zeta LR^2}{\varepsilon}})$ proven in Martínez-Rubio et al. (2023) do not match the Euclidean rates of $O(\sqrt{LR^2/\varepsilon})$ (Nesterov, 2005), where $R$ is the initial distance to a minimizer.

Implementing RIODA using Riemannian Gradient Descent (RGD) in the unconstrained setting, we improve the complexity from $\widetilde{O}(\zeta^4 LR^2/\varepsilon)$ (Martínez-Rubio et al., 2024) to $\widetilde{O}(\zeta^2 LR^2/\varepsilon)$, among the algorithms that do not require the knowledge of the initial distance to the solution.

In the constrained setting implementing RIODA using Projected RGD (PRGD), we improve the gradient oracle complexity by a factor of $\widetilde{O}(\zeta^{3.5})$ compared to the prior best rate (Martínez-Rubio et al., 2023). Furthermore, the algorithm does not require the knowledge of the Lipschitz constant of $f$ in the constraint set. We validate our theoretical results by running experiments on the robust Karcher mean problem in the symmetric positive definite manifold and the hyperbolic space, see Appendix E.

**Contributions.**

- **RIOD** An inexact Riemannian implicit online optimistic algorithm for the constrained setting on Hadamard manifolds matching the best known Euclidean regret bounds, while not relying on strong assumptions present in prior works. The regret bound does not depend on $\zeta$, a term arising from the geometry of the manifold.
- **RIODA** An inexact Riemannian implicit algorithm for min-max optimization on Hadamard manifolds with and without in-manifold constraints.
- Different implementations of the update rules of RIODA using first-order methods, which improve on prior works by either reducing the complexity or not requiring the knowledge of certain parameters. Notably, one variant nearly achieves the optimal Euclidean gradient oracle complexity, by removing the dependence on $\zeta$ up to log factors. This result is in contrast to smooth g-convex minimization, where we know that at least there is an extra curvature-dependent additive penalty, linear on $\zeta$, over the rate of the Euclidean-optimal accelerated solutions, cf. (Criscitiello & Boumal, 2023).

## 1.1. Preliminaries

A Riemannian manifold $(\mathcal{M}, \mathfrak{g})$ is a real $C^\infty$ manifold $\mathcal{M}$ equipped with a Riemannian metric $\mathfrak{g}$, which assigns a smoothly varying and positive definite inner product to each $x \in \mathcal{M}$. For $x \in \mathcal{M}$, denote by $T_x\mathcal{M}$ the tangent space of $\mathcal{M}$ at $x$. For vectors $v, w \in T_x\mathcal{M}$, we use $\langle v, w \rangle_x$ and $\|v\|_x \stackrel{\text{def}}{=} \sqrt{\langle v, v \rangle_x}$ for the metric's inner product and norm, and omit $x$ when it is clear from context. A geodesic of length $\ell$ is a curve $\gamma : [0, \ell] \to \mathcal{M}$ of unit speed that is locally distance minimizing.

The exponential map $\text{Exp}_x : T_x\mathcal{M} \to \mathcal{M}$ takes a point $x \in \mathcal{M}$, and a vector $v \in T_x\mathcal{M}$ and returns the point $y$ we obtain from following the geodesic from $x$ in the direction $v$ for length $\|v\|$, if this is possible. We denote its inverse by $\text{Log}_x(\cdot)$. It is well defined for uniquely geodesic manifolds, i.e., manifolds where every two points in that space are connected by one and only one geodesic, so we have $\text{Exp}_x(v) = y$ and $\text{Log}_x(y) = v$. We denote the distance between two points by $d(x, y) = \|\text{Log}_x(y)\|$. The manifold $\mathcal{M}$ comes with a natural parallel transport of vectors between tangent spaces, that formally is defined from the Levi-Civita connection $\nabla$. In that case, we use $\Gamma_x^y(v) \in T_y\mathcal{M}$ to denote the parallel transport of a vector $v$ in $T_x\mathcal{M}$ to $T_y\mathcal{M}$ along the unique geodesic that connects $x$ to $y$.

The sectional curvature of a manifold $\mathcal{M}$ at a point $x \in \mathcal{M}$ for a 2-dimensional space $V \subset T_x\mathcal{M}$ is the Gauss curvature of $\text{Exp}_x(V)$ at $x$. A Hadamard manifolds is a complete simply-connected Riemannian manifold of non-positive sectional curvature, which is in particular uniquely geodesic.

A set $\mathcal{X}$ is said to be g-convex if every two points are connected by a geodesic that remains in $\mathcal{X}$. For two points $x, y \in \mathcal{X}$, let $\gamma : [0, 1] \to \mathcal{M}$ be a constant speed geodesic joining $x$ and $y$ such that $\gamma(0) = x$ and $\gamma(1) = y$, then we call a function $f$ g-convex in $\mathcal{X}$ if $f(\gamma(t)) \leq tf(x) + (1-t)f(y)$ for all $x, y \in \mathcal{X}$. A differentiable function is $\mu$-strongly g-convex (resp., $L$-smooth) in $\mathcal{X}$, if we have ① (resp. ②) for any two points $x, y \in \mathcal{X}$:

$$\frac{\mu d(x,y)^2}{2} \stackrel{①}{\leq} f(y) - f(x) - \langle \nabla f(x), \text{Log}_x(y) \rangle \stackrel{②}{\leq} \frac{L d(x,y)^2}{2}.$$

The function g-convex if $\mu = 0$. Note the dependence on $\mathcal{X}$ is important, since for $\mu$-strongly g-convex and $L$-smooth function, the condition $L/\mu$ depends on the size of $\mathcal{X}$ (Criscitiello & Boumal, 2023, Proposition 53). A function $f$ is $G$-Lipschitz in $\mathcal{X}$ if $|f(x) - f(y)| \leq Gd(x,y)$ for all $x, y \in \mathcal{X}$.

Given $r > 0$, and a uniquely geodesic Riemannian manifold $\mathcal{M}$ with sectional curvature bounded in $[\kappa_{\min}, \kappa_{\max}]$. Then, we define the geometric constants $\zeta_r \stackrel{\text{def}}{=} r\sqrt{|\kappa_{\min}|}\coth(r\sqrt{|\kappa_{\min}|}) = \Theta(1 + r\sqrt{|\kappa_{\min}|})$ if $\kappa_{\min} <$

0 and $\zeta_r \stackrel{\text{def}}{=} 1$ otherwise, and $\delta_r \stackrel{\text{def}}{=} r\sqrt{\kappa_{\max}}\cot(r\sqrt{\kappa_{\max}})$ if $\kappa_{\max} > 0$ and $\delta_r \stackrel{\text{def}}{=} 1$ otherwise. It is $\delta_r \leq 1 \leq \zeta_r$.

We denote the closed Riemannian ball of center $x$ and radius $r$ by $\bar{B}(x, r)$. $\mathcal{P}_\mathcal{X}(x) \stackrel{\text{def}}{=} \arg\min_{y \in \mathcal{X}} d(y, x)$ denotes the metric projection onto a set $\mathcal{X}$. Note that for some sets, such as the Riemannian ball, $\mathcal{P}_\mathcal{X}$ can solved in closed form (Martínez-Rubio & Pokutta, 2023). The big-$O$ notation $\widetilde{O}(\cdot)$ omits $\log$ factors. We refer to Petersen (2006); Bacák (2014) for an overview of the Riemannian geometry used in this work.

## 1.2. Related Work

**Online Convex Optimization (OCO).** Online optimization is modeled as a sequential game between an agent and its environment, where in each round $t$ the agent chooses an action $x_t$ from some convex set $\mathcal{X}$. Then, the environment reveals a convex loss function $\ell_t$ and the agent pays the loss $\ell_t(x_t)$. Note that no assumptions are made about the environment, it could also act adversarially. The goal of the agent is to choose the sequences of action $x_t$ such that they minimize the difference between the cumulative losses paid by the agent and the losses associated to a fixed action $u \in \mathcal{X}$, i.e., its *regret*, $R_T(u) \stackrel{\text{def}}{=} \sum_{t=1}^T \ell_t(x_t) - \ell_t(u)$. While the OCO setting covers a wide range of settings, it can be overly pessimistic as real-world settings are rarely fully adversarial. The goal of *optimistic* methods is to improve the regret in settings that are not fully adversarial by predicting the next loss based on additional information about the environment, encoded by a *hint*. The regret of these methods depends on how well the hint approximates the real loss.

Another development in online optimization is the use of implicit updates. Typically, online optimization algorithms compute their next action minimizing the loss linearized at $x_t$ plus a regularizer. Implicit algorithm instead minimize the full loss function plus a regularizer, which is typically not solvable in closed form, hence the name. Recent works show improved regret guarantees for implicit versions of online mirror descent and follow the regularized leader (Campolongo & Orabona, 2020; Chen & Orabona, 2023). Some works also provide regret guarantees that allow for inexact solutions of the implicit updates (Chen & Orabona, 2023; Choi et al., 2023; Ajalloeian et al., 2020; Dixit et al., 2019).

**Online Riemannian Optimization.** The OCO framework has been extended to the Riemannian setting for g-convex loss functions. Bécigneul & Ganea (2019) provide $O(\sqrt{T})$ regret guarantees for adaptive online algorithms on Cartesian products of Riemannian manifolds. Wang et al. (2021) provide regret bounds for Lipschitz and g-convex Riemannian online optimization in the full information as well as the bandit and two-point feedback setting. Hu et al. (2023b)

analyze projection-free Riemannian methods in Hadamard manifolds using a linear minimization or a separation oracle in both the full-information as well as the bandit setting. Maass et al. (2022) provide regret guarantees for zeroth-order methods in Hadamard manifolds.

Hu et al. (2023a) introduce an optimistic algorithm for Hadamard manifolds with in-manifold constraints where the hint is an arbitrary vector in the tangent space of a secondary iterate sequence $M_t \in T_{y_t}\mathcal{M}$. This method achieves *improper* regret bounds of $O(\eta \zeta V_T + \frac{D^2}{\eta})$, where $V_T \overset{\text{def}}{=} \sum_{t=2}^{T} \|\nabla \ell_t(y_t) - M_t\|^2$ measures how well the hint predicts the loss functions. Improper regret is a relaxed notion where the agent's actions may lie outside the constraint set while the comparator must remain inside. Since the algorithm cannot guarantee that actions stay within the constraint set, a circular dependency arises: the size of the neighborhood around the constraint set depends on the step size, which in turn depends on function properties and geometric factors within this neighborhood. Consequently, it is unclear whether improper regret bounds translates to classical regret bounds.

Wang et al. (2023b) prove regret optimistic regret bounds for general Riemannian manifolds without in-manifolds constraints. They consider the specific case where the hint function is $M_t = \nabla \ell_{t-1}(x_{t-1})$ is fixed to the loss gradient from the last iteration. Therefore, their regret guarantee $O(\frac{D^2}{\eta} + \eta \frac{\zeta_D^2 (\bar{V}_T + G^2)}{\delta_D})$ scales with the gradient variation $\bar{V}_T \overset{\text{def}}{=} \sum_{t=2}^{T} \max_{x \in \mathcal{X}} \|\nabla \ell_t(x) - \nabla \ell_{t-1}(x)\|^2$. Further, the step size of the algorithm depends on the size of the optimization domain via geometric terms, which again influences the size of the optimization domain, leading to a circular relationship. They rely on the assumption that the iterates stay in a pre-defined domain without a mechanism of enforcing it. Since the loss functions change in every round, this assumption does not need to hold in practice.

**Riemannian Min-Max Algorithms**  We limit our discussion to the smooth and g-convex, g-concave setting. Zhang et al. (2023) introduced Riemannian Corrected Extra Gradient (RCEG), a variation of the Euclidean Extra Gradient (EG) algorithm and showed convergence rates, that are optimal up to geometric factors, for unconstrained g-convex, g-concave min-max problems on general Riemannian manifolds. Jordan et al. (2022) extended the analysis to the $\mu$-strongly g-convex, g-concave setting. Both works rely on the assumption that iterates of RCEG stay in a bounded domain without a mechanism to enforce it. Martínez-Rubio et al. (2023) remove this assumption by showing that the iterates of RCEG naturally stay in a compact set around a solution without impacting the rates by modifying the step size. Further, they introduce their algorithm RAMMA for Hadamard manifolds, which achieves faster rates with

Table 1: Comparison of Riemannian online algorithms. Our contribution is in **bold**. The entries in column $R_T$ denote the regret, where $\eta > 0$ is a parameter chosen by the algorithm. In column $\mathcal{X}$, ✔ indicates that the algorithm can enforce constraints. $M$ denotes the manifolds, where $\mathcal{H}$ denotes Hadamard manifolds, $E$ denotes the Euclidean space and $\mathcal{M}$ denotes general Riemannian manifolds. [1]Guarantees in terms of *improper* regret. [2]The geometric term $\zeta, \delta$ are with respect to a domain which is not guaranteed to be bounded.

| | $R_T$ | $\mathcal{X}$ | $M$ |
|---|---|---|---|
| **RIOD** | $O(\frac{D^2}{\eta} + \eta V_T)$ | ✔ | $\mathcal{H}$ |
| O-RCEG [HGA23][1,2] | $O(\frac{D^2}{\eta} + \eta \zeta V_T)$ | ✘ | $\mathcal{H}$ |
| ROOGD [WYH+23][2] | $O(\frac{D^2}{\eta} + \frac{\eta \zeta^2 (\bar{V}_T + G^2)}{\delta})$ | ✘ | $\mathcal{M}$ |
| OOMD [RS13] | $O(\frac{D^2}{\eta} + \eta V_T)$ | ✔ | $E$ |

Table 2: Comparison of Riemannian min-max algorithms for $\mu$-strongly g-convex, strongly g-concave and $L$-smooth problems. The entries of $\mu = 0$ and $\mu > 0$ contain the convergence rates for the two settings. The entries in column $M$ denote the manifolds, where $\mathcal{H}$ denotes Hadamard manifolds and $\mathcal{M}$ denotes general Riemannian manifolds. In column $\mathcal{X}$, ✔ indicates that the algorithm can enforce constraints. In column $R?$ and $G?$, ✔ means that the algorithm can be run *without* knowledge of the initial distance $R \overset{\text{def}}{=} d(x_1, x^*) + d(y_1, y^*)$ to a solution and Lipschitz constant $G$ of the function in the optimization domain, respectively. Note that $D$ refers to the diameter of the constraint set and $\tilde{R} \overset{\text{def}}{=} G/L + D$. [1]The iterates are not guaranteed to stay in a bounded domain. [2]In addition, they achieve a last-iterate rate of $O(\frac{\zeta_R}{\delta_R^{3/2}} \frac{LR^2}{\varepsilon^2})$ for $\mu = 0$. [3]The algorithm has faster rates in the fine-grained setting where the strong g-convexity and smoothness constants can vary between the variables. [4]Martínez-Rubio et al. (2023) showed that the iterates stay in a bounded set.

| Algorithm | $\mu = 0$ | $\mu > 0$ | $M$ | $\mathcal{X}$ | $R?$ | $G?$ |
|---|---|---|---|---|---|---|
| RIODA$_{\text{CRGD}}$ | $\widetilde{O}(\frac{LR^2}{\varepsilon})$ | $\widetilde{O}(\frac{L}{\mu})$ | $\mathcal{H}$ | ✔ | ✔ | ✔ |
| RIODA$_{\text{PRGD}}$ | $\widetilde{O}(\frac{\zeta_D \zeta_{\tilde{R}}}{\varepsilon} LR^2)$ | $\widetilde{O}(\frac{\zeta_D \zeta_{\tilde{R}} L}{\mu})$ | $\mathcal{H}$ | ✔ | ✔ | ✔ |
| RIODA$_{\text{RGD}}$ | $\widetilde{O}(\zeta_R^2 \frac{LR^2}{\varepsilon})$ | $\widetilde{O}(\zeta_R^2 \frac{L}{\mu})$ | $\mathcal{H}$ | ✘ | ✔ | ✔ |
| [MRP24] | $\widetilde{O}(\zeta_R^4 \frac{LR^2}{\varepsilon})$ | $\widetilde{O}(\zeta_R^4 \frac{L}{\mu})$ | $\mathcal{H}$ | ✘ | ✔ | ✔ |
| [CJL+23][1] | - | $\widetilde{O}(\frac{L^2}{\mu^2})$ | $\mathcal{M}$ | ✘ | ✘ | ✔ |
| [WYH+23] | $O\left(\frac{\zeta_R}{\delta_R} \frac{LR^2}{\varepsilon}\right)$ | - | $\mathcal{M}$ | ✘ | ✘ | ✘ |
| [HWW+23][2] | $O\left(\frac{\zeta_R}{\delta_R} \frac{LR^2}{\varepsilon}\right)$ | $\widetilde{O}(\frac{L(\zeta_R^2 + R^2 \kappa_{\max})}{\mu \delta_R})$ | $\mathcal{M}$ | ✘ | ✘ | ✘ |
| [MRC+23][3] | $\widetilde{O}(\zeta_{\tilde{R}} \zeta_D^{4.5} \frac{LR^2}{\varepsilon})$ | $\widetilde{O}(\zeta_{\tilde{R}} \zeta_D^{4.5} \frac{L}{\mu})$ | $\mathcal{H}$ | ✔ | ✘ | ✘ |
| [MLV22][4] | - | $\widetilde{O}(\frac{\sqrt{\zeta_R} L}{\sqrt{\delta_R} \mu})$ | $\mathcal{M}$ | ✘ | ✘ | ✔ |
| [ZZS23][4] | $O(\frac{\sqrt{\zeta_R} LR^2}{\sqrt{\delta_R} \varepsilon})$ | - | $\mathcal{M}$ | ✘ | ✘ | ✔ |

respect to function parameters in the fine-grained settings, where the smoothness and strong g-convexity parameters can vary between variables at the cost of worse dependence on $\zeta$. They achieve, among others, the optimal $O\left(1/\sqrt{\varepsilon}\right)$ rates with respect to $\varepsilon$ for the strongly g-convex, g-concave setting. While RAMMA can handle in-manifold constraints, the algorithm has five loops, making it complex to implement. Further, it requires the knowledge of the Lipschitz constant $G$ in the constrained setting and the initial distance to a solution $R$ in the unconstrained setting.

The following works consider the more general case of variational inequalities with monotone and Lipschitz operators, which encompass smooth and g-convex, g-concave min-max problems as special cases. Hu et al. (2023c) introduce two variants of the EG algorithm in general Riemannian manifolds, focusing on last-iterate convergence in the g-convex, g-concave setting. Cai et al. (2023) show linear convergence rates independent of geometric terms such as $\zeta$ in terms of the gradient norm in general Riemannian manifolds. However, they rely on the assumption that the iterates stay in a bounded domain without a mechanism to enforce it. Martínez-Rubio et al. (2024) introduce an inexact proximal point algorithm achieving accelerated rates. Implementing their update rule using the algorithm by Cai et al. (2023), they show that the iterates naturally stay in a bounded domain and achieve accelerated rates without requiring prior knowledge of the initial distance to a solution, for the first time.

Lastly, lower bounds specific to Riemannian optimization where established in (Hamilton & Moitra, 2021; Criscitiello & Boumal, 2021; 2023). These bounds hold for the $L$-smooth and g-convex minimization problem, which is a special case of g-convex, g-concave and $L$-smooth min-max optimization, and thus they apply to our min-max setting.

## 2. Riemannian Implicit Optimistic Online Gradient Descent

Before discussing the technical details of RIOD, we present the motivations underlying its design. There are two families of online optimistic algorithms, namely optimistic follow-the-regularized-leader (OFTRL) (Rakhlin & Sridharan, 2013a) and optimistic online mirror descent (OOMD) (Chiang et al., 2012; Rakhlin & Sridharan, 2013b).

In the Riemannian setting, a function linearized at a point $\bar{x} \in \mathcal{M}$ is defined with respect to the tangent space $T_{\bar{x}}\mathcal{M}$ and is not g-convex. However, it is star g-convex (and star g-concave) at $\bar{x}$, that is, convex (concave) along the geodesic going through $\bar{x}$. Thus, the OFTRL update rule for the action in round $t + 1$ in the Riemannian setting would

consist of minimizing

$$\sum_{i=1}^{t} \langle \nabla \ell_i(x_i), \mathrm{Log}_{x_i}(x) \rangle_{x_i} + \langle M_t, \mathrm{Log}_{x_t}(x) \rangle_{x_t} + \frac{d(x_t, x)^2}{2\eta},$$

where $\ell_t$, $x_t$ and $M_t$ refers to the loss, the action and the hint in round $t$. This function is neither g-convex nor star g-convex in general as each linearized function is only star g-convex with respect to one point. On the other hand, it is possible to implement OFTRL *without* linearizing the loss functions, i.e., minimizing $\sum_{i=1}^{t} \ell_i(x) + \tilde{\ell}_t(x) + \frac{1}{2\eta}d(x, x_t)^2$. If the loss functions $\ell_t$ and the hint function $\tilde{\ell}_t$ are g-convex, the update rule is also g-convex. Further, the analysis of OFTRL can be extended to the full-loss setting, both in the Euclidean and Riemannian setting. However, this approach is computationally impractical as it would require minimizing the sum of an increasing number of functions, making each iteration progressively more expensive to implement.

The original OOMD algorithm, due to Chiang et al. (2012) and further generalized by Rakhlin & Sridharan (2013b), is based on updating two iterate sequences, referred to as the *primary* and *secondary* iterates $\tilde{x}_t$ and $x_t$, where the agent chooses the primary iterates $\tilde{x}_t$ as actions. After the loss is observed in round $t$, the secondary iterate is updated based on the classical online mirror descent rule, i.e., $x_{t+1} \leftarrow \arg\min_x \langle \nabla \ell_t(x_t), x \rangle + \frac{1}{2\eta}\|x - x_t\|^2$. Then, the primary iterate is updated by $\tilde{x}_{t+1} \leftarrow \arg\min_x \langle M_t, x \rangle + \frac{1}{2\eta}\|x - x_{t+1}\|^2$ ensuring that the results stays close to the secondary iterate. Note that the secondary iterate is independent of the primary iterate, meaning that the agent always maintains a more conservative action from which it can compute an improved action based on the hint. This way the agent can benefit from the hint without accumulating errors arising when the hint does not perfectly predict the next loss. The method in Hu et al. (2023a) is based on this approach.

Joulani et al. (2017) introduced a single-iterate OOMD variant that corrects prediction errors by subtracting the previous round's hint at each iteration, i.e., $x_{t+1} \leftarrow \arg\min_x \langle \nabla \ell_t(x_t) + M_t - M_{t-1}, x \rangle + \frac{1}{2\eta}\|x - x_{t+1}\|^2$, thereby removing the need for a secondary iterate. Wang et al. (2023b) adapted this approach to the Riemannian setting for the special case where $M_t = \nabla \ell_t(x_t)$ by parallel-transporting the correction term to the tangent space of the current iterate, i.e. $x_{t+1} \leftarrow \arg\min_x \langle 2\nabla \ell_t(x_t) - \Gamma_{x_{t-1}}^{x_t} \nabla \ell_{t-1}(x_{t-1}), \mathrm{Log}_{x_t}(x) \rangle + \frac{1}{2\eta}d(x, x_{t+1})^2$.

Neither method can easily be extended to enforce in-manifold constraints. Recall that enforcing constraints is imperative for us since we aim to design an algorithm that does not suffer from a recurrent relationship between the size of the optimization domain and the step size. In general, the analysis of algorithms enforcing in-manifold constraints can be challenging in Riemannian manifolds. For manifolds

with positive curvature, the metric projection is not a non-expansive operator (Wang et al., 2023a, Section 6.1). Even in Hadamard manifolds, where the metric projection is non-expansive, the analysis remains challenging. Indeed, the first proof of linear convergence of PRGD for constrained strongly g-convex and smooth minimization problems in Hadamard manifolds without any non-standard assumptions was only recently established (Martínez-Rubio et al., 2023).

To address the challenges arising from the constraints, we propose an implicit algorithm based on the two-step OOMD, where the loss functions are not linearized. We do use a two-step approach in order to obtain a hint-correction term which makes the subproblems g-convex.

In our algorithm RIOD, $\tilde{x}_t$ and $x_t$ serve as primary and secondary iterates respectively. After playing $\tilde{x}_t$, the algorithm computes the secondary iterate $x_{t+1}$ via an implicit gradient step on the loss function $\ell_t$ received that round. Rather than playing $x_{t+1}$ in the next round, the agent selects a hint function $\tilde{\ell}_{t+1}$ to predict $\ell_{t+1}$ and takes an implicit step on that function to determine the next action $\tilde{x}_{t+1}$. As in OOMD, the secondary iterates $x_t$ are independent of the primary iterates $\tilde{x}_t$, thus preventing error accumulation from imperfect hint predictions.

---

**Algorithm 1** Riemannian Implicit Optimistic Online Gradient Descent (RIOD)

**Input:** Compact constraint set $\mathcal{X} \subset \mathcal{M}$ with diameter $D$, sectional curvature lower bound $\kappa_{\min}$, initial point $x_1 \in \mathcal{X}$, smoothness constant $L$ of loss and hint function $\ell_t$ and $\tilde{\ell}_t$, and proximal parameter $\eta > 0$.

**Definitions:** $\diamond$ *The alg. does not compute these quantities.*

- Exact solutions:
$$\tilde{x}_t^* \stackrel{\text{def}}{=} \arg\min_{x \in \mathcal{X}}\{\tilde{L}_t(x) \stackrel{\text{def}}{=} \tilde{\ell}_t(x) + \tfrac{1}{2\eta}d(x, x_t)^2\}$$
$$x_{t+1}^* \stackrel{\text{def}}{=} \arg\min_{x \in \mathcal{X}}\{L_t(x) \stackrel{\text{def}}{=} \ell_t(x) + \tfrac{1}{2\eta}d(x, x_t)^2\}$$

- Precision parameter:
$$\varepsilon_t \stackrel{\text{def}}{=} \frac{\max\{4, (t+1)^2(15 + 8\eta^2 L^2 + 2\eta^2 G^2(D^{-2} + 48|\kappa_{\min}|))\}^{-1}}{8\eta}$$

1: $\tilde{\ell}_1 \leftarrow 0$
2: **for** $t = 1$ **to** $T$ **do**
3:     Choose $\tilde{\ell}_t$
4:     Play $\tilde{x}_t \leftarrow (\varepsilon_t d(x_t, \tilde{x}_t^*)^2)$-minimizer of $\tilde{L}_t(x)$ in $\mathcal{X}$
5:     Receive $\ell_t$
6:     $x_{t+1} \leftarrow (\varepsilon_t d(x_t, x_{t+1}^*)^2)$-minimizer of $L_t(x)$ in $\mathcal{X}$
7: **end for**

---

The following result shows an optimistic regret guarantee for Hadamard manifolds with in-manifold constraints, improving on Hu et al. (2023a), as it can enforce the iterates to stay in $\mathcal{X}$ and therefore achieves classical regret bound instead of an improper one. Further, the regret is improved by

removing the dependence on $\zeta_D$, matching the best known Euclidean rates.

We also establish a dynamic regret bound for RIOD in Theorem 7, which compares the agent's actions against a sequence of comparators rather than a single fixed comparator.

**Theorem 1** (RIOD). [↓] *Let $\mathcal{M}$ be a Hadamard manifold with sectional curvature in $[\kappa_{\min}, 0]$. Further, for $t \in [T]$, let the loss and hint functions $\ell_t, \tilde{\ell}_t : \mathcal{M} \to \mathbb{R}$ be g-convex, and $L$-smooth in a compact and g-convex set $\mathcal{X} \subseteq \mathcal{M}$ with $D \stackrel{\text{def}}{=} \operatorname{diam}(\mathcal{X})$. For $\eta > 0$ we have for any $u \in \mathcal{X}$ that*

$$R_T(u) \leq \frac{3D^2}{2\eta} + \eta \sum_{t=1}^{T} \|\nabla \ell_t(\tilde{x}_t) - \nabla \tilde{\ell}_t(\tilde{x}_t)\|_{\tilde{x}_t}^2.$$

This result assumes that the hint function $\tilde{\ell}_t$ also satisfies smoothness and g-convexity, as $\ell_t$. In the linearized setting, the hint function is typically derived from the gradient of a function, such as the loss from the last round, as we do in our minmax application, making this a reasonable assumption. Further, the smoothness assumption can be removed when exact minimizers of the subproblems are found, as smoothness is only required to control the error induced in the inexact case.

We now discuss how the update rules in Lines 4 and 6 can be implemented up to the required precision. Importantly, in the online setting, the regret depends only on the cumulative cost $\sum_{t=1}^{T} \ell_t(x_t)$ paid for the agent's actions, and is independent of the gradient oracle complexity. Note that $\ell_t$ and $\tilde{\ell}_t$ are $L$-smooth and g-convex in $\mathcal{X}$ by assumption. The regularizer $\frac{1}{2}d(\cdot, x_t)^2$ is $(\zeta_D/\eta)$-smooth and $(1/\eta)$-strongly g-convex, cf. Corollary 16, and hence $L_t$ and $\tilde{L}_t$ are both $(1/\eta)$-strongly g-convex and $(L + \zeta_D/\eta)$-smooth functions. Therefore, we can efficiently implement the update rule using PRGD or CRGD, two minimization methods with linear convergence rates in this setting, defined by the following update rules. For a $\bar{L}$-smooth function $F : \mathcal{M} \to \mathbb{R}$,

$$x_{t+1} \leftarrow \mathcal{P}_{\mathcal{X}}\left(\operatorname{Exp}_{x_t}\left(-\bar{L}^{-1}\nabla F(x_t)\right)\right), \qquad \text{(PRGD)}$$

and for a composite function $F \stackrel{\text{def}}{=} f + g$

$$x_{t+1} \leftarrow \arg\min_{y \in \mathcal{X}}\{\langle \nabla f(x_t), \operatorname{Log}_{x_t}(y)\rangle + \frac{\bar{L}_f}{2}d(x_t, y)^2 + g(y)\},$$
$$\text{(CRGD)}$$

where $f$ is $\bar{L}_f$-smooth, see Appendix D.2 for more details. Note that $L_t$ and $\tilde{L}_t$ have condition number $L\eta + \zeta_D$. In comparison, the regularizer is $(1/\eta)$-strongly convex and smooth and the condition number becomes $L\eta$ in Euclidean space. This means that the curvature of the manifolds introduce an extra dependence on $\zeta_D$, since the convergence rate of PRGD depends on the condition number. We can circumvent this issue by leveraging the composite structure

of $L_t$ and $\tilde{L}_t$ using CRGD, as its convergence rate depends on a *composite* condition number, in this case the smoothness constant of $\ell_t$ and $\tilde{\ell}_t$ divided by the strong g-convexity constant of the regularizer, i.e., $L\eta$. Implementing each step of CRGD requires solving a potentially more expensive subproblem than PRGD. The subproblem in PRGD can be implemented easily at the cost of introducing a dependence on $\zeta$ on the computational time of the algorithm. In the following statement, we specify the gradient oracle complexity of implementing the implicit update steps using PRGD and CRGD.

**Corollary 2** (Implementing RIOD). [↓] *For the implementation of the update rules in Lines 4 and 6 of Algorithm 1, we require $\widetilde{O}((L\eta + \zeta_D)\zeta_{\tilde{R}})$ gradient oracle calls to $\ell_t$ and $\tilde{\ell}_t$ at iteration $t$ using PRGD or $\widetilde{O}(1 + L\eta)$ using CRGD (this includes a logarithmic dependence on $|\kappa_{\min}|$). Here $\tilde{R} \stackrel{\text{def}}{=} G/L + D$, where $G$ is the Lipschitz constants of $\ell_t$ and $\tilde{\ell}_t$ in $\mathcal{X}$. Note that these implementations do not require the knowledge of $G$.*

Since $\ell_t$ and $\tilde{\ell}_t$ are differentiable in the compact set $\mathcal{X}$, they are automatically Lipschitz continuous, so this does not need to be assumed separately.

## 3. Min-Max Optimization

Consider the following possibly constrained min-max optimization problem

$$\min_{x \in \mathcal{X}} \max_{y \in \mathcal{Y}} f(x, y) \qquad (\mathcal{P})$$

where $f$ is g-convex, g-concave and $L$-smooth and $\mathcal{X} \subseteq \mathcal{M}$ and $\mathcal{Y} \subseteq \mathcal{N}$. We call a function $f$ g-convex, g-concave in $\mathcal{X} \times \mathcal{Y}$ if $f(\cdot, y)$ and $-f(x, \cdot)$ are g-convex for all $x \in \mathcal{X}$ and $y \in \mathcal{Y}$, and $L$-smooth in $\mathcal{X} \times \mathcal{Y}$ if $\nabla_x f(\cdot, y)$, $\nabla_y f(x, \cdot)$, $\nabla_x f(x, \cdot)$, $\nabla_y f(\cdot, y)$ are $L$-Lipschitz for all $x \in \mathcal{X}$ and $y \in \mathcal{Y}$. We write $(x^*, y^*)$ for solutions of $(\mathcal{P})$, $R \stackrel{\text{def}}{=} d(x_1, x^*) + d(y_1, y^*)$ for the initial distance and, if finite, $D \stackrel{\text{def}}{=} \operatorname{diam}(\mathcal{X}) + \operatorname{diam}(\mathcal{Y})$.

We introduce a new method to solve $(\mathcal{P})$ based on updating $x$ and $y$ in parallel using RIOD. We make the following variable choices for $x$: $\tilde{x}_t \leftarrow \tilde{x}_t$, $x_t \leftarrow x_t$, $\ell_t(x) \leftarrow f(x, \tilde{y}_t)$, $\tilde{\ell}_t(x) \leftarrow f(x, y_t)$ and $u \leftarrow x^*$. For $y$, we set $x_t \leftarrow y_t$, $\tilde{x}_t \leftarrow \tilde{y}_t$, $\ell_t(x) \leftarrow -f(\tilde{x}_t, y)$, $\tilde{\ell}_t(x) \leftarrow -f(x_t, y)$, $u \leftarrow y^*$. We refer to the resulting algorithm as Riemannian Implicit Optimistic Gradient Descent-Ascent (RIODA),

$$\begin{cases} \tilde{x}_t \leftarrow \text{approx.} \arg\min_x \{f(x, y_t) + \frac{1}{2\eta} d(x, x_t)^2\} \\ \tilde{y}_t \leftarrow \text{approx.} \arg\max_y \{f(x_t, y) - \frac{1}{2\eta} d(y, y_t)^2\} \\ x_{t+1} \leftarrow \text{approx.} \arg\min_x \{f(x, \tilde{y}_t) + \frac{1}{2\eta} d(x, x_t)^2\} \\ y_{t+1} \leftarrow \text{approx.} \arg\max_y \{f(\tilde{x}_t, y) - \frac{1}{2\eta} d(y, y_t)^2\}. \end{cases}$$

See Algorithm 2 for a detailed description of the algorithm.

A common method to solve min-max problems is to use proximal algorithms (Tseng, 1995), which requires solving a regularized min-max problem at every iteration. However, for *constrained* min-max problems, we are not aware of any explicit method providing convergence rates. In contrast, RIODA only requires access to a g-convex minimization method, making it possible to implement the update rule using existing methods. The only other Riemannian min-max method that can handle constraints RAMMA (Martínez-Rubio et al., 2023) is also based on reducing the min-max problem to a series of minimization problems.

**Theorem 3** (RIODA). [↓] *Let $\mathcal{M}$, $\mathcal{N}$ be Hadamard manifolds with sectional curvature in $[\kappa_{\min}, 0]$ and $\mathcal{X} \subset \mathcal{M}$, $\mathcal{Y} \subset \mathcal{N}$ be compact and g-convex sets. Consider the $f : \mathcal{M} \times \mathcal{N} \to \mathbb{R}$, which is g-convex, g-concave and $L$-smooth in $\mathcal{X} \times \mathcal{Y}$. Further, let $(x^*, y^*)$ be a saddle point of $(\mathcal{P})$ and $(\tilde{x}_T, \tilde{y}_T)$ be the output of Algorithm 2 after $T$ iterations. Then we have $f(\tilde{x}_T, y^*) - f(x^*, \tilde{y}_T) \leq \varepsilon$ after $T = \lceil \frac{8LR^2}{\varepsilon} \rceil$ iterations, and $T = \lceil \frac{17L}{\mu} \log\left(\frac{4LR^2}{\varepsilon}\right) \rceil$, if $f$ is also $\mu$-strongly g-convex, strongly g-concave in $\mathcal{X} \times \mathcal{Y}$.*

In order to obtain the full gradient oracle complexity of RIODA, we need to take into account the number of gradient evaluations required to compute the update rule at every iterations. This is different from the online setting where we mainly care about *regret* which does not necessarily coincide with the gradient oracle complexity.

In the following, we show that similarly to RIOD, the update rule can be implemented efficiently. In particular, by our choice of $\eta = 1/(4L)$, the condition number of the subproblems becomes $1/4 + \zeta_D$, and the *composite condition number* becomes $1/4$.

**Corollary 4** (Implementing RIODA). [↓] *We use the notation from Algorithm 2. For the implementation of the update rules in Lines 2 and 3 of Algorithm 2, we require $\widetilde{O}(\zeta_D \zeta_{\tilde{R}})$ gradient oracle calls per iteration using PRGD or $\widetilde{O}(1)$ using CRGD (this includes a logarithmic dependence on $|\kappa_{\min}|$). Here $\tilde{R} \stackrel{\text{def}}{=} G/L + D$, where $G$ is the Lipschitz constant of $f$ in $\mathcal{X} \times \mathcal{Y}$. We refer to these algorithms as RIODA_PRGD and RIODA_CRGD, respectively. Note that these implementations do not require the knowledge of $G$.*

Since the update rule of $\tilde{x}_t$ is independent of $\tilde{y}_t$, both steps can be implemented in parallel. The same holds for $x_{t+1}$ and $y_{t+1}$.

The rates of RIODA_PRGD improve over RAMMA (Martínez-Rubio et al., 2023), the only other method for constrained, smooth and g-convex, g-concave min-max problems, by a factor of $\widetilde{O}(\zeta_R^{3.5})$ and also do not require the knowledge of the initial distance $R$ and the Lipschitz constant $G$.

We can also use RIODA to solve $(\mathcal{P})$ *without* in-manifold constraints as a reduction from RIOD. While RIOD requires

a compact constraint, the reduction works because we can show that the iterates of RIODA naturally stay in a ball around a solution $(x^*, y^*)$. Hence, one can add any hypothetical constraints which contain this ball without modifying the algorithm as they are guaranteed to never be active.

**Theorem 5** (RIODA – unconstrained). [↓] *Let $\mathcal{M}, \mathcal{N}$ be Hadamard manifolds with sectional curvature in $[\kappa_{\min}, 0]$. Consider the bi-function $f : \mathcal{M} \times \mathcal{N} \to \mathbb{R}$, which is g-convex, g-concave and $L$-smooth in $\mathcal{Z} = \bar{B}(x^*, 8R) \times \bar{B}(y^*, 8R)$, where $(x^*, y^*)$ is a saddle point of $f$. Then the iterates of [Algorithm 2] stay in $\mathcal{Z}$. Let $(x_T, y_T)$ be the output of [Algorithm 2] after $T$ iterations. Then we have $f(x_T, y^*) - f(x^*, y_T) \leq \varepsilon$ after $T = \lceil \frac{6LR^2}{\varepsilon} \rceil$ iterations and $T = \lceil \frac{17L}{\mu} \log \left( \frac{2LR^2}{\varepsilon} \right) \rceil$ if $f$ is in addition $\mu$-strongly g-convex, strongly g-concave in $\mathcal{Z}$.*

In the following, we quantify the complexity of implementing the update rules using the minimization algorithms RGD, defined by the update rule for a $\bar{L}$-smooth function

$$x_{t+1} \leftarrow \operatorname{Exp}_{x_t} \left( -\bar{L}^{-1} \nabla f(x_t) \right), \tag{1}$$

and CRGD, see [Appendix D.2]. We can use CRGD for unconstrained problems even though the analysis only covers the compact case, since the composite condition number of $\tilde{L}_t$, i.e., $L\eta = 1/4$, is smaller than one and we show that in this case, the distance of the iterates of CRGD to the optimizer is non-increasing [Corollary 24]. Hence we can add a hypothetical constraint which contains a ball around the optimizer with radius slightly larger than the initial distance to the optimizer, as this is guaranteed to never be inactive.

**Corollary 6** (Implementing RIODA – unconstrained). [↓] *Consider the setting of [Theorem 5]. Assume in addition that $f$ is g-convex, g-concave and $L$-smooth in $\bar{B}(x^*, 8R) \times \bar{B}(y^*, 8R)$. Then we require $\widetilde{O}(1)$ gradient oracle calls for implementing the update rules in Lines [2] and [3] of [Algorithm 2] using CRGD (this includes a logarithmic dependence on $|\kappa_{\min}|$) and the iterates stay in that set. If $f$ is g-convex, g-concave and $L$-smooth in $\bar{B}(x^*, \bar{D}) \times \bar{B}(y^*, \bar{D})$ with $\bar{D} \stackrel{\text{def}}{=} R(13\zeta_{8R} + 9)$, then we require $\widetilde{O}(\zeta_R^2)$ gradient oracle calls using RGD and the iterates stay in that set. We refer to these algorithms as RIODA$_{CRGD}$ and RIODA$_{RGD}$, respectively. Neither method requires prior knowledge of the initial distance to the saddle point $R$.*

Compared to RIPPA ([Martínez-Rubio et al., 2023]), the only other min-max algorithm which does not require knowledge of $R$, the complexity of RIODA$_{RGD}$ is reduced by a factor of $\widetilde{O}(\zeta_R^2)$. The rates of RIODA$_{CRGD}$ recover the optimal Euclidean rates and are independent $\zeta$, up to log factors. Further, note that implementing the subroutines of RAMMA with CRGD leads to total oracle complexities of $\widetilde{O}(\zeta_R^{2.5} \frac{LR^2}{\varepsilon})$ and $\widetilde{O}(\zeta_R^{2.5} \frac{L}{\mu})$ for $\mu=0$ and $\mu>0$, respectively.

This means that the rates of RIODA$_{CRGD}$ improve over the prior state of the art in the constrained setting by $O(\zeta_R^{2.5})$.

We empirically validate our theoretical results in [Appendix E] on a constrained formulation of the robust Karcher mean problem, testing RIODA$_{PRGD}$ on both the symmetric positive definite manifold and the hyperbolic space.

It might seem that the nearly curvature-independent rates of $T = \widetilde{O}(LR^2/\varepsilon)$ and $T = \widetilde{O}(L/\mu)$ for $\mu = 0$ and $\mu > 0$, respectively, achieved by RIODA$_{CRGD}$, are incompatible with existing lower bounds. Indeed, [Criscitiello & Boumal] ([2023]) established a lower bound of $T = \widetilde{\Omega}(\zeta_R)$ gradient oracle queries for $L$-smooth and g-convex minimization problems in the hyperbolic space, which is a special case of our g-convex, g-concave and $L$-smooth min-max setting.

For $\mu = 0$, the lower bound assumes that $\varepsilon = O(LR^2/\zeta_R)$, which implies that our upper bound of $T = \widetilde{O}(LR^2/\varepsilon)$ is larger than $\widetilde{\Omega}(\zeta_R)$, showing that the lower and upper bounds are consistent. Note that the lower bound is based on a function defined in the hyperbolic space, where the initial gap is $\widetilde{O}(LR^2/\zeta_R)$ ([Criscitiello & Boumal], [2023], Proposition 13), meaning that their assumption on $\varepsilon$ is justified.

For $\mu > 0$, we note that the condition number of functions in the hyperbolic space in a ball of radius $R$ is lower bounded by $L/\mu \geq \zeta_R$ ([Martínez-Rubio], [2020], Proposition 29). Hence our upper bound does not contradict the lower bound. [Criscitiello & Boumal] ([2021]) also develop similar lower bounds for the case $\mu > 0$ in a broader class of Hadamard manifolds. A similar argument via ([Criscitiello & Boumal], [2023], Proposition 53) shows there is no contradiction with our bounds either.

This shows that while the rates of RIODA$_{CRGD}$ do not explicitly depend on $\zeta_R$ (up to a log factor), they inevitably have an implicit dependence on the geometry, which is due to the fact above regarding the minimum condition number for this class of problems depending on the geometry. For well-conditioned functions or if the required precision is not too high, the hardness arising from the geometry dominates whereas whenever $L/\mu \gg \zeta_R$ or $LR^2/\varepsilon \gg \zeta_R$ the hardness arising from the function dominates.

## 4. Conclusion

We presented RIOD, a new optimistic online learning algorithm for Riemannian manifolds which can handle in-manifold constraints and achieves regret bounds that match the best known Euclidean rates. Based on RIOD, we developed RIODA, a novel algorithm for min-max optimization on Riemannian manifolds. We proposed multiple implementations of RIODA, with one variant achieving convergence rates that match the Euclidean rates up to logarithmic factors, for the first time.

## Impact Statement

This paper presents work whose goal is to advance the field of Machine Learning. There are many potential societal consequences of our work, none which we feel must be specifically highlighted here.

## Acknowledgements

This research was partially funded by the Research Campus Modal funded by the German Federal Ministry of Education and Research (fund numbers 05M14ZAM,05M20ZBM) as well as the Deutsche Forschungsgemeinschaft (DFG) through the DFG Cluster of Excellence MATH$^+$ (EXC-2046/1, project ID 390685689). David Martínez-Rubio was partially funded by the project IDEA-CM (TEC-2024/COM-89).

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

## A. RIOD proof

**Theorem 1** (RIOD). [↓] *Let $\mathcal{M}$ be a Hadamard manifold with sectional curvature in $[\kappa_{\min}, 0]$. Further, for $t \in [T]$, let the loss and hint functions $\ell_t, \tilde{\ell}_t : \mathcal{M} \to \mathbb{R}$ be g-convex, and L-smooth in a compact and g-convex set $\mathcal{X} \subseteq \mathcal{M}$ with $D \overset{\text{def}}{=} \operatorname{diam}(\mathcal{X})$. For $\eta > 0$ we have for any $u \in \mathcal{X}$ that*

$$R_T(u) \leq \frac{3D^2}{2\eta} + \eta \sum_{t=1}^{T} \|\nabla \ell_t(\tilde{x}_t) - \nabla \tilde{\ell}_t(\tilde{x}_t)\|_{\tilde{x}_t}^2.$$

*Proof.* (Theorem 1) The proof follows directly from Theorem 7 by setting $u_t = u$ for all $t \in [T]$ and $\mu = 0$. $\qquad \square$

**Theorem 7** (RIOD). *Let $\mathcal{M}$ be a Hadamard manifold with sectional curvature in $[\kappa_{\min}, 0]$. Further, let $\ell_t, \tilde{\ell}_t : \mathcal{M} \to \mathbb{R}$ be $\mu$-strongly g-convex, and L-smooth in a compact and g-convex set $\mathcal{X} \subseteq \mathcal{M}$ with $\operatorname{diam}(\mathcal{X}) = D$ and $\mu \geq 0$. Then for $\eta > 0$ we have for any $(u_t)_{t \in [T]} \in \mathcal{X}$ that*

$$\sum_{t=1}^{T} \ell_t(\tilde{x}_t) - \ell_t(u_t) \leq \frac{2P_T D + 3D^2}{2\eta}$$
$$+ \sum_{t=1}^{T} \left( \eta \|\nabla \ell_t(\tilde{x}_t) - \nabla \tilde{\ell}_t(\tilde{x}_t)\|^2 - \frac{\mu}{4} d(x_{t+1}, u_t)^2 \right),$$

*where $P_T \overset{\text{def}}{=} \sum_{t=1}^{T} d(u_t, u_{t+1})$.*

*Proof of Theorem 7.* Using the notation of Algorithm 1, we define $\bar{\varepsilon}_t \overset{\text{def}}{=} 2\eta \varepsilon_t$ and recall that $L_t(x) = \ell_t(x) + \frac{1}{2\eta} d(x, x_t)^2$ and $\tilde{L}_t(x) = \tilde{\ell}_t(x) + \frac{1}{2\eta} d(x, x_t)^2$. Note that $L_t$ and $\tilde{L}_t$ are $(1/\eta)$-strongly g-convex in $\mathcal{X}$, because the regularizer $\frac{1}{2\eta} d(x, x_t)^2$ is $(1/\eta)$-strongly g-convex in $\mathcal{X}$ by the first part of Fact 15 since $\nabla_x(\frac{1}{2} d(x, y)^2) = -\operatorname{Log}_x(y)$ and $\mathcal{M}$ is a Hadamard manifold and so $\delta_D = 1$. By the $(1/\eta)$-strong g-convexity of $L_t$ and $\tilde{L}_t$ in $\mathcal{X}$, the error criteria of $x_{t+1}$ and $\tilde{x}_t$ in Algorithm 1 and the optimality of $x_{t+1}^*$ and $\tilde{x}_t^*$, we have that

$$d(x_{t+1}, x_{t+1}^*)^2 \leq 2\eta(F_t(x_{t+1}) - F_t(x_{t+1}^*)) \leq \bar{\varepsilon}_t d(x_t, x_{t+1}^*)^2 \tag{2}$$

and

$$d(\tilde{x}_t, \tilde{x}_t^*)^2 \leq 2\eta(\tilde{L}_t(\tilde{x}_t) - \tilde{L}_t(\tilde{x}_t^*)) \leq \bar{\varepsilon}_t d(x_t, \tilde{x}_t^*)^2. \tag{3}$$

We also have

$$\ell_t(x_{t+1}) - \ell_t(u_t) = \ell_t(x_{t+1}) - \ell_t(x_{t+1}^*) + \ell_t(x_{t+1}^*) - \ell_t(u_t)$$

$$\overset{\text{①}}{\leq} \frac{1}{2\eta}((1 + \bar{\varepsilon}_t) d(x_{t+1}^*, x_t)^2 - d(x_{t+1}, x_t)^2) + \langle \nabla \ell_t(x_{t+1}^*), -\operatorname{Log}_{x_{t+1}^*}(u_t) \rangle - \frac{\mu}{2} d(x_{t+1}^*, u_t)^2$$

$$\overset{\text{②}}{\leq} \frac{1}{2\eta}((1 + \bar{\varepsilon}_t) d(x_{t+1}^*, x_t)^2 - d(x_{t+1}, x_t)^2) - \frac{1}{\eta} \langle \operatorname{Log}_{x_{t+1}^*}(x_t), \operatorname{Log}_{x_{t+1}^*}(u_t) \rangle - \frac{\mu}{2} d(x_{t+1}^*, u_t)^2 \tag{4}$$

$$\overset{\text{③}}{\leq} \frac{1}{2\eta} \left( d(x_t, u_t)^2 - (1 + \mu\eta) d(x_{t+1}^*, u_t)^2 - d(x_t, x_{t+1})^2 + \bar{\varepsilon}_t d(x_t, x_{t+1}^*)^2 \right)$$

where ① holds by the error criterion of $x_{t+1}$ defined in Line 6, i.e., $L_t(x_{t+1}) - L_t(x_{t+1}^*) \leq \frac{\bar{\varepsilon}_t}{2\eta} d(x_t, x_{t+1}^*)^2$, and the $\mu$-strong g-convexity of $\ell_t$ between $\tilde{x}_{t+1}^*$ and $u_t$. Further, ② holds since $x_{t+1}^*$ is the optimizer of $L_t$ and the first-order optimality condition, i.e., $\langle \nabla \ell_t(x_{t+1}^*) - \frac{1}{\eta} \operatorname{Log}_{x_{t+1}^*}(x_t), -\operatorname{Log}_{x_{t+1}^*}(u_t) \rangle \leq 0$ and ③ holds by Fact 15 noting that $\delta_D = 1$ since $\mathcal{M}$ is a Hadamard manifold. Further, we have

$$\langle \nabla \tilde{\ell}_t(\tilde{x}_t^*), -\operatorname{Log}_{\tilde{x}_t^*}(x_{t+1}) \rangle \overset{\text{①}}{\leq} -\frac{1}{\eta} \langle \operatorname{Log}_{\tilde{x}_t^*}(x_t), \operatorname{Log}_{\tilde{x}_t^*}(x_{t+1}) \rangle$$

$$\overset{\text{②}}{\leq} \frac{1}{2\eta} \left( d(x_t, x_{t+1})^2 - d(\tilde{x}_t^*, x_{t+1})^2 - d(\tilde{x}_t^*, x_t)^2 \right), \tag{5}$$

where ① holds first-order optimality condition of $\tilde{L}_t$ with minimizer $\tilde{x}_t^*$, and ② holds by Fact 15 noting that $\delta_D = 1$ as $\mathcal{M}$ is a Hadamard manifold. Using the previous inequalities, we obtain

$$
\begin{aligned}
\ell_t(\tilde{x}_t) - \ell_t(u_t) &= \ell_t(\tilde{x}_t) - \ell_t(x_{t+1}) + \ell_t(x_{t+1}) - \ell_t(u_t) \\
&\overset{①}{\le} \langle \nabla \ell_t(\tilde{x}_t), -\mathrm{Log}_{\tilde{x}_t}(x_{t+1}) \rangle + \ell_t(x_{t+1}) - \ell_t(u_t) + \langle \nabla \tilde{\ell}_t(\tilde{x}_t^*), \mathrm{Log}_{\tilde{x}_t^*}(x_{t+1}) \rangle \\
&\quad + \frac{1}{2\eta} \left( d(x_t, x_{t+1})^2 - d(\tilde{x}_t^*, x_{t+1})^2 - d(\tilde{x}_t^*, x_t)^2 \right) \\
&\overset{②}{\le} \langle \nabla \ell_t(\tilde{x}_t), -\mathrm{Log}_{\tilde{x}_t}(x_{t+1}) \rangle + \langle \nabla \tilde{\ell}_t(\tilde{x}_t^*), \mathrm{Log}_{\tilde{x}_t^*}(x_{t+1}) \rangle + \frac{1}{2\eta} \left( d(x_t, u_t)^2 - (1+\mu\eta)d(x_{t+1}^*, u_t)^2 \right) \\
&\quad + \frac{1}{2\eta} \left( -d(\tilde{x}_t^*, x_{t+1})^2 - d(\tilde{x}_t^*, x_t)^2 + \bar{\varepsilon}_t d(x_t, x_{t+1}^*)^2 \right)
\end{aligned}
\tag{6}
$$

where ① holds by (5) and the g-convexity of $\ell_t$ between $\tilde{x}_t$ and $x_{t+1}$ and ② holds by (4). We want to obtain a computable *optimism* term, i.e, a term that depends on the difference between the gradients of the loss function $\ell_t$ and the hint function $\tilde{\ell}_t$ evaluated at a point we know or can compute. Since computing the exact optimizer $\tilde{x}_t^*$ of $\tilde{L}_t$ is in general not possible, we bound $\langle \nabla \tilde{\ell}_t(\tilde{x}_t^*), \mathrm{Log}_{\tilde{x}_t^*}(x_{t+1}) \rangle$ by a term dependent on the gradient evaluated at the inexact point $\tilde{x}_t$, i.e., $\nabla \tilde{\ell}_t(\tilde{x}_t)$. Adding and subtracting terms, we get

$$
\begin{aligned}
\langle \nabla \tilde{\ell}_t(\tilde{x}_t^*), \mathrm{Log}_{\tilde{x}_t^*}(x_{t+1}) \rangle &= \langle \nabla \tilde{\ell}_t(\tilde{x}_t^*) - \Gamma_{\tilde{x}_t}^{\tilde{x}_t^*} \nabla \tilde{\ell}_t(\tilde{x}_t), \mathrm{Log}_{\tilde{x}_t^*}(x_{t+1}) \rangle + \langle \Gamma_{\tilde{x}_t}^{\tilde{x}_t^*} \nabla \tilde{\ell}_t(\tilde{x}_t), \mathrm{Log}_{\tilde{x}_t^*}(x_{t+1}) \rangle \\
&\overset{①}{=} \langle \nabla \tilde{\ell}_t(\tilde{x}_t^*) - \Gamma_{\tilde{x}_t}^{\tilde{x}_t^*} \nabla \tilde{\ell}_t(\tilde{x}_t), \mathrm{Log}_{\tilde{x}_t^*}(x_{t+1}) \rangle + \langle \nabla \tilde{\ell}_t(\tilde{x}_t), \mathrm{Log}_{\tilde{x}_t}(x_{t+1}) \rangle + \langle \Gamma_{\tilde{x}_t}^{\tilde{x}_t^*} \nabla \tilde{\ell}_t(\tilde{x}_t), \mathrm{Log}_{\tilde{x}_t^*}(x_{t+1}) - \Gamma_{\tilde{x}_t}^{\tilde{x}_t^*} \mathrm{Log}_{\tilde{x}_t}(x_{t+1}) \rangle,
\end{aligned}
\tag{7}
$$

where we used Gauß's Lemma in ①, i.e., $\langle \Gamma_{\tilde{x}_t}^{\tilde{x}_t^*} \nabla \tilde{\ell}_t(\tilde{x}_t), \Gamma_{\tilde{x}_t}^{\tilde{x}_t^*} \mathrm{Log}_{\tilde{x}_t}(x_{t+1}) \rangle_{\tilde{x}_t^*} = \langle \nabla \tilde{\ell}_t(\tilde{x}_t), \mathrm{Log}_{\tilde{x}_t}(x_{t+1}) \rangle_{\tilde{x}_t}$, for the second summand. Note that this is the term we wanted to obtain. We now go on to bound the other two terms. For the first of the two error terms, we have,

$$
\begin{aligned}
\langle \nabla \tilde{\ell}_t(\tilde{x}_t^*) - \Gamma_{\tilde{x}_t}^{\tilde{x}_t^*} \nabla \tilde{\ell}_t(\tilde{x}_t), \mathrm{Log}_{\tilde{x}_t^*}(x_{t+1}) \rangle &\overset{①}{\le} \| \nabla \tilde{\ell}_t(\tilde{x}_t^*) - \Gamma_{\tilde{x}_t}^{\tilde{x}_t^*} \nabla \tilde{\ell}_t(\tilde{x}_t) \| \cdot d(\tilde{x}_t^*, x_{t+1}) \overset{②}{\le} L d(\tilde{x}_t^*, \tilde{x}_t) d(\tilde{x}_t^*, x_{t+1}) \\
&\overset{③}{\le} \frac{1}{2} \left( 8\eta L^2 d(\tilde{x}_t^*, \tilde{x}_t)^2 + \frac{1}{8\eta} d(\tilde{x}_t^*, x_{t+1})^2 \right) \overset{④}{\le} \frac{1}{2} \left( 8\eta L^2 \bar{\varepsilon}_t d(\tilde{x}_t^*, x_t)^2 + \frac{1}{8\eta} d(\tilde{x}_t^*, x_{t+1})^2 \right),
\end{aligned}
\tag{8}
$$

where ① holds by the Cauchy-Schwarz inequality, ② holds by the $L$-smoothness of $\tilde{\ell}_t$, ③ holds by Young's inequality and ④ by (3). Before we bound the second error term, consider

$$
\begin{aligned}
\zeta_{\bar{D}}^2 &\overset{①}{\le} (1 + \sqrt{|\kappa_{\min}|}(d(\tilde{x}_t, x_{t+1}) + d(\tilde{x}_t^*, x_{t+1})))^2 \overset{②}{\le} 2(1 + 2|\kappa_{\min}|(d^2(\tilde{x}_t, x_{t+1}) + d^2(\tilde{x}_t^*, x_{t+1}))) \\
&\overset{③}{\le} 2(1 + 2|\kappa_{\min}|(2d(\tilde{x}_t, \tilde{x}_t^*)^2 + 3d(\tilde{x}_t^*, x_{t+1})^2)),
\end{aligned}
\tag{9}
$$

where $\bar{D} \overset{\mathrm{def}}{=} \max\{d(\tilde{x}_t, x_{t+1}), d(\tilde{x}_t^*, x_{t+1})\}$. Here, ① holds by the definition of $\zeta_{\bar{D}}$, ② holds by applying $(a+b)^2 \le 2a^2 + 2b^2$ for $a, b \ge 0$ twice and ③ also uses the last inequality and the triangle inequality. We now bound the second error term in (7) as follows,

$$
\begin{aligned}
\langle \Gamma_{\tilde{x}_t}^{\tilde{x}_t^*} \nabla \tilde{\ell}_t(\tilde{x}_t), \mathrm{Log}_{\tilde{x}_t^*}(x_{t+1}) - \Gamma_{\tilde{x}_t}^{\tilde{x}_t^*} \mathrm{Log}_{\tilde{x}_t}(x_{t+1}) \rangle &\overset{①}{\le} \| \nabla \tilde{\ell}_t(\tilde{x}_t) \| \cdot \| \mathrm{Log}_{\tilde{x}_t^*}(x_{t+1}) - \Gamma_{\tilde{x}_t}^{\tilde{x}_t^*} \mathrm{Log}_{\tilde{x}_t}(x_{t+1}) \| \\
&\overset{②}{\le} \zeta_{\bar{D}} \| \nabla \tilde{\ell}_t(\tilde{x}_t) \| d(\tilde{x}_t, \tilde{x}_t^*) \overset{③}{\le} \frac{\zeta_{\bar{D}}^2 E \Delta_t}{4(1 + 48|\kappa_{\min}|E\eta)} + \frac{\| \nabla \tilde{\ell}_t(\tilde{x}_t) \|^2 d(\tilde{x}_t, \tilde{x}_t^*)^2 (1 + 48|\kappa_{\min}|E\eta)}{E\Delta_t} \\
&\overset{④}{\le} \frac{E\Delta_t}{2} + \left( \frac{\| \nabla \tilde{\ell}_t(\tilde{x}_t) \|^2 (1 + 48|\kappa_{\min}|E\eta)}{E\Delta_t} + \frac{\Delta_t}{24\eta} \right) d(\tilde{x}_t, \tilde{x}_t^*)^2 + \frac{\Delta_t}{16\eta} d(\tilde{x}_t^*, x_{t+1})^2 \\
&\overset{⑤}{\le} \frac{E\Delta_t}{2} + \left( \frac{\| \nabla \tilde{\ell}_t(\tilde{x}_t) \|^2 (E^{-1} + 48|\kappa_{\min}|\eta)}{\Delta_t} + \frac{\Delta_t}{24\eta} \right) \bar{\varepsilon}_t d(x_t, \tilde{x}_t^*)^2 + \frac{\Delta_t}{16\eta} d(\tilde{x}_t^*, x_{t+1})^2,
\end{aligned}
\tag{10}
$$

where ① holds by the Cauchy-Schwarz inequality, ② holds by Proposition 18 with $\bar{D} = \max\{d(\tilde{x}_t, x_{t+1}), d(\tilde{x}_t^*, x_{t+1})\}$, ③ holds by Young's inequality for any $E > 0$ and $\Delta_t \in (0, 1)$, ④ holds by (9), ⑤ holds by (3). Using (8) and (10) to bound (6) yields,

$$\ell_t(\tilde{x}_t) - \ell_t(u_t) \leq \langle \nabla \ell_t(\tilde{x}_t) - \nabla \tilde{\ell}_t(\tilde{x}_t), -\mathrm{Log}_{\tilde{x}_t}(x_{t+1}) \rangle + \frac{1}{2\eta} \left( d(x_t, u_t)^2 - (1 + \mu\eta)d(x_{t+1}^*, u_t)^2 \right) + \frac{E\Delta_t}{2}$$

$$+ \frac{1}{2\eta} \left( \frac{\Delta_t - 7}{8} d(\tilde{x}_t^*, x_{t+1})^2 + \bar{\varepsilon}_t d(x_t, x_{t+1}^*)^2 \right)$$

$$+ \frac{1}{2\eta} \left( \left( \bar{\varepsilon}_t (2\eta\Delta_t^{-1} \|\nabla \tilde{\ell}_t(\tilde{x}_t)\|^2 (E^{-1} + 48|\kappa_{\min}|\eta) + 8\eta^2 L^2 + \Delta_t/12) - 1 \right) d(\tilde{x}_t^*, x_t)^2 \right)$$

$$\overset{①}{\leq} \langle \nabla \ell_t(\tilde{x}_t) - \nabla \tilde{\ell}_t(\tilde{x}_t), -\mathrm{Log}_{\tilde{x}_t}(x_{t+1}) \rangle + \frac{1}{2\eta} \left( d(x_t, u_t)^2 + (\Delta_t - 1)d(x_{t+1}, u_t)^2 \right) - \frac{\mu}{2} d(x_{t+1}^*, u_t)^2 + \frac{E\Delta_t}{2}$$

$$+ \frac{1}{2\eta} \left( -\frac{3}{4} d(\tilde{x}_t^*, x_{t+1})^2 + \bar{\varepsilon}_t (1 + \Delta_t^{-1}) d(x_t, x_{t+1}^*)^2 \right)$$

$$+ \frac{1}{2\eta} \left( \left( \bar{\varepsilon}_t (2\eta\Delta_t^{-1} \|\nabla \tilde{\ell}_t(\tilde{x}_t)\|^2 (E^{-1} + 48|\kappa_{\min}|\eta) + 8\eta^2 L^2 + 1) - 1 \right) d(\tilde{x}_t^*, x_t)^2 \right)$$

(11)

where ① holds by noting that $\Delta_t \in (0, 1)$ and applying the following bound to $-\frac{1}{2\eta} d(x_{t+1}^*, u_t)^2$,

$$-d(x_{t+1}^*, u_t)^2 = -d(x_{t+1}, u_t)^2 + d(x_{t+1}, u_t)^2 - d(x_{t+1}^*, u_t)^2$$

$$\overset{②}{\leq} -d(x_{t+1}, u_t)^2 + 2\langle \mathrm{Log}_{x_{t+1}}(u_t), \mathrm{Log}_{x_{t+1}}(x_{t+1}^*) \rangle$$

$$\overset{③}{\leq} -d(x_{t+1}, u_t)^2 + 2d(x_{t+1}, u_t)d(x_{t+1}, x_{t+1}^*)$$

$$\overset{④}{\leq} -d(x_{t+1}, u_t)^2 + \Delta_t d(x_{t+1}, u_t)^2 + \Delta_t^{-1} d(x_{t+1}, x_{t+1}^*)^2$$

$$\overset{⑤}{\leq} -d(x_{t+1}, u_t)^2 + \Delta_t d(x_{t+1}, u_t)^2 + \bar{\varepsilon}_t \Delta_t^{-1} d(x_t, x_{t+1}^*)^2.$$

(12)

Here ② holds by applying Fact 15 and dropping the negative term $-d(x_{t+1}, x_{t+1}^*)^2$, ③ holds by the Cauchy-Schwarz inequality, ④ holds by applying Young's inequality and ⑤ holds by (2). We now use the following bound for the strong g-convexity term $-\frac{\mu}{2} d(x_{t+1}^*, u_t)^2$ in (11),

$$-d(x_{t+1}^*, u_t)^2 \overset{①}{\leq} -\frac{1}{2} d(x_{t+1}, u_t)^2 + d(x_{t+1}, x_{t+1}^*)^2 \overset{②}{\leq} -\frac{1}{2} d(x_{t+1}, u_t)^2 + \bar{\varepsilon}_t d(x_t, x_{t+1}^*)^2,$$

(13)

where ① holds by the triangle inequality and ② holds by (2). Using (13) to bound (11), we obtain

$$\ell_t(\tilde{x}_t) - \ell_t(u_t) \leq \langle \nabla \ell_t(\tilde{x}_t) - \nabla \tilde{\ell}_t(\tilde{x}_t), -\mathrm{Log}_{\tilde{x}_t}(x_{t+1}) \rangle + \frac{1}{2\eta} \left( d(x_t, u_t)^2 + (\Delta_t - 1 - \frac{\mu\eta}{2})d(x_{t+1}, u_t)^2 \right) + \frac{E\Delta_t}{2}$$

$$+ \frac{1}{2\eta} \left( -\frac{3}{4} d(\tilde{x}_t^*, x_{t+1})^2 + \bar{\varepsilon}_t (2 + \Delta_t^{-1}) d(x_t, x_{t+1}^*)^2 \right)$$

$$+ \frac{1}{2\eta} \left( \bar{\varepsilon}_t (2\eta\Delta_t^{-1} \|\nabla \tilde{\ell}_t(\tilde{x}_t)\|^2 (E^{-1} + 48|\kappa_{\min}|\eta) + 8\eta^2 L^2 + 1) - 1 \right) d(\tilde{x}_t^*, x_t)^2.$$

(14)

Let us briefly comment on why we bounded the two terms depending on $d(x_{t+1}^*, u_t)^2$ differently. If we had bounded them both using (12), we would have obtained $(\Delta_t - 1)(1 + \mu\eta)d(x_{t+1}, u_t)^2$ and since $\Delta_t > 0$, this means that the update rule of Algorithm 1 is not a contraction in the min-max setting where all $u_t$ correspond to the saddle point $(x^*, y^*)$, see Theorem 3. On the other hand, if we had bounded both terms using (13), we would have obtained $\frac{-(1+\mu\eta)}{4\eta} d(x_{t+1}, u_t)^2$, which would have meant that this term does not telescope out with $\frac{1}{2\eta} d(x_t, u_t)^2$ later in the proof. We now bound our error dependent on

$d(x_t, x_{t+1}^*)^2$ such that we can cancel it out with the negative terms which come up in the analysis. We bound,

$$d(x_t, x_{t+1}^*)^2 \overset{①}{\leq} 2d(x_t, \tilde{x}_t^*)^2 + 2d(\tilde{x}_t^*, x_{t+1}^*)^2$$

$$\overset{②}{\leq} 2d(x_t, \tilde{x}_t^*)^2 + 4d(\tilde{x}_t^*, x_{t+1})^2 + 4d(x_{t+1}, x_{t+1}^*)^2 \qquad (15)$$

$$\overset{③}{\leq} 2d(x_t, \tilde{x}_t^*)^2 + 4d(\tilde{x}_t^*, x_{t+1})^2 + 4\bar{\varepsilon}_t d(x_t, x_{t+1}^*)^2,$$

where ① and ② follow by the triangle inequality, ③ follows by (2). Hence, by rearranging and noting that $\bar{\varepsilon}_t < 1/4$ by definition, we obtain

$$d(x_t, x_{t+1}^*)^2 \leq \frac{2}{(1 - 4\bar{\varepsilon}_t)} d(x_t, \tilde{x}_t^*)^2 + \frac{4}{1 - 4\bar{\varepsilon}_t} d(\tilde{x}_t^*, x_{t+1})^2. \qquad (16)$$

Using this inequality, we get

$$\ell_t(\tilde{x}_t) - \ell_t(u_t) \leq \langle \nabla \ell_t(\tilde{x}_t) - \nabla \tilde{\ell}_t(\tilde{x}_t), -\text{Log}_{\tilde{x}_t}(x_{t+1}) \rangle + \frac{1}{2\eta} \left( d(x_t, u_t)^2 + (\Delta_t^{-1} - 1 - \mu\eta/2)d(x_{t+1}, u_t)^2 \right)$$

$$+ \frac{1}{2\eta} \left( (C_t - 3/4)d(\tilde{x}_t^*, x_{t+1})^2 + (C_t - 1)d(\tilde{x}_t^*, x_t)^2 \right) + \frac{E\Delta_t}{2}, \qquad (17)$$

where

$$C_t \overset{\text{def}}{=} \frac{\bar{\varepsilon}_t}{1 - 4\bar{\varepsilon}_t} \left( 9 + 8\eta^2 L^2 + \Delta_t^{-1} \left( 4 + 2\eta \|\nabla \tilde{\ell}_t(\tilde{x}_t)\|^2 (E^{-1} + 48|\kappa_{\min}|\eta) \right) \right)$$

$$\geq \max \left\{ \frac{4\bar{\varepsilon}_t(2 + \Delta_t^{-1})}{1 - 4\bar{\varepsilon}_t}, \frac{2\bar{\varepsilon}_t(2 + \Delta_t^{-1})}{1 - 4\bar{\varepsilon}_t} + \bar{\varepsilon}_t \left( \frac{2\eta \|\nabla \tilde{\ell}_t(\tilde{x}_t)\|^2 (E^{-1} + 48|\kappa_{\min}|\eta)}{\Delta_t} + 8L^2\eta^2 + 1 \right) \right\}.$$

We now address the mismatch between $d(x_t, u_t)^2$ and $d(x_{t+1}, u_t)^2$,

$$\sum_{t=1}^{T} d(x_t, u_t)^2 - d(x_{t+1}, u_t)^2 = \sum_{t=1}^{T} d(x_t, u_t)^2 - d(x_{t+1}, u_{t+1})^2 + d(x_{t+1}, u_{t+1})^2 - d(x_{t+1}, u_t)^2$$

$$\overset{①}{=} d(x_1, u_1)^2 - d(x_{T+1}, u_{T+1})^2 + \sum_{t=1}^{T} d(x_{t+1}, u_{t+1})^2 - d(x_{t+1}, u_t)^2 \qquad (18)$$

$$\overset{②}{\leq} d(x_1, u_1)^2 - d(x_{T+1}, u_{T+1})^2 + 2D \underbrace{\sum_{t=1}^{T} d(u_t, u_{t+1})}_{=P_T},$$

where ① holds by telescoping the first two summands and ② holds by

$$d(x_{t+1}, u_{t+1})^2 - d(x_{t+1}, u_t)^2 \overset{③}{\leq} d(u_t, u_{t+1})(d(x_{t+1}, u_{t+1}) + d(x_{t+1}, u_t)) \overset{④}{\leq} 2d(u_t, u_{t+1})D,$$

where we apply the triangle inequality $d(x_{t+1}, u_{t+1}) \leq d(x_{t+1}, u_t) + d(u_t, u_{t+1})$ in ③, and in ④, we use that $d(x_{t+1}, u_{t+1}), d(x_{t+1}, u_t) \leq D$. Summing (17) from $t = 1$ to $T$ and using (18), we get

$$\sum_{t=1}^{T} \ell_t(\tilde{x}_t) - \ell_t(u_t) \leq \frac{d(x_1, u_1)^2 - d(x_{T+1}, u_{T+1})^2 + 2P_T D}{2\eta} + \sum_{t=1}^{T} \langle \nabla \ell_t(\tilde{x}_t) - \nabla \tilde{\ell}_t(\tilde{x}_t), -\text{Log}_{\tilde{x}_t}(x_{t+1}) \rangle$$

$$+ \frac{1}{2\eta} \sum_{t=1}^{T} \left[ (C_t - 3/4)d(\tilde{x}_t^*, x_{t+1})^2 + (C_t - 1)d(\tilde{x}_t^*, x_t)^2 - \frac{\mu\eta}{2}d(x_{t+1}, u_t)^2 \right] + \sum_{t=1}^{T} \frac{\Delta_t(E\eta + d(x_{t+1}, u_t)^2)}{2\eta}. \qquad (19)$$

Further, choosing $E \leftarrow D^2/\eta$ and $\Delta_t = (t + 1)^{-2}$, we have that

$$\frac{1}{2\eta} \sum_{t=1}^{T} \Delta_t(D^2 + d(x_{t+1}, u_t)^2) \overset{①}{\leq} \frac{D^2}{\eta} \sum_{t=1}^{T} \Delta_t \overset{②}{\leq} \frac{D^2}{\eta}, \qquad (20)$$

where ① holds since the $x_{t+1}$ and $u_t$ lie in $\mathcal{X}$ and ② holds by Proposition 14. We obtain

$$
\begin{aligned}
\sum_{t=1}^{T} \ell_t(\tilde{x}_t) - \ell_t(u_t) &\leq \frac{d(x_1, u_1)^2 - d(x_{T+1}, u_{T+1})^2 + 2P_T D + 2D^2}{2\eta} + \sum_{t=1}^{T} \langle \nabla \ell_t(\tilde{x}_t) - \nabla \tilde{\ell}_t(\tilde{x}_t), -\mathrm{Log}_{\tilde{x}_t}(x_{t+1}) \rangle \\
&\quad + \frac{1}{2\eta} \sum_{t=1}^{T} \left[ (C_t - 3/4)d(\tilde{x}_t^*, x_{t+1})^2 + (C_t - 1)d(\tilde{x}_t^*, x_t)^2 - \frac{\mu\eta}{2}d(x_{t+1}, u_t)^2 \right].
\end{aligned}
\tag{21}
$$

Further, we bound

$$
\langle \nabla \ell_t(\tilde{x}_t) - \nabla \tilde{\ell}_t(\tilde{x}_t), -\mathrm{Log}_{\tilde{x}_t}(x_{t+1}) \rangle \overset{①}{\leq} \|\nabla \ell_t(\tilde{x}_t) - \nabla \tilde{\ell}_t(\tilde{x}_t)\| \cdot d(\tilde{x}_t, x_{t+1}) \overset{②}{\leq} \eta\|\nabla \ell_t(\tilde{x}_t) - \nabla \tilde{\ell}_t(\tilde{x}_t)\|^2 + \frac{1}{4\eta}d(\tilde{x}_t, x_{t+1})^2
$$

$$
\overset{③}{\leq} \eta\|\nabla \ell_t(\tilde{x}_t) - \nabla \tilde{\ell}_t(\tilde{x}_t)\|^2 + \frac{1}{2\eta}(\bar{\varepsilon}_t d(\tilde{x}_t, \tilde{x}_t^*)^2 + d(\tilde{x}_t^*, x_{t+1})^2),
$$

where we use Cauchy-Schwarz in ①, Young's inequality in ② and the triangle inequality and (3) in ③. Bounding $d(x_1, u_1)^2 \leq D^2$, we obtain

$$
\begin{aligned}
\sum_{t=1}^{T} \ell_t(\tilde{x}_t) - \ell_t(u_t) &\leq \frac{2P_T D + 3D^2}{2\eta} + \sum_{t=1}^{T} \left( \eta\|\nabla \ell_t(\tilde{x}_t) - \nabla \tilde{\ell}_t(\tilde{x}_t)\|^2 - \frac{\mu}{4}d(x_{t+1}, u_t)^2 \right) \\
&\quad + \frac{1}{2\eta} \sum_{t=1}^{T} \left( (C_t + \bar{\varepsilon}_t - 1)d(x_t, \tilde{x}_t)^2 + (C_t - 1/4)d(\tilde{x}_t^*, x_{t+1})^2 \right) \\
&\overset{①}{\leq} \frac{2P_T D + 3D^2}{2\eta} + \sum_{t=1}^{T} \left( \eta\|\nabla \ell_t(\tilde{x}_t) - \nabla \tilde{\ell}_t(\tilde{x}_t)\|^2 - \frac{\mu}{4}d(x_{t+1}, u_t)^2 \right).
\end{aligned}
$$

Here ① holds because our choice of $\varepsilon_t$ ensures that $C_t + \bar{\varepsilon}_t \leq \frac{1}{4}$. □

## B. RIODA proof

**Lemma 8** (Online-to-Minmax Reduction). *Let $\mathcal{M}, \mathcal{N}$ be Hadamard manifolds with sectional curvature in $[\kappa_{\min}, 0]$ and $\mathcal{X} \subset \mathcal{M}, \mathcal{Y} \subset \mathcal{N}$ be compact and g-convex sets. Consider the bi-function $f : \mathcal{M} \times \mathcal{N} \to \mathbb{R}$, which is $\mu$-SCSC and $L$-smooth in $\mathcal{X} \times \mathcal{Y}$. Further, let $(x^*, y^*)$ be a saddle point of $f$. Consider two instantiations of Algorithm 1 running for the $x$ and $y$ variables in parallel. In particular, for $x$, let $\tilde{x}_t \leftarrow \tilde{x}_t, x_t \leftarrow x_t, \ell_t(x) \leftarrow f(x, \tilde{y}_t), \tilde{\ell}_t(x) \leftarrow f(x, y_t), \varepsilon_t \leftarrow \varepsilon_t^x, E \leftarrow E_y$ and $u_t \leftarrow x^*$. For $y$, let $x_t \leftarrow y_t, \tilde{x}_t \leftarrow \tilde{y}_t, \ell_t(x) \leftarrow -f(\tilde{x}_t, y), \tilde{\ell}_t(x) \leftarrow -f(x_t, y), \varepsilon_t \leftarrow \varepsilon_t^y, E \leftarrow E_y$ and $u_t \leftarrow y^*$. Then we have that*

$$
\begin{aligned}
f(\tilde{x}_t, y^*) - f(x^*, \tilde{y}_t) &\leq \frac{1}{2\eta} \left( d(y_t, y^*)^2 + d(x_t, x^*)^2 + (\Delta_t - 1 - \mu\eta/2)(d(y_{t+1}, y^*)^2 + d(x_{t+1}, x^*)^2) \right) \\
&\quad + \left( L + \frac{(C_t^x - 3/4)}{2\eta} \right) d(\tilde{x}_t^*, x_{t+1})^2 + \left( L + \frac{(C_t^y - 3/4)}{2\eta} \right) d(\tilde{y}_t^*, y_{t+1})^2 + (E_x + E_y)\Delta_t/2 \\
&\quad + \left( L(1 + 2\bar{\varepsilon}_t) + \frac{(C_t^y + 4\bar{\varepsilon}_t - 1)}{2\eta} \right) d(\tilde{y}_t^*, y_t)^2 + \left( L(1 + 2\bar{\varepsilon}_t) + \frac{(C_t^x + 4\bar{\varepsilon}_t - 1)}{2\eta} \right) d(\tilde{x}_t^*, x_t)^2,
\end{aligned}
$$

*where $\bar{\varepsilon}_t \overset{\text{def}}{=} 2\eta \max\{\varepsilon_t^x, \varepsilon_t^y\}$ and*

$$
\begin{aligned}
C_t^x &\overset{\text{def}}{=} \frac{\bar{\varepsilon}_t}{1 - 4\bar{\varepsilon}_t} \left( 9 + 8\eta^2 L^2 + \Delta_t^{-1} \left( 4 + 2\eta\|\nabla_x f(\tilde{x}_t, y_t)\|^2 (E_x^{-1} + 48|\kappa_{\min}|\eta) \right) \right) \\
C_t^y &\overset{\text{def}}{=} \frac{\bar{\varepsilon}_t}{1 - 4\bar{\varepsilon}_t} \left( 9 + 8\eta^2 L^2 + \Delta_t^{-1} \left( 4 + 2\eta\|\nabla_y f(x_t, \tilde{y}_t)\|^2 (E_y^{-1} + 48|\kappa_{\min}|\eta) \right) \right).
\end{aligned}
\tag{22}
$$

*Proof.* Following the proof of Theorem 7 until (17) for both the instantiations of Algorithm 1 updating $x$ and $y$ and summing

---

**Algorithm 2** Riemannian Implicit Optimistic Gradient Descent-Ascent (RIODA)

---

**Input:** Sets $\mathcal{X} \subseteq \mathcal{M}$, $\mathcal{Y} \subseteq \mathcal{N}$, sectional curvature $\kappa_{\min}$, initial points $(x_1, y_1) \in \mathcal{X} \times \mathcal{Y}$, smoothness and strong g-convexity constants $L$ and $\mu$ of $f$ and final precision $\varepsilon$ or total number of iterations $T$.

---

**Definitions:**                                    ⋄ *The algorithm does not compute these quantities.*

- Proximal parameter $\eta \leftarrow \frac{1}{4L}$

- $H_t(x, y) \overset{\text{def}}{=} f(x, y) + \frac{1}{2\eta} d(x, x_t)^2 - \frac{1}{2\eta} d(y, y_t)^2$

- Exact solutions:

  $\tilde{x}_t^* \overset{\text{def}}{=} \underset{x \in \mathcal{X}}{\arg\min}\, H_t(x, y_t)$ and $x_{t+1}^* \overset{\text{def}}{=} \underset{x \in \mathcal{X}}{\arg\min}\, H_t(x, \tilde{y}_t)$

  $\tilde{y}_t^* \overset{\text{def}}{=} \underset{y \in \mathcal{Y}}{\arg\max}\, H_t(x_t, y)$ and $x_{t+1}^* \overset{\text{def}}{=} \underset{x \in \mathcal{X}}{\arg\min}\, H_t(\tilde{x}_t, y)$

- Constrained case:                              ⋄ Knowledge of $G$ is not required, see [Corollary 4]

  $\mu = 0$: $\varepsilon_t = L \min\left\{ 1/8, \left((t+1)^2(40 + G^2/L(\varepsilon/6 + 12|\kappa_{\min}|/L))\right)^{-1} \right\}$

  $\mu > 0$: $\varepsilon_t = L \min\left\{ 1/8, \left(\max\{(t+1)^2, 16L/\mu\}(40 + G^2/L(\varepsilon/4 + 12|\kappa_{\min}|/L))\right)^{-1} \right\}$

- Unconstrained case:                            ⋄ Knowledge of $R$ is not required, see [Corollary 6]

  $\mu = 0$: $\varepsilon_t = \frac{L}{8} \min\left\{ 1, \left(2(t+1)^2(37 + 2385R^2|\kappa_{\min}|)\right)^{-1} \right\}$

  $\mu > 0$: $\varepsilon_t = \frac{L}{8} \min\left\{ 1, \mu\left(8L(37 + 2385R^2|\kappa_{\min}|)\right)^{-1} \right\}$

---

1: **for** $t = 1$ **to** $T$ **do**
2:    $\tilde{x}_t \leftarrow (\varepsilon_t d(x_t, \tilde{x}_t^*)^2)$-minimizer of $H_t(x, y_t)$,    $\tilde{y}_t \leftarrow (\varepsilon_t d(y_t, \tilde{y}_t^*)^2)$-maximizer of $H_t(x_t, y)$
3:    $x_{t+1} \leftarrow (\varepsilon_t d(x_t, x_{t+1}^*)^2)$-minimizer of $H_t(x, \tilde{y}_t)$,    $y_{t+1} \leftarrow (\varepsilon_t d(y_t, y_{t+1}^*)^2)$-maximizer of $H_t(\tilde{x}_t, y)$
4: **end for**

---

**Output:** $\tilde{x}_T, \tilde{y}_T$ **if** $\mu > 0$, **else** uniform geodesic average of $(\tilde{x}_1, \tilde{y}_1), \ldots, (\tilde{x}_T, \tilde{y}_T)$ defined by (GEO-AVG).

---

the bounds, we obtain,

$$f(\tilde{x}_t, y^*) - f(x^*, y^*) + f(x^*, y^*) - f(x^*, \tilde{y}_t)$$
$$\leq \frac{1}{2\eta}\left(d(y_t, y^*)^2 + d(x_t, x^*)^2 + (\Delta_t - 1 - \mu\eta/2)(d(y_{t+1}, y^*)^2 + d(x_{t+1}, x^*)^2)\right)$$
$$+ \langle \nabla_y f(x_t, \tilde{y}_t) - \nabla_y f(\tilde{x}_t, \tilde{y}_t), -\mathrm{Log}_{\tilde{y}_t}(y_{t+1}) \rangle + \langle \nabla_x f(\tilde{x}_t, \tilde{y}_t) - \nabla_x f(\tilde{x}_t, y_t), -\mathrm{Log}_{\tilde{x}_t}(x_{t+1}) \rangle$$
$$+ \frac{(C_t^x - 3/4)}{2\eta} d(\tilde{x}_t^*, x_{t+1})^2 + \frac{(C_t^y - 3/4)}{2\eta} d(\tilde{y}_t^*, y_{t+1})^2 + \frac{(C_t^x - 1)}{2\eta} d(\tilde{x}_t^*, x_t)^2 + \frac{(C_t^y - 1)}{2\eta} d(\tilde{y}_t^*, y_t)^2 + (E_x + E_y)\Delta_t/2.$$
$$(23)$$

We have that

$$\langle \nabla_y f(x_t, \tilde{y}_t) - \nabla_y f(\tilde{x}_t, \tilde{y}_t), -\mathrm{Log}_{\tilde{y}_t}(y_{t+1}) \rangle + \langle \nabla_x f(\tilde{x}_t, \tilde{y}_t) - \nabla_x f(\tilde{x}_t, y_t), -\mathrm{Log}_{\tilde{x}_t}(x_{t+1}) \rangle$$

$$\overset{①}{\leq} \|\nabla_y f(x_t, \tilde{y}_t) - \nabla_y f(\tilde{x}_t, \tilde{y}_t)\| \cdot d(\tilde{y}_t, y_{t+1}) + \|\nabla_x f(\tilde{x}_t, \tilde{y}_t) - \nabla_x f(\tilde{x}_t, y_t)\| \cdot d(\tilde{x}_t, x_{t+1})$$

$$\overset{②}{\leq} Ld(\tilde{x}_t, x_t)d(\tilde{y}_t, y_{t+1}) + Ld(\tilde{y}_t, y_t)d(\tilde{x}_t, x_{t+1})$$

$$\overset{③}{\leq} \frac{L}{2}\left(d(\tilde{x}_t, x_t)^2 + d(\tilde{y}_t, y_{t+1})^2 + d(\tilde{y}_t, y_t)^2 + d(\tilde{x}_t, x_{t+1})^2\right)$$

$$\overset{④}{\leq} L(1 + 2\bar{\varepsilon}_t)(d(x_t, \tilde{x}_t^*)^2 + d(y_t, \tilde{y}_t^*)^2) + L(d(\tilde{y}_t^*, y_{t+1})^2 + d(\tilde{x}_t^*, x_{t+1})^2).$$

Here we use the Cauchy-Schwarz inequality in ①, ② follows by applying the $L$-smoothness of $f$, Young's inequality in ③

and ④ by the triangle inequality and the error criteria of $\tilde{x}_t$ and $\tilde{y}_t$, i.e.,

$$d(\tilde{x}_t, x_{t+1})^2 + d(\tilde{x}_t, x_t)^2 \le 2d(\tilde{x}_t, \tilde{x}_t^*) + 2d(\tilde{x}_t^*, x_t)^2 + 2d(\tilde{x}_t, \tilde{x}_t^*) + 2d(\tilde{x}_t^*, x_{t+1})^2 \le 4\bar{\varepsilon}_t(x_t, \tilde{x}_t^*) + 2d(\tilde{x}_t^*, x_t)^2 + 2d(\tilde{x}_t^*, x_{t+1})^2,$$

and analogously for $y$. It follows that

$$f(\tilde{x}_t, y^*) - f(x^*, \tilde{y}_t) \le \frac{1}{2\eta}\left(d(y_t, y^*)^2 + d(x_t, x^*)^2 + (\Delta_t - 1 - \mu\eta/2)(d(y_{t+1}, y^*)^2 + d(x_{t+1}, x^*)^2)\right)$$
$$+ \left(L + \frac{(C_t^x - 3/4)}{2\eta}\right)d(\tilde{x}_t^*, x_{t+1})^2 + \left(L + \frac{(C_t^y - 3/4)}{2\eta}\right)d(\tilde{y}_t^*, y_{t+1})^2 + (E_x + E_y)\Delta_t/2$$
$$+ \left(L(1 + 2\bar{\varepsilon}_t) + \frac{(C_t^y + 4\bar{\varepsilon}_t - 1)}{2\eta}\right)d(\tilde{y}_t^*, y_t)^2 + \left(L(1 + 2\bar{\varepsilon}_t) + \frac{(C_t^x + 4\bar{\varepsilon}_t - 1)}{2\eta}\right)d(\tilde{x}_t^*, x_t)^2,$$

which concludes the proof. □

**Theorem 3** (RIODA). [↓] *Let $\mathcal{M}, \mathcal{N}$ be Hadamard manifolds with sectional curvature in $[\kappa_{\min}, 0]$ and $\mathcal{X} \subset \mathcal{M}, \mathcal{Y} \subset \mathcal{N}$ be compact and g-convex sets. Consider the $f : \mathcal{M} \times \mathcal{N} \to \mathbb{R}$, which is g-convex, g-concave and L-smooth in $\mathcal{X} \times \mathcal{Y}$. Further, let $(x^*, y^*)$ be a saddle point of (P) and $(\tilde{x}_T, \tilde{y}_T)$ be the output of Algorithm 2 after $T$ iterations. Then we have $f(\tilde{x}_T, y^*) - f(x^*, \tilde{y}_T) \le \varepsilon$ after $T = \lceil \frac{8LR^2}{\varepsilon} \rceil$ iterations, and $T = \lceil \frac{17L}{\mu} \log\left(\frac{4LR^2}{\varepsilon}\right)\rceil$, if $f$ is also $\mu$-strongly g-convex, strongly g-concave in $\mathcal{X} \times \mathcal{Y}$.*

*Proof.* (Theorem 3) Let $\bar{\varepsilon}_t \overset{\text{def}}{=} 2\eta \max\{\varepsilon_t^x, \varepsilon_t^y\}$. Hence by (2) and (3), we have that

$$d(x_{t+1}, x_{t+1}^*)^2 \le \bar{\varepsilon}_t d(x_t, x_{t+1}^*)^2, d(\tilde{x}_t, \tilde{x}_t^*)^2 \le \bar{\varepsilon}_t d(x_t, \tilde{x}_t^*)^2, d(y_{t+1}, y_{t+1}^*)^2 \le \bar{\varepsilon}_t d(y_t, y_{t+1}^*)^2, d(\tilde{y}_t, \tilde{y}_t^*)^2 \le \bar{\varepsilon}_t d(y_t, \tilde{y}_t^*)^2. \tag{24}$$

By Lemma 8, we have that

$$f(\tilde{x}_t, y^*) - f(x^*, \tilde{y}_t) \le \frac{1}{2\eta}\left(d(y_t, y^*)^2 + d(x_t, x^*)^2 + (\Delta_t - 1 - \mu\eta/2)(d(y_{t+1}, y^*)^2 + d(x_{t+1}, x^*)^2)\right)$$
$$+ \left(L + \frac{(C_t^x - 3/4)}{2\eta}\right)d(\tilde{x}_t^*, x_{t+1})^2 + \left(L + \frac{(C_t^y - 3/4)}{2\eta}\right)d(\tilde{y}_t^*, y_{t+1})^2 + (E_x + E_y)\Delta_t/2 \tag{25}$$
$$+ \left(L(1 + 2\bar{\varepsilon}_t) + \frac{(C_t^y + 4\bar{\varepsilon}_t - 1)}{2\eta}\right)d(\tilde{y}_t^*, y_t)^2 + \left(L(1 + 2\bar{\varepsilon}_t) + \frac{(C_t^x + 4\bar{\varepsilon}_t - 1)}{2\eta}\right)d(\tilde{x}_t^*, x_t)^2,$$

with

$$C_t^x \overset{\text{def}}{=} \frac{\bar{\varepsilon}_t}{1 - 4\bar{\varepsilon}_t}\left(9 + 8\eta^2 L^2 + \Delta_t^{-1}\left(4 + \frac{2\eta\|\nabla_x f(\tilde{x}_t, y_t)\|^2(E_x^{-1} + 48|\kappa_{\min}|\eta)}{\Delta_t}\right)\right)$$
$$C_t^y \overset{\text{def}}{=} \frac{\bar{\varepsilon}_t}{1 - 4\bar{\varepsilon}_t}\left(9 + 8\eta^2 L^2 + \Delta_t^{-1}\left(4 + \frac{2\eta\|\nabla_y f(x_t, \tilde{y}_t)\|^2(E_y^{-1} + 48|\kappa_{\min}|\eta)}{\Delta_t}\right)\right).$$

By our choice of $\varepsilon_t$ and $\eta$ as well as $E_x$, $E_y$ and $\Delta_t$, which we specify below for $\mu = 0$ and $\mu > 0$ separately, we have that $\max\{C_t^x, C_t^y\} + 5\bar{\varepsilon}_t \le 1/4$ and hence (25) can be bounded by

$$f(\tilde{x}_t, y^*) - f(x^*, \tilde{y}_t) \le \frac{1}{2\eta}\left(d(y_t, y^*)^2 + d(x_t, x^*)^2 + (\Delta_t - 1 - \mu\eta/2)(d(y_{t+1}, y^*)^2 + d(x_{t+1}, x^*)^2)\right) + (E_x + E_y)\Delta_t/2. \tag{26}$$

We now analyze the g-convex and the strongly g-convex cases separately.

**Case $\mu = 0$.** We set $\Delta_t = (t + 1)^{-2}$ and $E_x = E_y = \varepsilon/6$. By definition, the LHS of (26) is non-negative, hence by dropping it and rearranging we obtain

$$d(y_{t+1}, y^*)^2 + d(x_{t+1}, x^*)^2 \le (1 - \Delta_t)^{-1}(d(y_t, y^*)^2 + d(x_t, x^*)^2 + \varepsilon\eta\Delta_t/3)$$
$$\overset{①}{\le} (d(y_1, y^*)^2 + d(x_1, x^*)^2)\prod_{i=1}^{t}\frac{1}{1 - \Delta_i} + \frac{\varepsilon\eta}{2}\sum_{i=1}^{t}\Delta_t \overset{②}{\le} 2(d(y_1, y^*)^2 + d(x_1, x^*)^2) + \varepsilon\eta/2 \tag{27}$$

where ① follows by repeatedly applying this inequality and since $(1 - \Delta_t)^{-1} \leq 4/3$ and ② holds by definition of $\Delta_t$ and Proposition 10. Summing (26) from $t = 1$ to $T$, dividing by $T$ and telescoping the sum, it follows that

$$\frac{1}{T} \sum_{t=1}^{T} (f(\tilde{x}_t, y^*) - f(x^*, \tilde{y}_t)) \leq \frac{1}{2\eta T} R^2 + \sum_{t=1}^{T} \left[ \frac{\Delta_t(d(y_{t+1}, y^*)^2 + d(x_{t+1}, x^*)^2)}{2\eta T} + \frac{\varepsilon \Delta_t}{4T} \right]$$

$$\overset{①}{\leq} \frac{R^2}{\eta T} + \frac{\varepsilon}{2T} \overset{②}{\leq} \frac{4LR^2}{T} + \frac{\varepsilon}{2},$$

where ① holds by (27), Proposition 14 and dropping the negative terms and ② by the definition of $\eta = 1/(4L)$ and $T \geq 1$. By applying (GEO-AVG) to the sequence $(\tilde{x}_t, \tilde{y}_t)$, we obtain

$$f(\bar{x}_T, y^*) - f(x^*, \bar{y}_T) \leq \frac{1}{T} \sum_{t=1}^{T} (f(\tilde{x}_t, y^*) - f(x^*, \tilde{y}_t)).$$

Hence after $T = \lceil \frac{8LR^2}{\varepsilon} \rceil$ iterations of Algorithm 2, we have that $f(\bar{x}_T, y^*) - f(x^*, \bar{y}_T) \leq \varepsilon$. This concludes the proof for the g-convex case.

**Case $\mu > 0$.** We set $\Delta_t = \min\{(t+1)^{-2}, \frac{\mu\eta}{4}\}$, $E_x = E_y = \varepsilon/4$. By definition, the LHS of (26) is non-negative, hence by rearranging we obtain

$$d(y_{t+1}, y^*)^2 + d(x_{t+1}, x^*)^2 \leq (1 + \mu\eta/2 - \Delta_t)^{-1}(d(y_t, y^*)^2 + d(x_t, x^*)^2) + \frac{\varepsilon\eta}{2(\Delta_t^{-1}(1 + \mu\eta/2) - 1)}$$

$$\overset{①}{\leq} (1 + \mu\eta/4)^{-1}(d(y_t, y^*)^2 + d(x_t, x^*)^2) + \frac{\varepsilon\eta}{2t^2},$$

where ① holds by the definition of $\Delta_t$. Then by repeatedly applying the inequality, we have

$$d(y_{t+1}, y^*)^2 + d(x_{t+1}, x^*)^2 \leq (1 + \mu\eta/4)^{-t} R^2 + \frac{\varepsilon\eta}{2} \sum_{k=1}^{t} k^{-2} \overset{①}{\leq} (1 + \frac{\mu}{16L})^{-t} R^2 + \frac{\varepsilon}{4L} \overset{②}{\leq} \exp\left(\frac{-t\mu}{17L}\right) R^2 + \frac{\varepsilon}{4L},$$

where ① holds since $\sum_{t=1}^{t+1} t^{-2} \leq \frac{\pi^2}{6} \leq 2$ and by definition of $\eta = 1/(4L)$ and ② holds by the following inequality

$$\frac{1}{1 + \mu/(16L)} = 1 - \frac{\mu/(16L)}{1 + \mu/(16L)} \leq 1 - \frac{\mu}{17L} \leq \exp\left(\frac{-\mu}{17L}\right), \tag{28}$$

which holds as $\mu/L \leq 1$ and $1 + x \leq \exp(x)$. Hence, running Algorithm 2 for $T = \lceil \frac{17L}{\mu} \log\left(\frac{4LR^2}{\varepsilon}\right) \rceil$ iterations implies that

$$d(y_T, y^*)^2 + d(x_T, x^*)^2 \leq \frac{\varepsilon}{2L}.$$

Dropping the non-positive distance from (26) for $t \leftarrow T$ and using the definition of $\eta$, we obtain

$$f(\tilde{x}_T, y^*) - f(x^*, \tilde{y}_T) \leq 2L(d(y_T, y^*)^2 + d(x_T, x^*)^2) \leq \varepsilon,$$

which concludes the proof. □

**Theorem 5** (RIODA – unconstrained). [↓] *Let $\mathcal{M}$, $\mathcal{N}$ be Hadamard manifolds with sectional curvature in $[\kappa_{\min}, 0]$. Consider the bi-function $f : \mathcal{M} \times \mathcal{N} \to \mathbb{R}$, which is g-convex, g-concave and L-smooth in $\mathcal{Z} = \bar{B}(x^*, 8R) \times \bar{B}(y^*, 8R)$, where $(x^*, y^*)$ is a saddle point of $f$. Then the iterates of Algorithm 2 stay in $\mathcal{Z}$. Let $(x_T, y_T)$ be the output of Algorithm 2 after $T$ iterations. Then we have $f(x_T, y^*) - f(x^*, y_T) \leq \varepsilon$ after $T = \lceil \frac{6LR^2}{\varepsilon} \rceil$ iterations and $T = \lceil \frac{17L}{\mu} \log\left(\frac{2LR^2}{\varepsilon}\right) \rceil$ if $f$ is in addition $\mu$-strongly g-convex, strongly g-concave in $\mathcal{Z}$.*

*Proof.* (Theorem 5) Let $\bar{\varepsilon}_t \overset{\text{def}}{=} 2\eta \max\{\varepsilon_t^x, \varepsilon_t^y\}$. Hence by (2) and (3), we have that

$$d(x_{t+1}, x_{t+1}^*)^2 \le \bar{\varepsilon}_t d(x_t, x_{t+1}^*)^2, d(\tilde{x}_t, \tilde{x}_t^*)^2 \le \bar{\varepsilon}_t d(x_t, \tilde{x}_t^*)^2, d(y_{t+1}, y_{t+1}^*)^2 \le \bar{\varepsilon}_t d(y_t, y_{t+1}^*)^2, d(\tilde{y}_t, \tilde{y}_t^*)^2 \le \bar{\varepsilon}_t d(y_t, \tilde{y}_t^*)^2. \tag{29}$$

We show that the iterates stay in a bounded set $\bar{B}(x^*, 7R) \times \bar{B}(y^*, 7R)$. In particular, we show via induction that $d(x_t, x^*) + d(y_t, y^*) \le 2R$ and then show that $d(\tilde{x}_t, x^*) + d(\tilde{y}_t, y^*) \le 7R$ and $d(x_{t+1}, x^*) + d(y_{t+1}, y^*) \le 7R$. For $t = 1$, we have that $d(x_1, x^*) + d(y_1, y^*) \le 2R$ by definition. Assume that $d(x_t, x^*) + d(y_t, y^*) \le 2R$ holds, now we will prove that it also holds for $t + 1$. We have that

$$d(\tilde{x}_t, x^*) + d(\tilde{y}_t, y^*) \overset{①}{\le} d(\tilde{x}_t, \tilde{x}_t^*) + d(\tilde{y}_t, \tilde{y}_t^*) + d(\tilde{x}_t^*, x^*) + d(\tilde{y}_t^*, y^*) \overset{②}{\le} \frac{d(x_t, \tilde{x}_t^*) + d(y_t, \tilde{y}_t^*)}{4} + d(\tilde{x}_t^*, x^*) + d(\tilde{y}_t^*, y^*)$$

$$\overset{③}{\le} \frac{1}{4}(d(x_t, x^*) + d(y_t, y^*)) + \frac{5}{4}(d(\tilde{x}_t^*, x^*) + d(\tilde{y}_t^*, y^*)) \overset{④}{\le} \frac{7}{2}(d(x_t, x^*) + d(y_t, y^*)), \tag{30}$$

where ① holds by the triangle inequality, ② holds by (29) and $\sqrt{\bar{\varepsilon}_t} \le 1/4$, ③ holds by the triangle inequality and ④ holds since $d(\tilde{x}_t^*, x^*) + d(\tilde{y}_t^*, y^*) \le (9/4)(d(x_t, x^*) + d(y_t, y^*))$ by Proposition 9.

Further, we have that

$$d(x_{t+1}, x^*) + d(y_{t+1}, y^*) \overset{①}{\le} d(x_{t+1}, x_{t+1}^*) + d(y_{t+1}, y_{t+1}^*) + d(x_{t+1}^*, x^*) + d(y_{t+1}^*, y^*)$$

$$\overset{②}{\le} \frac{1}{4}(d(x_t, x_{t+1}^*) + d(y_t, y_{t+1}^*)) + d(x_{t+1}^*, x^*) + d(y_{t+1}^*, y^*)$$

$$\overset{③}{\le} \frac{1}{4}(d(x_t, x^*) + d(y_t, y^*))) + \frac{5}{4}(d(x_{t+1}^*, x^*) + d(y_{t+1}^*, y^*))$$

$$\overset{④}{\le} \frac{7}{2}(d(x_t, x^*) + d(y_t, y^*)), \tag{31}$$

where ① holds by the triangle inequality, ② holds by (29) and $\sqrt{\bar{\varepsilon}_t} \le 1/4$, ③ holds by the triangle inequality and ④ holds since $d(x_{t+1}^*, x^*) + d(y_{t+1}^*, y^*) \le \frac{41}{16}(d(x_t, x^*) + d(y_t, y^*))$ by Proposition 9. We have thus established that $\tilde{x}_t$, $x_{t+1}$ and $\tilde{y}_t$, $y_{t+1}$ lie in $\bar{B}(x^*, 7R)$ and $\bar{B}(y^*, 7R)$, respectively. Recall that by assumption, $f$ is $\mu$-SCSC and $L$-smooth in $\bar{B}(x^*, 8R) \times \bar{B}(y^*, 8R)$. Therefore, we can apply Lemma 8 with $\mathcal{X} \leftarrow \bar{B}(x^*, 8R)$ and $\mathcal{Y} \leftarrow \bar{B}(y^*, 8R)$ as the iterates are guaranteed to lie in the interior of the sets and hence the constraints are never active. Thus, applying Lemma 8 with $E_x = \frac{d(x_t, \tilde{x}_t^*)^2}{8\eta}$ and $E_y = \frac{d(y_t, \tilde{y}_t^*)^2}{8\eta}$, we obtain

$$f(\tilde{x}_t, y^*) - f(x^*, \tilde{y}_t) \le \frac{1}{2\eta} \left( d(y_t, y^*)^2 + d(x_t, x^*)^2 + (\Delta_t - 1 - \mu\eta/2)(d(y_{t+1}, y^*)^2 + d(x_{t+1}, x^*)^2) \right)$$

$$+ \left( L + \frac{(C_t^x - 3/4)}{2\eta} \right) d(\tilde{x}_t^*, x_{t+1})^2 + \left( L + \frac{(C_t^y - 3/4)}{2\eta} \right) d(\tilde{y}_t^*, y_{t+1})^2$$

$$+ \left( L(1 + 2\bar{\varepsilon}_t) + \frac{(C_t^y + 4\bar{\varepsilon}_t - 3/4)}{2\eta} \right) d(\tilde{y}_t^*, y_t)^2 + \left( L(1 + 2\bar{\varepsilon}_t) + \frac{(C_t^x + 4\bar{\varepsilon}_t - 3/4)}{2\eta} \right) d(\tilde{x}_t^*, x_t)^2 \tag{32}$$

where

$$C_t^x \overset{\text{def}}{=} \frac{\bar{\varepsilon}_t}{1 - 4\bar{\varepsilon}_t} \left( 9 + \Delta_t^{-1} \left( 4 + 2\eta \|\nabla_x f(\tilde{x}_t, y_t)\|^2 (E_x^{-1} + 48|\kappa_{\min}|\eta) \right) + 8\eta^2 L^2 \right)$$

$$C_t^y \overset{\text{def}}{=} \frac{\bar{\varepsilon}_t}{1 - 4\bar{\varepsilon}_t} \left( 9 + \Delta_t^{-1} \left( 4 + 2\eta \|\nabla_y f(x_t, \tilde{y}_t)\|^2 (E_y^{-1} + 48|\kappa_{\min}|\eta) \right) + 8\eta^2 L^2 \right).$$

We need to ensure that $\max\{C_t^x, C_t^y\} + 5\bar{\varepsilon}_t \le 1/8$ in order to cancel out the summands in the second and third line of (32). Therefore, we show a bound for $C_t^x$ and $C_t^y$ in order to finish the induction argument. Note that

$$d(x_t, \tilde{x}_t^*) + d(y_t, \tilde{y}_t^*) \overset{①}{\le} d(x_t, x^*) + d(y_t, y^*) + d(x^*, \tilde{x}_t^*) + d(y^*, \tilde{y}_t^*) \overset{②}{\le} \frac{13}{4}(d(x_t, x^*) + d(y_t, y^*)), \tag{33}$$

where ① holds by the triangle inequality and ② holds by Proposition 9. Squaring (33) and noting that $a^2 + b^2 \leq (a+b)^2 \leq 2(a^2 + b^2)$ for $a, b > 0$, we have

$$d(x_t, \tilde{x}_t^*)^2 + d(y_t, \tilde{y}_t^*)^2 \leq 22(d(x_t, x^*)^2 + d(y_t, y^*)^2). \tag{34}$$

Further, we have

$$\|\nabla_x f(\tilde{x}_t, y_t)\| \overset{①}{\leq} \|\nabla_x f(\tilde{x}_t, y_t) - \Gamma_{\tilde{x}_t^*}^{\tilde{x}_t} \nabla_x f(\tilde{x}_t^*, y_t)\| + \|\nabla_x f(\tilde{x}_t^*, y_t)\| \overset{②}{\leq} L d(\tilde{x}_t, \tilde{x}_t^*) + \eta^{-1} d(x_t, \tilde{x}_t^*) \overset{③}{\leq} \frac{17L}{4} d(x_t, \tilde{x}_t^*). \tag{35}$$

Here ① holds by the triangle inequality, ② holds by smoothness of $f$ between $x_t$ and $\tilde{x}_t$, which both lie in $\bar{B}(x^*, 7R/\sqrt{2})$ by the induction assumption, and (30). The optimality condition of $\tilde{x}_t^*$, i.e., $\nabla_x f(\tilde{x}_t^*, y_t) = \frac{1}{\eta}\text{Log}_{\tilde{x}_t^*}(x_t)$ was also used. Lastly, ③ follows by definition of $\eta = 1/(4L)$ and by (29) and $\sqrt{\bar{\varepsilon}_t} \leq 1/4$. Analogously one can show that $\|\nabla_y f(y_t, \tilde{y}_t)\| \leq \frac{17L}{4} d(y_t, \tilde{y}_t^*)$. Using these bounds and the definition of $\eta$, we obtain

$$
\begin{aligned}
C_t^x \overset{①}{\leq} \frac{\bar{\varepsilon}_t}{1 - 4\bar{\varepsilon}_t} \left(10 + \Delta_t^{-1} \left(22 + 109 d(x_t, \tilde{x}_t^*)^2 |\kappa_{\min}|\right)\right) \overset{②}{\leq} \frac{\bar{\varepsilon}_t}{1 - 4\bar{\varepsilon}_t} \left(10 + \Delta_t^{-1} \left(22 + 2385 R^2 |\kappa_{\min}|\right)\right) \\
C_t^x \overset{①}{\leq} \frac{\bar{\varepsilon}_t}{1 - 4\bar{\varepsilon}_t} \left(10 + \Delta_t^{-1} \left(22 + 109 d(y_t, \tilde{y}_t^*)^2 |\kappa_{\min}|\right)\right) \overset{②}{\leq} \frac{\bar{\varepsilon}_t}{1 - 4\bar{\varepsilon}_t} \left(10 + \Delta_t^{-1} \left(22 + 2385 R^2 |\kappa_{\min}|\right)\right)
\end{aligned}
\tag{36}
$$

where for both inequalities, ① follows by (35) and ② follows by (34) and the induction hypothesis. Our bounds on $C_t^x$ and $C_t^y$ and our choice of $\varepsilon_t$, $E_x$, $E_y$ and $\eta$ as well as $\Delta_t$, which we specify below for $\mu = 0$ and $\mu > 0$ separately, ensure that $\max\{C_t^x, C_t^y\} + 5\bar{\varepsilon}_t \leq 1/8$ and the update rules in Lines 2 and 3 can be implemented efficiently. Hence we have,

$$f(\tilde{x}_t, y^*) - f(x^*, \tilde{y}_t) \leq \frac{1}{2\eta} \left(d(y_t, y^*)^2 + d(x_t, x^*)^2 + (\Delta_t - 1 - \mu\eta/2)(d(y_{t+1}, y^*)^2 + d(x_{t+1}, x^*)^2)\right). \tag{37}$$

We now analyze the g-convex and the strongly g-convex case separately.

**Case $\mu = 0$.** We set $\Delta_t = (t+1)^{-2}$. By definition, the LHS of (37) is non-negative, hence by rearranging we obtain

$$
d(x_{t+1}, x^*)^2 + d(y_{t+1}, y^*)^2 \leq \frac{d(x_t, x^*)^2 + d(y_t, y^*)^2}{1 - \Delta_t} \overset{①}{\leq} (d(x_1, x^*)^2 + d(y_1, y^*)^2) \prod_{i=1}^{t} \frac{1}{1 - \Delta_i}
$$

$$
\overset{②}{\leq} 2(d(x_1, x^*)^2 + d(y_1, y^*)^2)
\tag{38}
$$

where ① follows by repeatedly applying this inequality and ② holds by Proposition 10. Noting that $a^2 + b^2 \leq (a+b)^2 \leq 2(a^2 + b^2)$ for $a, b > 0$, this proves the induction statement, as $d(x_{t+1}, x^*) + d(y_{t+1}, y^*) \leq 2R$. Summing (37) from $t = 1$ to $T$, dividing by $T$ and telescoping the sum, it follows that

$$
\frac{1}{T} \sum_{t=1}^{T} (f(\tilde{x}_t, y^*) - f(x^*, \tilde{y}_t)) \leq \frac{1}{2\eta T}(d(x_1, x^*)^2 + d(y_1, y^*)^2) + \sum_{t=1}^{T} \frac{\Delta_t(d(x_t, x^*)^2 + d(y_t, y^*)^2)}{2\eta T}
$$

$$
\overset{①}{\leq} \frac{d(x_1, x^*)^2 + d(y_1, y^*)^2}{2\eta T} \left(2 + \sum_{t=1}^{T} \Delta_t\right) \overset{②}{\leq} \frac{d(x_1, x^*)^2 + d(y_1, y^*)^2}{\eta T} \overset{③}{=} \frac{6LR^2}{T},
$$

where ① holds by (38), ② holds by Proposition 14 and ③ by definition of $\eta$ and $R$. By applying (GEO-AVG) to the sequence $(\tilde{x}_t, \tilde{y}_t)$, we obtain

$$f(\bar{x}_T, y^*) - f(x^*, \bar{y}_T) \leq \frac{1}{T} \sum_{t=1}^{T} (f(\tilde{x}_t, y^*) - f(x^*, \tilde{y}_t)).$$

Hence after $T = \lceil \frac{6LR^2}{\varepsilon} \rceil$ iterations of Algorithm 2, we have that $f(\bar{x}_T, y^*) - f(x^*, \bar{y}_T) \leq \varepsilon$. This concludes the proof for the g-convex case.

**Case $\mu > 0$.** We set $\Delta_t = \frac{\mu\eta}{4}$. By definition, the LHS of (37) is non-negative, hence by rearranging and using the definition of $\Delta_t$, we obtain

$$d(y_{t+1}, y^*)^2 + d(x_{t+1}, x^*)^2 \leq \frac{1}{1 + \mu\eta/4}(d(y_t, y^*)^2 + d(x_t, x^*)^2)$$

$$\overset{\textcircled{1}}{\leq} \exp\left(\frac{-t\mu}{17L}\right)(d(y_t, y^*)^2 + d(x_t, x^*)^2),$$

where $\textcircled{1}$ holds by (28). Noting that $a^2 + b^2 \leq (a+b)^2 \leq 2(a^2 + b^2)$ for $a, b > 0$. This proves the induction statement, as $d(x_{t+1}, x^*) + d(y_{t+1}, y^*) \leq 2R$. Then by repeatedly applying the inequality, we have

$$d(y_{t+1}, y^*)^2 + d(x_{t+1}, x^*)^2 \leq \exp\left(\frac{-t\mu}{17L}\right)R^2.$$

Hence, running Algorithm 2 for $T = \frac{17L}{\mu}\log\left(\frac{2LR^2}{\varepsilon}\right)$ iterations implies that

$$d(y_T, y^*)^2 + d(x_T, x^*)^2 \leq \frac{\varepsilon}{2L}.$$

Dropping negative terms from (37) for $t \leftarrow T$ and using the definition of $\eta$, we obtain

$$f(\tilde{x}_T, y^*) - f(x^*, \tilde{y}_T) \leq 2L(d(y_T, y^*)^2 + d(x_T, x^*)^2) \leq \varepsilon,$$

which concludes the proof. $\qquad\square$

## C. Implementing RIOD and RIODA

In this section, we use $x_t^\tau$ and $\tilde{x}_t^\tau$ to refer to the $\tau$-th iterates of subroutines minimizing $L_t$ and $\tilde{L}_t$ starting from $x_t$, i.e., $x_t = x_t^0 = \tilde{x}_t^0$. Further, if $\tau$ is the last iterate, we write $\tilde{x}_t^\tau = \tilde{x}_t$ and $x_t^\tau = x_{t+1}$.

**Corollary 2** (Implementing RIOD). [↓] *For the implementation of the update rules in Lines 4 and 6 of Algorithm 1, we require $\widetilde{O}((L\eta + \zeta_D)\zeta_{\tilde{R}})$ gradient oracle calls to $\ell_t$ and $\tilde{\ell}_t$ at iteration $t$ using PRGD or $\widetilde{O}(1 + L\eta)$ using CRGD (this includes a logarithmic dependence on $|\kappa_{\min}|$). Here $\tilde{R} \overset{\text{def}}{=} G/L + D$, where $G$ is the Lipschitz constants of $\ell_t$ and $\tilde{\ell}_t$ in $\mathcal{X}$. Note that these implementations do not require the knowledge of $G$.*

*Proof.* (Corollary 2) Note that we provide the analysis for the criterion of $\tilde{x}_t$, but the analysis for $x_{t+1}$ follows by the same arguments. Since $\tilde{\ell}_t$ is differentiable, there exists a constant $G \geq 0$ such that $\|\nabla\tilde{\ell}_t(x)\| \leq G$ for all $x$ in the compact set $\mathcal{X}$, which implies $G$-Lipschitzness. Note that $\tilde{L}_t$ is $(L + \zeta_D/\eta)$-smooth and $(1/\eta)$-strongly g-convex in $\mathcal{X}$, since the regularizer $\frac{1}{2\eta}d(x, x_t)^2$ is $(\zeta_D/\eta)$-smooth and $1/(\eta)$-strongly g-convex in $\mathcal{X}$ by Fact 15. Further we have for $x \in \mathcal{X}$ that

$$\|\nabla\tilde{L}_t(x)\| = \|\nabla\tilde{\ell}_t(x) - \eta^{-1}\text{Log}_x(x_t)\| \leq \|\nabla\tilde{\ell}_t(x)\| + \eta^{-1}d(x, x_t) \overset{\textcircled{1}}{\leq} G + \frac{D}{\eta},$$

where $\textcircled{1}$ follows since $\tilde{\ell}_t$ is $G$-Lipschitz and $x, x_t \in \mathcal{X}$, which implies that $\tilde{L}_t$ is $(G + D/\eta)$-Lipschitz. Recall that we require the following for the subproblem

$$\tilde{L}_t(\tilde{x}_t) - \tilde{L}_t(\tilde{x}_t^*) \leq \varepsilon_t d(x_t, \tilde{x}_t^*)^2. \tag{39}$$

In the following, we compute the gradient oracle complexity of ensuring (39) using the upper bound $G$ in order to show the worst-case complexity. We then show that the criterion can also be implemented in a way that adapts to the local gradient norm and does not require knowledge of $G$.

**PRGD.** By Fact 21, running $\tau$ iterations of PRGD on $\tilde{L}_t$, starting at $x_t$, we have that

$$\tilde{L}_t(\tilde{x}_t) - \tilde{L}_t(\tilde{x}_t^*) \leq \frac{(L + \zeta_D/\eta)\zeta_{\tilde{R}}}{2}\exp\left(\frac{-(\tau - 1)}{4(L\eta + \zeta_D)\zeta_{\tilde{R}}}\right)d(x_t, \tilde{x}_t^*)^2,$$

where $\tilde{R} = \frac{G+D/\eta}{L+\zeta_D/\eta}$. Hence, it is sufficient to run PRGD for

$$\tau = O\left((L\eta + \zeta_D)\zeta_{\tilde{R}} \log\left(\frac{(L+\zeta_D/\eta)\zeta_{\tilde{R}}}{\varepsilon_t}\right)\right)$$

iterations in order to satisfy (39). We now show that this criterion can be implemented without knowledge of $G$. We have that

$$\tilde{L}_t(\tilde{x}_t^\tau) - \tilde{L}_t(\tilde{x}_t^*) \overset{①}{\leq} (\tilde{L}_t(\tilde{x}_t^{\tau-1}) - \tilde{L}_t(\tilde{x}_t^*))\left(1 - \frac{1}{4(L\eta + \zeta_D)\zeta_{\tilde{R}_{\tau-1}}}\right) \overset{②}{\leq} (\tilde{L}_t(\tilde{x}_t^1) - \tilde{L}_t(\tilde{x}_t^*))\prod_{i=1}^{\tau-1}\left(1 - \frac{1}{4(L\eta + \zeta_D)\zeta_{\tilde{R}_i}}\right)$$

$$\overset{③}{\leq} \frac{(L+\zeta_D/\eta)\zeta_{\tilde{R}_0}}{2} d(x_t, \tilde{x}_t^*)^2 \prod_{i=1}^{\tau-1}\left(1 - \frac{1}{4(L\eta + \zeta_D)\zeta_{\tilde{R}_i}}\right),$$

where $\tilde{R}_\tau \overset{\text{def}}{=} \|\nabla\tilde{L}_t(\tilde{x}_t^\tau)\|/(L + \zeta_D/\eta) = \|\nabla\tilde{\ell}_t(\tilde{x}_t^\tau) - \eta^{-1}\text{Log}_{\tilde{x}_t^\tau}(x_t)\|/(L + \zeta_D/\eta)$. Here ① holds by Fact 21, ② by repeatedly applying the prior inequality and ③ by (Martínez-Rubio & Pokutta, 2023, Lemma 18). It follows that we need to run PRGD for $\tau \geq 2$ iterations until

$$\frac{(L+\zeta_D/\eta)\zeta_{\tilde{R}_0}}{2}\prod_{i=1}^{\tau-1}\left(1 - \frac{1}{4(L\eta + \zeta_D)\zeta_{\tilde{R}_i}}\right) \leq \varepsilon_t = \frac{\max\{4, (t+1)^2(15 + 8\eta^2 L^2 + 2\eta^2\|\nabla\tilde{\ell}_t(\tilde{x}_t^\tau)\|^2(D^{-2} + 48|\kappa_{\min}|))\}^{-1}}{8\eta}.$$

Note that $\nabla\tilde{L}_t(\tilde{x}_t^\tau) = \nabla\tilde{\ell}_t(\tilde{x}_t^\tau) - \eta^{-1}\text{Log}_{\tilde{x}_t^\tau}(x_t)$ has to be computed anyways for each iterations of PRGD, hence this criterion can be checked with little computational overhead.

**CRGD.** After $\tau$ iterations of CRGD on $\tilde{L}_t$ starting from $x_t$, we have

$$\tilde{L}_t(\tilde{x}_t^\tau) - \tilde{L}_t(\tilde{x}_t^*) \leq \frac{L}{2}\exp\left(-(\tau-1)\min\left\{\frac{1}{4L\eta}, \frac{1}{2}\right\}\right) d(x_t, \tilde{x}_t^*)^2. \tag{40}$$

Here we used Proposition 22 and Corollary 23 with $f \leftarrow \tilde{\ell}_t$ and $g \leftarrow \frac{1}{2\eta}d(\cdot, x_t)^2$, noting that $\tilde{L}_t$ is $(1/\eta)$-strongly g-convex and $\tilde{\ell}_t$ is $L$-smooth. Hence it is sufficient to run CRGD for $\tau = O\left((1 + L\eta)\log\left(\frac{L}{\varepsilon_t}\right)\right)$ iterations in order to satisfy (39).

We now show that this criterion can be implemented without knowledge of $G$. By (40), it follows that we need to run CRGD for $\tau \geq 1$ iterations until

$$\frac{L}{2}\exp\left(-(\tau-1)\min\left\{\frac{1}{4L\eta}, \frac{1}{2}\right\}\right) \leq \varepsilon_t = \frac{\max\{4, (t+1)^2(15 + 8\eta^2 L^2 + 2\eta^2\|\nabla\tilde{\ell}_t(\tilde{x}_t^\tau)\|^2(D^{-2} + 48|\kappa_{\min}|))\}^{-1}}{8\eta}$$

Note that $\nabla\tilde{L}_t(\tilde{x}_t^\tau) = \nabla\tilde{\ell}_t(\tilde{x}_t^\tau) - \frac{1}{\eta}\text{Log}_{\tilde{x}_t^\tau}(x_t)$ has to be computed anyways for each iterations of CRGD, hence this criterion can be checked with little computational overhead.

$\square$

**Corollary 4** (Implementing RIODA). [↓] *We use the notation from Algorithm 2. For the implementation of the update rules in Lines 2 and 3 of Algorithm 2, we require $\widetilde{O}(\zeta_D\zeta_{\tilde{R}})$ gradient oracle calls per iteration using PRGD or $\widetilde{O}(1)$ using CRGD (this includes a logarithmic dependence on $|\kappa_{\min}|$). Here $\tilde{R} \overset{\text{def}}{=} G/L + D$, where $G$ is the Lipschitz constant of $f$ in $\mathcal{X} \times \mathcal{Y}$. We refer to these algorithms as* RIODA$_{PRGD}$ *and* RIODA$_{CRGD}$, *respectively. Note that these implementations do not require the knowledge of $G$.*

*Proof.* (Corollary 4) We discuss the implementation of the criteria for $x$, but the proof for $y$ follows analogously. The proof follows by applying Corollary 2 to the update rules of $x$ and $y$, taking into account the definitions of $\varepsilon_t$ and $\eta$ in Algorithm 2 and the following properties of $f$. We have that $\tilde{L}_t(x) = f(x, y_t) + \frac{1}{2\eta}d(x, x_t)^2$ is $(L + \zeta_D/\eta)$-smooth and $(1/\eta)$-strongly g-convex, because $f(x, y_t)$ is $L$-smooth and the regularizer $\frac{1}{2\eta}d(x, x_t)^2$ is $(\zeta_D/\eta)$-smooth and $(1/\eta)$-strongly g-convex in

$\mathcal{X}$ by Fact 15. Further, since $f(\cdot, y_t)$ is differentiable, there exists a constant $G$, such that $\|\nabla_x f(x, y_t)\| \leq G$ for all $x \in \mathcal{X}$. Since $f(\cdot, y_t)$ is also g-convex, this implies $G$-Lipschitzness. It follows that

$$\|\nabla \tilde{L}_t(x)\| = \|\nabla_x f(x, y_t) - \eta^{-1} \text{Log}_x(x_t)\| \overset{\textcircled{1}}{\leq} \|\nabla_x f(x, y_t)\| + \eta^{-1} d(x, x_t) \leq G + 4LD,$$

where $\textcircled{1}$ follows since $f(\cdot, x_t)$ is $G$-Lipschitz and $x, x_t \in \mathcal{X}$. This implies that $\tilde{L}_t$ is $(G + 4LD)$-Lipschitz. $\qquad\square$

**Corollary 6** (Implementing RIODA – unconstrained). [↓] *Consider the setting of Theorem 5. Assume in addition that $f$ is g-convex, g-concave and $L$-smooth in $\bar{B}(x^*, 8R) \times \bar{B}(y^*, 8R)$. Then we require $\tilde{O}(1)$ gradient oracle calls for implementing the update rules in Lines 2 and 3 of Algorithm 2 using CRGD (this includes a logarithmic dependence on $|\kappa_{\min}|$) and the iterates stay in that set. If $f$ is g-convex, g-concave and $L$-smooth in $\bar{B}(x^*, \bar{D}) \times \bar{B}(y^*, \bar{D})$ with $\bar{D} \overset{\text{def}}{=} R(13\zeta_{8R} + 9)$, then we require $\tilde{O}(\zeta_R^2)$ gradient oracle calls using RGD and the iterates stay in that set. We refer to these algorithms as* RIODA$_{CRGD}$ *and* RIODA$_{RGD}$, *respectively. Neither method requires prior knowledge of the initial distance to the saddle point $R$.*

*Proof.* (Corollary 6) We discuss the implementation of the criteria for $x$, but the proof for $y$ follows analogously. Recall that we have

$$C_t^x \overset{\textcircled{1}}{\leq} \frac{\bar{\varepsilon}_t}{1 - 4\bar{\varepsilon}_t} \left(10 + \Delta_t^{-1}\left(22 + 109 d(x_t, \tilde{x}_t^*)^2 |\kappa_{\min}|\right)\right) \overset{\textcircled{2}}{\leq} \frac{\bar{\varepsilon}_t}{1 - 4\bar{\varepsilon}_t} \left(10 + \Delta_t^{-1}\left(22 + 2385 R^2 |\kappa_{\min}|\right)\right)$$

$$C_t^x \overset{\textcircled{1}}{\leq} \frac{\bar{\varepsilon}_t}{1 - 4\bar{\varepsilon}_t} \left(10 + \Delta_t^{-1}\left(22 + 109 d(y_t, \tilde{y}_t^*)^2 |\kappa_{\min}|\right)\right) \overset{\textcircled{2}}{\leq} \frac{\bar{\varepsilon}_t}{1 - 4\bar{\varepsilon}_t} \left(10 + \Delta_t^{-1}\left(22 + 2385 R^2 |\kappa_{\min}|\right)\right). \tag{41}$$

We will make use of $\textcircled{2}$ to compute the worst-case complexity of implementing the criterion using $R$ and $\textcircled{1}$ to analyze the implementation that adapts to the value of $d(x_t, \tilde{x}_t^*)^2$ or $d(y_t, \tilde{y}_t^*)^2$.

**CRGD.** Recall that $L_t(x) = f(x, y_t) + \frac{1}{2\eta} d(x, x_t)^2$ and $\tilde{L}_t(x) = f(x, y_t) + \frac{1}{2\eta} d(x, \tilde{x}_t)^2$ with $\eta = 1/(4L)$. Note that $L_t$ and $\tilde{L}_t$ are $(1/\eta)$-strongly g-convex and $f$ is $L$-smooth in $x \in \bar{B}(x^*, 8R)$. Then we have

$$d(\tilde{x}_t^{\tau+1}, \tilde{x}_t^*) \overset{\textcircled{1}}{\leq} 2^{-\tau} d(x_t, \tilde{x}_t^*) \overset{\textcircled{2}}{\leq} 2^{-\tau}(d(x_t, x^*) + d(\tilde{x}_t^*, x^*)) \overset{\textcircled{3}}{\leq} \frac{13}{4} R, \tag{42}$$

and

$$d(x_t^{\tau+1}, x_{t+1}^*) \overset{\textcircled{1}}{\leq} 2^{-\tau} d(x_t, x_{t+1}^*) \overset{\textcircled{2}}{\leq} 2^{-\tau}(d(x_t, x^*) + d(x_{t+1}^*, x^*)) \overset{\textcircled{3}}{\leq} \frac{57}{16} R \tag{43}$$

where for both inequalities, $\textcircled{1}$ holds by repeatedly applying Corollary 24, $\textcircled{2}$ holds by the triangle inequality and $\textcircled{3}$ holds by Proposition 9, since $d(x_t, x^*) + d(y_t, y^*) \leq 2R$ and, $\tau \geq 0$. Hence we have for any $\tau \geq 0$

$$d(\tilde{x}_t^\tau, x^*) \overset{\textcircled{1}}{\leq} d(\tilde{x}_t^\tau, \tilde{x}_t^*) + d(\tilde{x}_t^*, x^*) \overset{\textcircled{2}}{\leq} \left(\frac{13}{4} + \frac{9}{2}\right) R$$

and

$$d(x_t^\tau, x^*) \overset{\textcircled{1}}{\leq} d(x_t^\tau, x_{t+1}^*) + d(x_{t+1}^*, x^*) \overset{\textcircled{2}}{\leq} \left(\frac{57}{16} + \frac{41}{8}\right) R,$$

where for both inequalities, $\textcircled{1}$ holds by the triangle inequality and $\textcircled{2}$ holds by (42) and (43) and Proposition 9. As described in Corollary 2, the complexity of implementing the criterion using CRGD is $O\left(\log(L/\varepsilon_t)\right)$.

We now show that this criterion can be implemented without knowledge of $R$. We have by (40) that

$$\tilde{L}_t(\tilde{x}_t^\tau) - \tilde{L}_t(\tilde{x}_t^*) \leq 2^{-\tau} L d(x_t, \tilde{x}_t^*)^2.$$

Note that

$$d(x_t, \tilde{x}_t^*)^2 \overset{\textcircled{1}}{\leq} 2d(x_t, \tilde{x}_t)^2 + 2d(\tilde{x}_t, \tilde{x}_t^*)^2 \overset{\textcircled{2}}{\leq} 2d(x_t, \tilde{x}_t)^2 + \frac{1}{8}d(x_t, \tilde{x}_t^*)^2$$

$$\Leftrightarrow d(x_t, \tilde{x}_t^*)^2 \overset{\textcircled{3}}{\leq} 3d(x_t, \tilde{x}_t)^2,$$

(44)

where $\textcircled{1}$ holds by the triangle inequality, $\textcircled{2}$ holds by (29) and $\bar{\varepsilon}_t$, and $\textcircled{3}$ by rearranging. Applying this bound to the RHS of $\textcircled{1}$ in (41), it follows that choosing $\varepsilon_t = L \min\left\{1/8, \left(\Delta_t^{-1}(32 + 327d(x_t, \tilde{x}_t^\tau)^2|\kappa_{\min}|)\right)^{-1}\right\}$ is sufficient to ensure $C_t^x + 5\bar{\varepsilon}_t \leq 1/8$. It follows that it is enough to run CRGD for $\tau \geq 1$ iterations until

$$2^{-\tau} \leq \varepsilon_t = L \min\left\{1/8, \left((t+1)^2(32 + 327d(x_t, \tilde{x}_t^\tau)^2|\kappa_{\min}|)\right)^{-1}\right\}$$

for $\mu = 0$ and

$$2^{-\tau} \leq \varepsilon_t = L \min\left\{1/8, 4L\left(\mu(25 + 220d(x_t, \tilde{x}_t^\tau)^2|\kappa_{\min}|)\right)^{-1}\right\}$$

for $\mu > 0$. Note that $\nabla \tilde{L}_t(\tilde{x}_t^\tau) = \nabla \tilde{\ell}_t(\tilde{x}_t^\tau) - \frac{1}{\eta}\text{Log}_{\tilde{x}_t^\tau}(x_t)$ has to be computed anyways for each iterations of CRGD and $d(x_t, \tilde{x}_t^\tau) = \|\text{Log}_{\tilde{x}_t^\tau}(x_t)\|$, hence this criterion can be checked with little computational overhead.

**RGD.** Applying Fact 20 to $\tilde{L}_t$ and $L_t$, we have for the iterates of RGD that

$$d(\tilde{x}_t^\tau, \tilde{x}_t^*) \leq \frac{1+\sqrt{5}}{2}\zeta_{d(x_t, \tilde{x}_t^*)}d(x_t, \tilde{x}_t^*) \text{ and } d(x_t^\tau, x_{t+1}^*) \leq \frac{1+\sqrt{5}}{2}\zeta_{d(x_t, x_{t+1}^*)}d(x_t, x_{t+1}^*)$$

(45)

for all $\tau \geq 0$. Further, we have after $\tau$ iterations that

$$\tilde{L}_t(\tilde{x}_t) - \tilde{L}_t(\tilde{x}_t^*) \leq \exp\left(\frac{-\tau}{\zeta_{\bar{D}}}\right)\frac{(L + \zeta_{\bar{D}}/\eta)d(x_t, \tilde{x}_t^*)^2}{2}$$

and

$$L_t(x_{t+1}) - L_t(\tilde{x}_t^*) \leq \exp\left(\frac{-\tau}{\zeta_{\bar{D}}}\right)\frac{(L + \zeta_{\bar{D}}/\eta)d(x_t, x_{t+1}^*)^2}{2}.$$

where

$$\bar{D} = \frac{1+\sqrt{5}}{2}\max\{\zeta_{d(x_t, \tilde{x}_t^*)}d(x_t, \tilde{x}_t^*), \zeta_{d(x_t, x_{t+1}^*)}d(x_t, x_{t+1}^*)\}.$$

We go on to bound $\bar{D}$. Note that

$$d(x_t, \tilde{x}_t^*) \overset{\textcircled{1}}{\leq} d(x_t, x^*) + d(x^*, \tilde{x}_t^*) \overset{\textcircled{2}}{\leq} \frac{13}{2}R \leq 8R,$$

(46)

and

$$d(x_t, x_{t+1}^*) \overset{\textcircled{1}}{\leq} d(x_t, x^*) + d(x^*, x_{t+1}^*) \overset{\textcircled{2}}{\leq} 2\left(1 + \frac{41}{16}\right)R \leq 8R$$

(47)

where for both inequalities, $\textcircled{1}$ holds by the triangle inequality and $\textcircled{2}$ holds by Proposition 9. Further,

$$d(\tilde{x}_t^\tau, x^*) \overset{\textcircled{1}}{\leq} d(\tilde{x}_t^\tau, \tilde{x}_t^*) + d(x^*, \tilde{x}_t^*) \overset{\textcircled{2}}{\leq} R\frac{1+\sqrt{5}}{2}\left(8\zeta_{8R} + \frac{9}{2}\right),$$

and

$$d(x_t^\tau, x^*) \overset{\textcircled{1}}{\leq} d(x_t^\tau, x_{t+1}^*) + d(x^*, x_{t+1}^*) \overset{\textcircled{2}}{\leq} R\frac{1+\sqrt{5}}{2}\left(8\zeta_{8R} + \frac{41}{8}\right),$$

where for both inequalities, $\textcircled{1}$ holds by the triangle inequality and $\textcircled{2}$ holds by applying (46) to (45). It follows that $\bar{D} = R(13\zeta_{8R} + 9)$ and hence $\zeta_{\bar{D}} = O(\zeta_R^2)$. The complexity of implementing the criterion is $\tau = O\left(\zeta_R^2 \log(L\zeta_R^2/\varepsilon_t)\right)$.

We show that it is possible to implement without the knowledge of $R$. We can address the first obstacle as in the CRGD implementation by noting that choosing $\varepsilon_t = L \min\left\{1/8, \left(\Delta_t^{-1}(32 + 327 d(x_t, \tilde{x}_t^\tau)^2 |\kappa_{\min}|)\right)^{-1}\right\}$ is sufficient to ensure $C_t^x + 5\bar{\varepsilon}_t \leq 1/8$. But the number of iterations still depends on $R$ via $\zeta_R$. To circumvent this, assume $\|\nabla \tilde{L}_t(\tilde{x}_t)\|^2 \leq \tilde{\varepsilon}_t \stackrel{\text{def}}{=} \frac{\varepsilon_t d(x_t, \tilde{x}_t)^2}{\eta + 2\eta^2 \varepsilon_t}$. We show this suffices to satisfy the criterion. We have

$$
\begin{aligned}
\tilde{L}_t(\tilde{x}_t^*) &\stackrel{①}{\geq} \tilde{L}_t(\tilde{x}_t) + \langle \nabla \tilde{L}_t(\tilde{x}_t), \mathrm{Log}_{\tilde{x}_t}(\tilde{x}_t^*)\rangle + \frac{1}{2\eta} d(\tilde{x}_t, \tilde{x}_t^*)^2 \\
&\geq \tilde{L}_t(\tilde{x}_t) + \min_{z \in \mathcal{M}}\left[\langle \nabla \tilde{L}_t(\tilde{x}_t), \mathrm{Log}_{\tilde{x}_t}(z)\rangle + \frac{1}{2\eta} d(\tilde{x}_t, z)^2\right] \\
&\stackrel{②}{\geq} \tilde{L}_t(\tilde{x}_t) - \frac{\eta}{2}\|\nabla \tilde{L}_t(\tilde{x}_t)\|^2
\end{aligned}
\tag{48}
$$

where ① holds by the $(1/\eta)$-strong g-convexity of $\tilde{L}_t$, ② by noting that

$$
\operatorname*{arg\,min}_{z \in \mathcal{M}}\left\{\langle \nabla \tilde{L}_t(\tilde{x}_t), \mathrm{Log}_{\tilde{x}_t}(z)\rangle + \frac{1}{2\eta} d(\tilde{x}_t, z)^2\right\} = \mathrm{Exp}_{\tilde{x}_t}\left(\operatorname*{arg\,min}_{v \in T_{\tilde{x}_t}\mathcal{M}} \| -\eta \nabla \tilde{L}_t(\tilde{x}_t) - v\|_{\tilde{x}_t}^2\right) = \mathrm{Exp}_{\tilde{x}_t}(-\eta \nabla \tilde{L}_t(\tilde{x}_t)).
$$

We have

$$
\frac{1}{2\eta} d(\tilde{x}_t, \tilde{x}_t^*)^2 \stackrel{①}{\leq} \tilde{L}_t(\tilde{x}_t) - \tilde{L}_t(\tilde{x}_t^*) \stackrel{②}{\leq} \frac{\eta}{2}\|\nabla \tilde{L}_t(\tilde{x}_t)\|^2 \stackrel{③}{\leq} \eta \tilde{\varepsilon}_t/2,
\tag{49}
$$

where ① holds by the $(1/\eta)$-strong g-convexity of $\tilde{L}_t$, ② holds by (48) and ③ holds by assumption. Hence, in order to satisfy the criterion, i.e., $\tilde{L}_t(\tilde{x}_t) - \tilde{L}_t(\tilde{x}_t^*) \leq \varepsilon_t d(x_t, \tilde{x}_t^*)^2$, we require

$$
\eta \tilde{\varepsilon}_t/2 = \frac{\varepsilon_t d(x_t, \tilde{x}_t)^2}{2 + 4\eta \varepsilon_t} \leq \varepsilon_t d(x_t, \tilde{x}_t^*)^2.
$$

Note that

$$
\begin{aligned}
d(x_t, \tilde{x}_t)^2 &\stackrel{①}{\leq} 2d(x_t, \tilde{x}_t^*)^2 + 2d(\tilde{x}_t, \tilde{x}_t^*)^2 \stackrel{②}{\leq} 2d(x_t, \tilde{x}_t^*)^2 + \frac{2\eta \varepsilon_t d(x_t, \tilde{x}_t)^2}{1 + 2\eta \varepsilon_t} \\
&\stackrel{③}{\Leftrightarrow} d(x_t, \tilde{x}_t)^2 \stackrel{③}{\leq} 2(1 + \eta \varepsilon_t) d(x_t, \tilde{x}_t^*)^2,
\end{aligned}
\tag{50}
$$

where ① holds by the triangle inequality, ② by (49) and by definition of $\tilde{\varepsilon}_t$ and ③ holds by rearranging. Hence we conclude that

$$
\eta \tilde{\varepsilon}_t/2 = \frac{\varepsilon_t d(x_t, \tilde{x}_t)^2}{2 + 4\eta \varepsilon_t} \stackrel{①}{\leq} \frac{2(1 + \eta \varepsilon_t)\varepsilon_t d(x_t, \tilde{x}_t^*)^2}{2 + 4\eta \varepsilon_t} \leq \varepsilon_t d(x_t, \tilde{x}_t^*)^2,
$$

where ① holds by (50). $\qquad\square$

## D. Technical Results

**Proposition 9.** *Let $\mathcal{M}$, $\mathcal{N}$ be Hadamard manifolds with sectional curvature in $[\kappa_{\min}, 0]$. Consider the bi-function $f : \mathcal{M} \times \mathcal{N} \to \mathbb{R}$, which is CC and $L$-smooth in $\mathcal{X} \times \mathcal{Y}$, where $\mathcal{X} \stackrel{\text{def}}{=} \bar{B}(x^*, \bar{D})$, $\mathcal{Y} \stackrel{\text{def}}{=} \bar{B}(y^*, \bar{D})$ where $(x^*, y^*)$ is a saddle point of $f$ and $\bar{D} \stackrel{\text{def}}{=} 3(d(x_t, x^*) + d(y_t, y^*))$. Further, let $\tilde{x}_t^*$, $\tilde{y}_t^*$, $x_{t+1}^*$ and $y_{t+1}^*$ be defined as in Algorithm 2 for the unconstrained case and $\eta \leq 1/4L$. Then, we have that*

$$
d(\tilde{x}_t^*, x^*) + d(\tilde{y}_t^*, y^*) \leq \frac{9}{4}(d(y_t, y^*) + d(x_t, x^*)).
$$

*and*

$$
d(x_{t+1}^*, x^*) + d(y_{t+1}^*, y^*) \leq \frac{41}{16}(d(y_t, y^*) + d(x_t, x^*)).
$$

*Proof.* Let $H_t(x,y) \stackrel{\text{def}}{=} f(x,y) + \frac{1}{2\eta}d(x,x_t)^2 - \frac{1}{2\eta}d(y,y_t)^2$, $y^+(x) \stackrel{\text{def}}{=} \arg\min_{y\in\mathcal{Y}} H_t(x,y)$, $x^+(y) \stackrel{\text{def}}{=}$ $\arg\min_{x\in\mathcal{X}} H_t(x,y)$, $x^+(y^*) = \min_{x\in\mathcal{X}} f(x,y^*) + \frac{1}{2\eta}d(x,x_t)^2$ and $y^+(x^*) = \max_{y\in\mathcal{Y}} f(x^*,y) - \frac{1}{2\eta}d(y,y_t)^2$. Further, we have $x^* = \arg\min_{x\in\mathcal{X}} f(x,y^*)$ and $y^* = \arg\max_{y\in\mathcal{Y}} f(x^*,y)$. Note that $(x^*,y^*)$, $(x_t,y_t)$, $(x^+(y^*),y^+(x^*))$ and $(x^+(y_t),y^+(x_t))$ lie in $\mathcal{X}\times\mathcal{Y}$ by definition. Using (Martínez-Rubio & Pokutta, 2023, Lemma 10), we obtain that

$$d(x_t, x^+(y^*)) \leq d(x_t, x^*), \quad d(y_t, y^+(x^*)) \leq d(y_t, y^*). \tag{51}$$

Note that $H_t$ is $(1/\eta)$-SCSC in $\mathcal{X}\times\mathcal{Y}$ and $\nabla_x H_t(x,\cdot)$ and $\nabla_y H_t(\cdot,y)$ are $L$-Lipschitz for all $x\in\mathcal{X}$ and $y\in\mathcal{Y}$, respectively. Hence, applying (Martínez-Rubio et al., 2023, Lemma 40) to $H_t$, we have that $x^+(y)$ and $y^+(x)$ are $(L\eta)$-Lipschitz for all $x\in\mathcal{X}$ and $y\in\mathcal{Y}$, respectively and in particular, we have

$$d(x^+(y^*), x^+(y_t)) \leq L\eta d(y_t, y^*), \quad d(y^+(x^*), y^+(x_t)) \leq L\eta d(x_t, x^*) \tag{52}$$

$$d(x^+(y^*), x^+(\tilde{y}_t)) \leq L\eta d(\tilde{y}_t, y^*), \quad d(y^+(x^*), y^+(\tilde{x}_t)) \leq L\eta d(\tilde{x}_t, x^*). \tag{53}$$

It follows that,

$$d(x^*, x^+(y_t)) + d(y^*, y^+(x_t)) \stackrel{①}{\leq} d(x^*, x^+(y^*)) + d(y^*, y^+(x^*)) + d(x^+(y^*), x^+(y_t)) + d(y^+(x^*), y^+(x_t))$$

$$\stackrel{②}{\leq} d(x_t, x^+(y^*)) + d(y_t, y^+(x^*)) + (1+L\eta)(d(y_t, y^*) + d(x_t, x^*)) \tag{54}$$

$$\stackrel{③}{\leq} (2+L\eta)(d(y_t, y^*) + d(x_t, x^*)) \stackrel{④}{\leq} \frac{9}{4}(d(y_t, y^*) + d(x_t, x^*))$$

where ① follows by the triangle inequality, ② follows by the triangle inequality and (52), ③ follows by (51) and ④ by definition of $\eta$. Note that by (54), we have that $x^+(y_t)$ and $y^+(x_t)$ lie in the interior of $\mathcal{X}$ and $\mathcal{Y}$ respectively and hence the constraints $\bar{B}(x^*,\bar{D})$ and $\bar{B}(y^*,\bar{D})$ are inactive. Recall that $\tilde{x}_t^* = \arg\min_{x\in\mathcal{M}} f(x,y_t) + \frac{1}{2\eta}d(x,x_t)^2$ and $\tilde{y}_t^* = \arg\max_{y\in\mathcal{N}} f(x_t,y) - \frac{1}{2\eta}d(y,y_t)^2$, hence we have that $\tilde{x}_t^* = x^+(y_t)$, $\tilde{y}_t^* = y^+(x_t)$. Further, we have that

$$d(x^*, x^+(\tilde{y}_t)) + d(y^*, y^+(\tilde{x}_t)) \stackrel{①}{\leq} d(x^*, x^+(y^*)) + d(y^*, y^+(x^*)) + d(x^+(y^*), x^+(\tilde{y}_t)) + d(y^+(x^*), y^+(\tilde{x}_t))$$

$$\stackrel{②}{\leq} d(x_t, x^+(y^*)) + d(y_t, y^+(x^*)) + d(y_t, y^*) + d(x_t, x^*) + L\eta(d(\tilde{y}_t, y^*) + d(\tilde{x}_t, x^*))$$

$$\stackrel{③}{\leq} (2+\frac{9L\eta}{4})(d(y_t, y^*) + d(x_t, x^*)) \stackrel{④}{\leq} \frac{41}{16}(d(y_t, y^*) + d(x_t, x^*))$$

$$\tag{55}$$

where ① follows by the triangle inequality, ② follows by the triangle inequality and (53), ③ follows by (51) and (54) with $\tilde{x}_t^* = x^+(y_t)$, $\tilde{y}_t^* = y^+(x_t)$ and ④ by definition of $\eta$. Note that by (55), we have that $x^+(\tilde{y}_t)$ and $y^+(\tilde{x}_t)$ lie in the interior of $\mathcal{X}$ and $\mathcal{Y}$ respectively and hence the constraints $\bar{B}(x^*,\bar{D})$ and $\bar{B}(y^*,\bar{D})$ are inactive. Recall that $x_{t+1}^* = \arg\min_{x\in\mathcal{M}} f(x,\tilde{y}_t) + \frac{1}{2\eta}d(x,x_t)^2$ and $y_{t+1}^* = \arg\max_{y\in\mathcal{N}} f(\tilde{x}_t,y) - \frac{1}{2\eta}d(y,\tilde{y}_t)^2$, hence we have that $x_{t+1}^* = x^+(\tilde{y}_t)$, $y_{t+1}^* = y^+(\tilde{x}_t)$.

$\square$

**Proposition 10.** *For $c > 1$, and $T \in \mathbb{N}_0$ we have that*

$$\prod_{t=0}^{T} \frac{1}{1-(t+c)^{-2}} = \frac{c(c+T)}{(c-1)(c+T+1)} \leq \frac{c}{c-1}.$$

*Proof.* We show $\prod_{t=0}^{T} \frac{1}{1-(t+c)^{-2}} = \frac{c(c+T)}{(c-1)(c+T+1)}$ by induction. The statement holds for $T=0$. Now assume that the statement holds for $T-1$. Then the statement also holds for $T$, which can be shown by noting that ① below holds by the induction hypothesis and rearranging

$$\prod_{t=0}^{T} \frac{1}{1-(t+c)^{-2}} \stackrel{①}{=} \frac{c(c+T-1)}{(c-1)(c+T)} \frac{1}{1-(T+c)^{-2}} = \frac{c(c+T)}{(c-1)(c+T+1)} \leq \frac{c}{c-1}.$$

$\square$

Note that in the two following proposition, we specify *where* we require the smoothness and g-convexity to hold, which is important for the analysis in the paper.

**Proposition 11.** *Let $\mathcal{M}$ be a Riemannian manifold and let $f : \mathcal{M} \to \mathbb{R}$ be g-convex and $L$-smooth in $\mathcal{X} = \bar{B}(x^*, 2d(\bar{x}, x^*))$, where $x^* \in \arg\min_{x \in \mathcal{M}} f(x)$ and $\bar{x} \in \mathcal{M}$. Further, let $x^+ \stackrel{\text{def}}{=} \exp(-\frac{1}{L}\nabla f(\bar{x}))$, then it is*

$$\frac{1}{2L}\|\nabla f(\bar{x})\|^2 \leq f(\bar{x}) - f(x^+) \leq f(\bar{x}) - f(x^*).$$

*Proof.* First, note that

$$d(x^+, x^*) \stackrel{\text{\textcircled{1}}}{\leq} d(x^+, \bar{x}) + d(\bar{x}, x^*) \stackrel{\text{\textcircled{2}}}{\leq} \frac{1}{L}\|\nabla f(\bar{x}) - \nabla f(x^*)\| + d(\bar{x}, x^*) \stackrel{\text{\textcircled{3}}}{\leq} 2d(\bar{x}, x^*) \tag{56}$$

where ① holds by the triangle inequality, ② by the update rule of $x^+$ and by $\nabla f(x^*) = 0$, ③ holds by the $L$-smoothness of $f$ in $\mathcal{X}$. This implies that $x^+ \in \mathcal{X}$. Then we have,

$$f(x^+) - f(\bar{x}) \stackrel{\text{\textcircled{1}}}{\leq} \langle \nabla f(\bar{x}), \text{Log}_{\bar{x}}(x^+)\rangle + \frac{L}{2}d(\bar{x}, x^+)^2 \stackrel{\text{\textcircled{2}}}{\leq} -\frac{1}{2L}\|\nabla f(\bar{x})\|^2,$$

where ① holds by $L$-smoothness of $f$ in $\mathcal{X}$ and since $x^+ \in \mathcal{X}$ by (56), ② by the update rule of $x^+$. Finally, by definition of $x^*$, we have that $f(x^*) \leq f(x^+)$ and it follows that $f(\bar{x}) - f(x^+) \leq f(\bar{x}) - f(x^*)$. $\square$

**Proposition 12.** *Let $\mathcal{M}$ be a Riemannian manifold and $\mathcal{X} \subset \mathcal{M}$ be a closed and g-convex set and let $f : \mathcal{M} \to \mathbb{R}$ be $\mu$-strongly g-convex and differentiable in $\mathcal{X}$. Then, it holds for all $x, y \in \mathcal{X}$ that*

$$\mu d(x, y)^2 \leq \langle \nabla f(x) - \Gamma_y^x \nabla f(y), -\text{Log}_x(y)\rangle.$$

*In particular, it follows that*

$$\mu d(x, y) \leq \|\nabla f(x) - \Gamma_y^x f(y)\|.$$

*Proof.* By $\mu$-strong g-convexity of $f$, we have

$$f(x) - f(y) \leq \langle \nabla f(x), -\text{Log}_x(y)\rangle - \frac{\mu}{2}d(x, y)^2$$

$$f(y) - f(x) \leq \langle \nabla f(y), -\text{Log}_y(x)\rangle - \frac{\mu}{2}d(x, y)^2.$$

Adding both inequalities, we have

$$\mu d(x, y)^2 \leq \langle \nabla f(x) - \Gamma_y^x \nabla f(y), -\text{Log}_x(y)\rangle.$$

Further bounding the right hand side using the Cauchy-Schwarz inequalities and dividing by $d(x, y)$, we have

$$\mu d(x, y) \leq \|\nabla f(x) - \Gamma_y^x \nabla f(y)\|.$$

$\square$

**Proposition 13.** *Let $\mathcal{M}$ be a Riemannian manifold and $\mathcal{X} \subset \mathcal{M}$ be a closed and g-convex set and let $f : \mathcal{M} \to \mathbb{R}$ be g-convex, lower semicontinuous and proper in $\mathcal{X}$. Then it holds that*

$$0 \leq \langle g^*, \text{Log}_{x^*}(x)\rangle \leq \langle g, -\text{Log}_x(x^*)\rangle, \quad \forall x \in \mathcal{X},$$

*where $g^* \in \partial f(x^*)$ and $g \in \partial f(x)$.*

*Proof.* By g-convexity of $f$, we have $\langle g, \text{Log}_x(x^*)\rangle \leq f(x^*) - f(x) \leq \langle g^*, -\text{Log}_{x^*}(x)\rangle$. By the first-order optimality condition, for an optimizer $x^*$ of $f$, we have $0 \leq \langle g^*, \text{Log}_{x^*}(x)\rangle$ for all $x \in \mathcal{X}$, which concludes the proof. $\square$

**Proposition 14.** *We have that $\sum_{t=1}^{T} \frac{1}{(t+1)^2} \leq 1$.*

*Proof.*

$$\sum_{t=1}^{T} \frac{1}{(t+1)^2} = \sum_{t=1}^{T+1} \frac{1}{t^2} - 1 \leq \sum_{t=1}^{\infty} \frac{1}{t^2} - 1 \leq \frac{\pi^2}{6} - 1 \leq 1.$$

$\square$

### D.1. Geometric results

**Fact 15** (Riemannian Cosine-Law Inequalities). *For the vertices $x, y, p \in \mathcal{M}$ of a uniquely geodesic triangle of diameter $D$, we have*

$$\langle \text{Log}_x(y), \text{Log}_x(p) \rangle \geq \frac{\delta_D}{2} d(x,y)^2 + \frac{1}{2} d(p,x)^2 - \frac{1}{2} d(p,y)^2.$$

*and*

$$\langle \text{Log}_x(y), \text{Log}_x(p) \rangle \leq \frac{\zeta_D}{2} d(x,y)^2 + \frac{1}{2} d(p,x)^2 - \frac{1}{2} d(p,y)^2.$$

See (Martínez-Rubio & Pokutta, 2023) for a proof.

**Corollary 16.** *Under the assumptions of Fact 15, the squared distance function $\frac{1}{2} d(\cdot, p)^2$ is $\zeta_D$-smooth and $\delta_D$-strongly g-convex in the geodesic triangle defined by the vertices $x, y, p \in \mathcal{M}$.*

*Proof.* The proof follows directly by noting that the first equation of Fact 15 implies $\delta_D$-strong g-convexity and the second equation implies $\zeta_D$-smoothness. $\qquad\square$

**Remark 17.** *In spaces with lower bounded sectional curvature, if we substitute the constants $\zeta_D$ in the previous Fact 15 by the tighter constant and $\zeta_{d(p,x)}$, the result also holds. See (Zhang & Sra, 2016).*

We note that if $\kappa_{\min} < 0$, it is $\zeta_D = \Theta(1 + D\sqrt{|\kappa_{\min}|})$ and therefore if $c$ is a constant, we have $\zeta_{cD} = O(\zeta_D)$. If $\kappa_{\min} \geq 0$ it is $\zeta_r = 1$, for all $r \geq 0$, so it also holds $\zeta_{cD} = O(\zeta_D)$.

**Proposition 18.** *Let $\mathcal{M}$ be a Hadamard manifold with sectional curvature lower bounded by $\kappa_{\min}$, then for $x, y, p \in \mathcal{M}$ it holds,*

$$\|\Gamma_y^x \text{Log}_y(p) - \text{Log}_x(p)\| \leq \zeta(\kappa_{\min}, \bar{D}) d(x,y), \tag{57}$$

*where $\bar{D} \stackrel{\text{def}}{=} \max\{d(x,p), d(y,p)\}$.*

*Proof.* Let $\Phi_p(x) \stackrel{\text{def}}{=} \frac{1}{2} d(x,y)^2$. Then $\nabla_x \Phi_p(x) = -\text{Log}_x(p)$ and $\Phi_p(x)$ is $(\zeta_{d(x,p)})$-smooth between $x$ and $p$ as the eigenvalues of the Hessian of $\Phi_p(x)$ are upper bounded by $\zeta_{d(x,p)}$ by Alimisis et al. (2020, Lemma 2). Note that the smoothness constant increases with the distance to $p$. Since in Hadamard manifolds, the distance between $p$ and other points in the geodesic triangle defined by $x, y, p$ is maximized at the vertices, then $\Phi_p$ is $\zeta_{\bar{D}}$ smooth in this geodesic triangle, and thus we have that

$$\|\nabla \Phi_p(x) - \Gamma_y^x \nabla \Phi_p(y)\| = \|\Gamma_y^x \text{Log}_y(p) - \text{Log}_x(p)\| \leq \zeta_{\bar{D}} d(x,y).$$

$\qquad\square$

**Fact 19** ((Zhang et al., 2023), Lemma C.2). *Suppose $f$ is geodesically convex-concave. Then for any iteration $(x_t, y_t)$, the geodesic averages $(\bar{x}_t, \bar{y}_t)$, i.e.,*

$$(\bar{x}_1, \bar{y}_1) = (x_1, y_1), \quad t \in \{1, \ldots, T-1\}: \quad \begin{cases} \bar{x}_{t+1} &= \text{Exp}_{\bar{x}_t}(\frac{1}{t+1} \text{Log}_{\bar{x}_t}(x_{t+1})) \\ \bar{y}_{t+1} &= \text{Exp}_{\bar{y}_t}(\frac{1}{t+1} \text{Log}_{\bar{y}_t}(x_{t+1})) \end{cases} \quad \text{(GEO-AVG)}$$

*satisfy for any positive integer $T$,*

$$f(\bar{x}_T, y) - f(x, \bar{y}_T) \leq \frac{1}{T} \sum_{t=1}^{T} [f(x_t, y) - f(x, y_t)].$$

### D.2. G-convex minimization

**Fact 20** (Riemannian Gradient Descent (RGD)). *Consider a uniquely geodesic Riemannian manifold $\mathcal{M}$ with sectional curvature in $[\kappa_{\min}, \kappa_{\max}]$ and a function $f : \mathcal{M} \to \mathbb{R}$ which is $\mu$-strongly g-convex and $L$-smooth in $\mathcal{X} \stackrel{\text{def}}{=} \bar{B}(x^*, (1 + \sqrt{5})d(x_0, x^*)\zeta_R/2) \subset \mathcal{M}$. Then the iterates of RGD, i.e., $x_{t+1} \leftarrow \text{Exp}_{x_t}(-\frac{1}{L}\nabla f(x_t))$, satisfy $x_t \in \mathcal{X}$ and we obtain an $\varepsilon$-minimizer in $O(\frac{L}{\mu} \log(\frac{L d(x_0, x^*)^2}{\varepsilon}))$ iterations.*

See Martínez-Rubio et al. (2024, Proposition 2) for the proof.

**Fact 21** (Projected Riemannian Gradient Descent (PRGD)). *Let $f : \mathcal{M} \to \mathbb{R}$ be a $\mu$-strongly g-convex, $L$-smooth and $L_p$-Lipschitz function in a g-convex compact subset $\mathcal{X} \subset \mathcal{M}$ of a Hadamard manifold $\mathcal{M}$. For an initial point $x_0 \in \mathcal{X}$ and $\tilde{R} \overset{\text{def}}{=} L_p/L$, after*

$$T \geq \min\left\{ \frac{2\zeta_R L}{\mu} \log\left( \frac{f(x_0) - f(x^*)}{\varepsilon} \right), 1 + \frac{2\zeta_{\tilde{R}} L}{\mu} \log\left( \frac{L\zeta_{\tilde{R}} d^2(x_0, x^*)}{2\varepsilon} \right) \right\}$$

*steps of PRGD with update rule $x_{t+1} \leftarrow \mathcal{P}_\mathcal{X}\left( \text{Exp}_{x_t}\left( -\frac{1}{L}\nabla f(x_t) \right) \right)$, we have $f(x_T) - f(x^*) \leq \varepsilon$.*

See Martínez-Rubio et al. (2023, Proposition 6) for the proof.

We note that the original convergence result of Composite Riemannian Gradient Descent (Martínez-Rubio et al., 2024, Proposition 5) contains a minor error. The following proposition is a corrected version.

**Proposition 22** (Composite Riemannian Gradient Descent (CRGD)). *Let $\mathcal{M}$ be a uniquely geodesic Riemannian manifold and let $\mathcal{X} \subset \mathcal{M}$ be compact and g-convex. Let $f : \mathcal{M} \to \mathbb{R}$ be g-convex and $L$-smooth in $\mathcal{X}$ and $g : \mathcal{M} \to \mathbb{R}$ be g-convex, proper and lower semicontinuous in $\mathcal{X}$ such that $F \overset{\text{def}}{=} f + g$ is $\mu$-strongly g-convex in $\mathcal{X}$, and $x^* \overset{\text{def}}{=} \arg\min_{x \in \mathcal{X}} F(x)$. Define the update rule of CRGD as follows*

$$x_{t+1} \leftarrow \arg\min_{y \in \mathcal{X}} \left\{ \langle \nabla f(x_t), \text{Log}_{x_t}(y) \rangle + \frac{L}{2} d(x_t, y)^2 + g(y) \right\}.$$

*Then*

$$F(x_{t+1}) - F(x^*) \leq C(F(x_t) - F(x^*)), \quad \text{where } C \overset{\text{def}}{=} \begin{cases} 1 - \mu/(4L) & \text{if } L/\mu \geq 1/2 \\ \frac{L}{\mu} & \text{if } L/\mu < 1/2 \end{cases}.$$

*and in particular*

$$F(x_T) - F(x^*) \leq C^T(F(x_0) - F(x^*)).$$

*Proof.* We first note that the $\arg\min$ in the update rule exists. Since $g$ is proper, lower semicontinuous and g-convex in $\mathcal{X}$, we have that $\mathcal{Y} \overset{\text{def}}{=} \mathcal{X} \cap \text{dom}(g)$ is non-empty, closed and if $x \in \mathcal{Y}$ and $v \in \partial g(x)$, we have that $\{y \in \mathcal{Y} \mid \frac{L}{4}d(x_t, y)^2 + \langle v, \text{Log}_x(y) \rangle \leq \frac{L}{4}d(x_t, x)^2\}$ is compact by strong convexity of $x \mapsto d(x_t, x)^2$. We also have that $\{y \in Y \mid \frac{L}{4}d(x_t, y)^2 + \langle \nabla f(x), \text{Log}_{x_t}(y) \rangle \leq \frac{L}{4}d(x_t, x)^2 + \langle \nabla f(x), \text{Log}_{x_t}(x) \rangle\}$ is compact. The union of these two compact sets is compact and if we consider $z$ not in this union, we have ② below

$$\langle \nabla f(x_t), \text{Log}_{x_t}(z) \rangle + \frac{L}{2}d(x_t, z)^2 + g(z) \overset{①}{\geq} \langle \nabla f(x_t), \text{Log}_{x_t}(z) \rangle + \frac{L}{2}d(x_t, z)^2 + g(x) + \langle v, \text{Log}_x(z) \rangle$$

$$\overset{②}{>} \langle \nabla f(x_t), \text{Log}_{x_t}(x) \rangle + \frac{L}{2}d(x_t, x)^2 + g(x),$$

where ① uses $v \in \partial g(x)$. This means that the minimization problem can be constrained to this union only and since it is compact the $\arg\min$ exists.

Now we prove the convergence result. We have

$$F(x_{t+1}) \overset{①}{\leq} \min_{x \in \mathcal{X}} \left\{ f(x_t) + \langle \nabla f(x_t), x - x_t \rangle_{x_t} + \frac{L}{2}d(x, x_t)^2 + g(x) \right\}$$

$$\overset{②}{\leq} \min_{x \in \mathcal{X}} \left\{ F(x) + \frac{L}{2}d(x, x_t)^2 \right\}$$

$$\overset{③}{\leq} \min_{\alpha \in [0,1]} \left\{ \alpha F(x^*) + (1-\alpha)F(x_t) + \frac{L\alpha^2}{2}d(x^*, x_t)^2 \right\} \tag{58}$$

$$\overset{④}{\leq} \min_{\alpha \in [0,1]} \left\{ F(x_t) - \alpha\left(1 - \alpha\frac{L}{\mu}\right)(F(x_t) - F(x^*)) \right\}$$

$$\overset{⑤}{=} F(x_t) - \min\left\{ \frac{\mu}{4L}, \frac{1}{2} \right\}(F(x_t) - F(x^*)).$$

Above, ① holds by smoothness and the update rule of the composite Riemannian gradient descent algorithm. The g-convexity of $f$ implies ②. Inequality ③ results from restricting the min to the geodesic segment between $x^*$ and $x_t$ so that $x = \mathrm{Exp}_{x_t}(\alpha \mathrm{Log}_{x_t}(x^*) + (1-\alpha)\mathrm{Log}_{x_t}(x_t))$. We also use the g-convexity of $F$. In ④, we used strong convexity of $F$ to bound $\frac{\mu}{2}d(x^*, x_t)^2 \leq F(x_t) - F(x^*)$. Finally, in ⑤ we substituted $\alpha$ by the value that minimizes the expression, which is $\max\{1, \mu/2L\}$. Subtracting $F(x^*)$ to the inequality above yields

$$F(x_{t+1}) - F(x^*) \leq \left(1 - \min\left\{\frac{\mu}{4L}, \frac{1}{2}\right\}\right)(F(x_t) - F(x^*)).$$

This bound does not improve for $L/\mu \leq 1/2$. We can further improve the bound for $L/\mu \leq 1/2$. Indeed, we have

$$F(x_{t+1}) - F(x^*) \overset{①}{\leq} \frac{L}{2}d(x_t, x^*)^2 \overset{②}{\leq} \frac{L}{\mu}(F(x_t) - F(x^*)),$$

where ① holds by Corollary 23 and ② holds by $\mu$-strong g-convexity of $F$ in $\mathcal{X}$. Subtracting $F(x^*)$ to the inequality above yields

$$F(x_{t+1}) - F(x^*) \leq \frac{L}{\mu}(F(x_t) - F(x^*)).$$

For $L/\mu \leq 1/2$, we have $L/\mu \leq \left(1 - \min\left\{\frac{\mu}{4L}, \frac{1}{2}\right\}\right)$. Recursively applying the inequalities from $t = 1$ to $T$ yields

$$F(x_T) - F(x^*) \leq C^T(F(x_0) - F(x^*)), \quad \text{where } C \overset{\text{def}}{=} \begin{cases} 1 - \mu/(4L) & \text{if } L/\mu \geq 1/2 \\ \frac{L}{\mu} & \text{if } L/\mu < 1/2 \end{cases}.$$

$\square$

**Corollary 23** (Composite warm start)**.** *Consider the setting of Proposition 22. Then, we have for all $z \in \mathcal{X}$ that $F(x_{t+1}) - F(z) \leq \frac{L}{2}d(x_t, z)^2$.*

*Proof.* The proof follows from ② in (58) by noting that $\min_{x \in \mathcal{X}} F(x) + \frac{L}{2}d(x, x_t)^2 \leq F(z) + \frac{L}{2}d(z, x_t)^2$ for all $z \in \mathcal{X}$. $\square$

**Corollary 24.** *Let $\mathcal{M}$ be a finite-dimensional uniquely geodesic Riemannian manifold. Further let $f$ be $L$-smooth and g-convex in $\mathcal{X}$ and let $g$ be g-convex, lower semicontinuous and proper in $\mathcal{X}$ such that $F \overset{\text{def}}{=} f + g$ is $\mu$-strongly g-convex in $\mathcal{X}$, where $\mathcal{X} \overset{\text{def}}{=} \bar{B}(x^*, (1+c)d(x, x^*))$ for some $c > 0$ and $x^* \overset{\text{def}}{=} \arg\min_{x \in \mathcal{M}} F(x)$. Let*

$$x^+ \overset{\text{def}}{=} \underset{y \in \mathcal{M}}{\arg\min} \left\{\langle \nabla f(x), \mathrm{Log}_x(y)\rangle + \frac{L}{2}d(x, y)^2 + g(y)\right\}.$$

*Then, if $L/\mu \leq 1$*

$$d(x^+, x^*)^2 \leq \frac{L}{\mu}d(x, x^*)^2 \leq d(x, x^*)^2.$$

*Proof.* Let the following be the optimizer, constrained to $\mathcal{X}$,

$$\bar{x}^+ \overset{\text{def}}{=} \underset{y \in \mathcal{X}}{\arg\min} \left\{\langle \nabla f(x), \mathrm{Log}_x(y)\rangle + \frac{L}{2}d(x, y)^2 + g(y)\right\}.$$

Then we have

$$d(\bar{x}^+, x^*)^2 \overset{①}{\leq} \frac{2}{\mu}(F(\bar{x}^+) - F(x^*)) \overset{②}{\leq} \frac{L}{\mu}d(x, x^*)^2,$$

where ① hold by $\mu$-strong g-convexity of $F$ in $\mathcal{X}$ and ② follows by Corollary 23. Hence $\bar{x}^+$ lies in the interior of $\mathcal{X}$, which implies that the constraints are inactive and $x^+ = \bar{x}^+$. $\square$

# E. Experiments

We consider a robust version of the Karcher mean (Karcher, 1977) for points $y_1, \ldots, y_n \in \mathcal{M}$. Previous works controlled the degree of robustness through regularization (Zhang et al., 2023; Jordan et al., 2022), i.e.,

$$\min_{x \in \mathcal{M}} \max_{\tilde{y}_i \in \mathcal{M}} \left\{ F(x, (y_1, \ldots, y_n)) \stackrel{\text{def}}{=} \frac{1}{n} \sum_{i=1}^{n} d(x, \tilde{y}_i)^2 - \frac{\gamma}{n} \sum_{i=1}^{n} d(y_i, \tilde{y}_i)^2 \right\}, \tag{59}$$

where $\gamma$ controls the amount of robustness. We formulate a robust Karcher variant based on constraints, i.e.,

$$\min_{x \in \mathcal{M}} \max_{\tilde{y}_i \in \mathcal{Y}_i} F(x, (y_1, \ldots, y_n)), \tag{60}$$

where $\mathcal{Y}_i \stackrel{\text{def}}{=} \bar{B}(y_i, \bar{R})$ for all $i \in [n]$. This formulation allows for a more fine-grained influence of the robustness via the radius $\bar{R}$ of the constraints balls.

We implement the experiments using the Pymanopt Library (Townsend et al., 2016) in the symmetric positive definite (SPD) manifold $\mathcal{S}_+^d \stackrel{\text{def}}{=} \{M \in \mathbb{R}^{d \times d} : M = M^T, M \succ 0\}$ equipped with the affine-invariant metric and the $d$-dimensional hyperbolic space $\mathbb{H}^d$. We measure the performance of point $(\hat{x}, (\hat{y}_1, \ldots, \hat{y}_n))$ in terms of the duality gap

$$\max_{\tilde{y}_i \in \mathcal{Y}_i} F(\hat{x}, (y_1, \ldots, y_n)) - \min_{x \in \mathcal{M}} F(x, (\hat{y}_1, \ldots, \hat{y}_n)).$$

Note that for both (59) and (60), we require $\gamma > \zeta_{\bar{D}}$, where $\bar{D}$ is the diameter a the set containing $x$ and $\tilde{y}_i$, in order to ensure that the problem is g-convex, g-concave. In the following, we show that it is sufficient to choose $\bar{D} \stackrel{\text{def}}{=} 1 + \bar{R}$ based on how we generate instances of (60). First, we generate a random point $\bar{y}$ on the manifold using the `manifold.random_point()` function. Then, we generate the centers $y_i = \text{Log}_{\bar{y}}(v_i/\|v_i\|_{\bar{y}})$ based on sampling random tangent vectors $v_i \in T_{\bar{y}}\mathcal{M}$ using the `manifold.random_tangent_vector()` function. That way, we know that all $y_i \in \bar{B}(\bar{y}, 1)$. This ensures that the Karcher mean of the points $y_1, \ldots, y_n$, i.e., $\text{KM}(y_1, \ldots, y_n) = \arg\min_{x \in \mathcal{M}} \frac{1}{2n} \sum_{i=1}^{n} d(x, y_i)^2$ also lies in $\bar{B}(\bar{y}, 1)$ by Martínez-Rubio et al. (2024, Proposition 30). Since we constrain the variables $\tilde{y}_i$ to lie in $\bar{B}(y_i, \bar{R})$, we have that $\tilde{y}_i \in \bar{B}(\bar{y}, 1 + \bar{R})$ and it follows that $\text{KM}(\tilde{y}_1, \ldots, \tilde{y}_n)$ also lies in $\bar{B}(\bar{y}, 1 + \bar{R})$ by Martínez-Rubio et al. (2024, Proposition 30).

We have that $F(\cdot, (y_1, \ldots, y_n))$ is 1-strongly g-convex and $(\gamma - \zeta_{\bar{D}})$-strongly g-concave, for $\gamma > \zeta_{\bar{D}}$. That means that we can ensure in particular that the problem is strongly g-convex and strongly g-concave.

We run RIODA$_{\text{PRGD}}$ on (60) with a fixed number of 3 PRGD steps per subroutine, which means that each iteration of RIODA$_{\text{PRGD}}$ require 12 PRGD steps. Setting $\bar{R} = 0.01$ and $\gamma = \zeta_{\bar{D}}$ ensures that the problem is strongly g-concave in $(y_1, \ldots, y_n)$ as our bound on $\bar{D}$ is loose. For the experiments, RIODA$_{\text{PRGD}}$ is run for $1k$ iterations, corresponding to 12k gradient oracle calls. The step size $\lambda = \{10^{-1}, 10^{-2}, 10^{-3}\}$ of PRGD and the proximal parameter $\eta \in \{10^{-1}, 10^{-2}\}$ are optimized to find the best hyperparameters via a grid search.

The following two figures show the convergence behavior of RIODA$_{\text{PRGD}}$ in terms of the duality gap for experiments run in both the hyperbolic space $\mathbb{H}^{5000}$ and the SPD manifold $\mathcal{S}_+^{100}$, each with $n = 50$ points. We observe linear convergence in both cases, which aligns with our theoretical analysis.

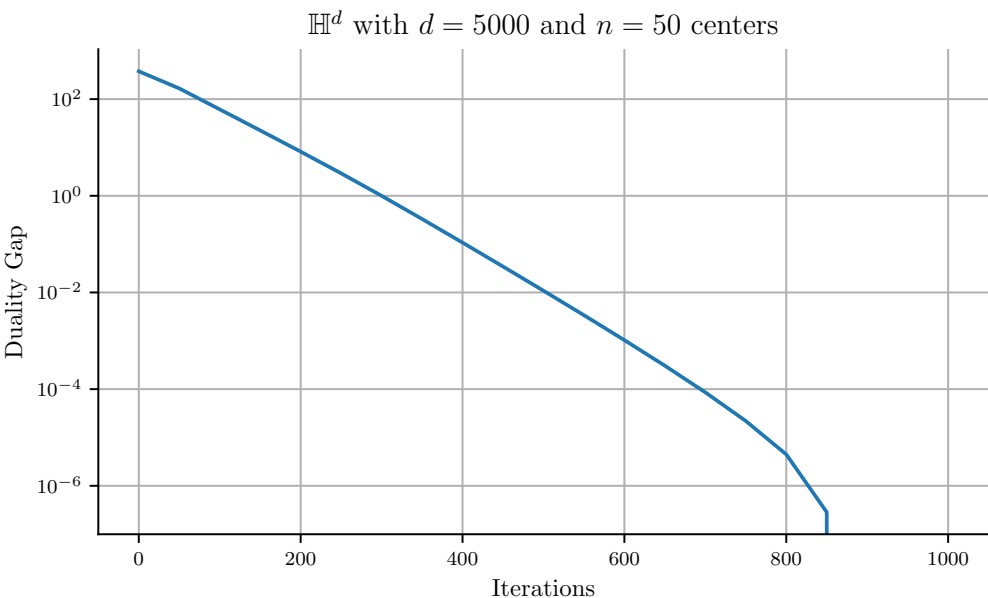

Figure 1: Convergence of RIODA_PRGD on the robust Karcher mean problem (60) in terms of the duality gap with $\lambda = 0.01$, $\eta = 0.01$

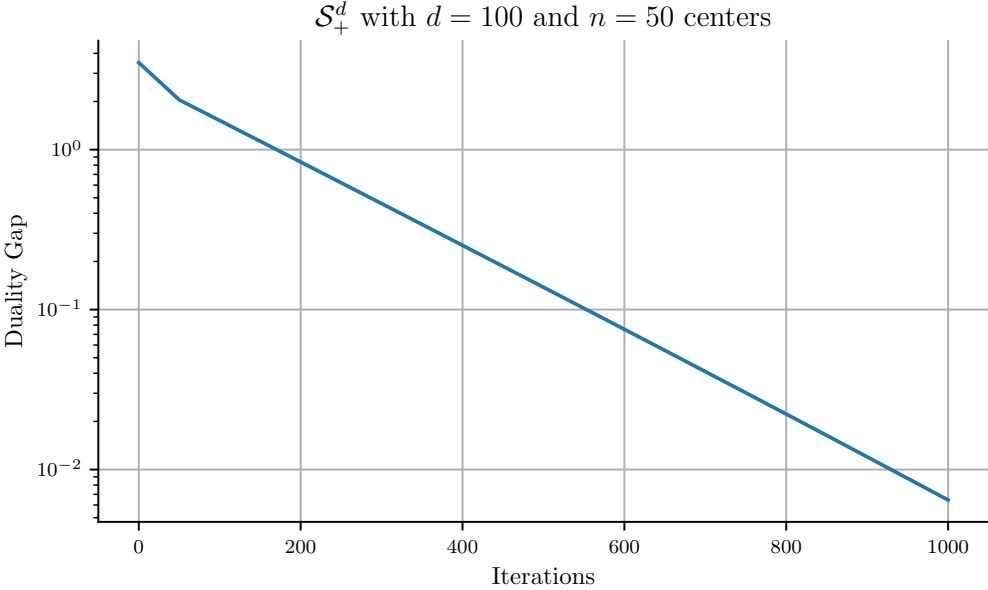

Figure 2: Convergence of RIODA_PRGD on the robust Karcher mean problem (60) in terms of the duality gap with $\lambda = 0.1$, $\eta = 0.0001$

