# OpenReview forum: "Implicit Riemannian Optimism with Applications to Min-Max Problems"
_ICML.cc/2025/Conference — ICML 2025 poster_

### Official Review · Reviewer_hBQp · 2025-03-13

**Overall Recommendation:** 5

**Summary:**

The paper addresses online optimization and min-max optimization on Hadamard manifolds. The authors propose a Riemannian optimistic online learning algorithm called RIOD, designed to match state-of-the-art Euclidean regret bounds while handling in-manifold constraints. Leveraging RIOD, they develop a min-max solver that likewise exhibits near-optimal rates and can enforce constraints.

**Claims And Evidence:**

The claims are well supported.

**Essential References Not Discussed:**

The references are well discussed.

**Experimental Designs Or Analyses:**

The experiments are well designed.

**Methods And Evaluation Criteria:**

The paper addresses the previously incomplete story of optimistic methods on manifolds by introducing implicit updates that exactly preserve in-manifold constraints. This resolves the limitation that arises in other works.

**Other Comments Or Suggestions:**

No comments

**Other Strengths And Weaknesses:**

The results in the paper are novel and strong. This work is a substantial step in unifying the benefits of optimistics online learning with the geometry of negatively curved manifolds.

**Questions For Authors:**

No questions.

**Relation To Broader Scientific Literature:**

Most prior works either avoided constraints or accepted “improper” iterates that stray from the feasible set. By contrast, RIOD ensures proper iterates with classical regret bounds.

**Theoretical Claims:**

The theorems and propositions are well posed and formally proved.

---

> ### Author Rebuttal · Authors · 2025-03-28
>
> We want to thank the reviewer for their time taken revising our manuscript, as well as the appendix and the experimental section.

---

### Official Review · Reviewer_D6BU · 2025-03-14

**Overall Recommendation:** 4

**Summary:**

This paper studies Riemannian online optimization as well as min-max optimization problems. The main contribution is to propose algorithms for these problems and provide theoretical analyses.

**Claims And Evidence:**

Yes.

**Essential References Not Discussed:**

None that I am aware of.

**Experimental Designs Or Analyses:**

There are no experiments, but I don't think they are necessary.

**Methods And Evaluation Criteria:**

Yes.

**Other Comments Or Suggestions:**

None.

**Other Strengths And Weaknesses:**

None.

**Questions For Authors:**

None.

**Relation To Broader Scientific Literature:**

For online Riemannian optimization, the authors apparently remove a spurious curvature-dependent term from the upper bound in previous analyses. However, curvature-dependent terms re-appear in the time-complexity in the inner steps of their algorithm. It appears that the main improvement relative to previous work is the removal of pre-existing assumptions that circularly linked the curvature and the step-sizes, those being difficult to verify.

**Theoretical Claims:**

I did check the proofs in any detail.

---

> ### Author Rebuttal · Authors · 2025-03-28
>
> > For online Riemannian optimization, the authors apparently remove a spurious curvature-dependent term from the upper bound in previous analyses. However, curvature-dependent terms re-appear in the time-complexity in the inner steps of their algorithm.
>
> Note that in the online setting, the computational or gradient oracle complexity is a metric independent from the regret. That being said, in minmax and other applications, both metrics translate to computational or gradient oracle complexity, since they are multiplied to obtain the final complexity.
>
> The implementation using our CRGD subroutine has an oracle complexity which **does not introduce curvature terms**, except in a mild logarithmic term. This showcases the power of CRGD when it comes to reducing the subproblem complexity. We provided the time complexity of PRGD as well, for comparison, since it is a widely used and natural subroutine, this one does have curvature terms.
>
> > It appears that the main improvement relative to previous work is the removal of pre-existing assumptions that circularly linked the curvature and the step-sizes, those being difficult to verify.
>
> We would like to note that beyond the contribution mentioned by the reviewer we also improve in the following:
>
>  - Our method can explicitly handle bounded in-manifold constraints, which are interesting in their own right, since constraints are also critical for online optimization algorithms, even in the Euclidean setting.
>
>  - We further obtain important new results in min-max optimization.
>     - Our methods reduce the computational complexity with respect to other constrained methods, it obtains a near optimal gradient oracle complexity, and does not require the knowledge of the initial distance to a solution $R$ and the Lipschitz constant $G$ (see table 2).
>     - It was previously unknown if achieving our rate was possible, since in fact in g-convex and $L$-smooth minimization, nearly matching Euclidean upper bounds is impossible due to lower bounds implying an inherent hardness coming from the geometry, but we showed that the minmax problems are of a different nature.

---

### Official Review · Reviewer_kVH1 · 2025-03-17

**Overall Recommendation:** 3

**Summary:**

This paper studies Riemannian online optimization and min-max optimization. The authors propose a novel online optimistic Riemannian optimization algorithm with inexact updates, which achieves optimal dynamic regret, and also enforces in-mainfold constraints. The authors then applied their online optimization method to Riemannian min-max optimization and obtain either reduced complexity or same rates without requiring the knowledge of certain parameters.

## Updated Review ###

I have read the rebuttal and the other reviews. I would like to thank the authors for their detailed feedback, which addresses most of my major concerns. Regarding the issue raised by Reviewer fAnn, I agree with the authors that online learning on Hadamard manifolds is a challenging problem, and there has been a long line of research in this area.

**Claims And Evidence:**

The claims made in the submission supported by clear and convincing evidence.

**Essential References Not Discussed:**

N/A

**Experimental Designs Or Analyses:**

this is a theory paper and does not have experiments.

**Methods And Evaluation Criteria:**

The evaluation Criteria are dynamic regret for online optimization convergence rate for min-max optimization, which are standard.

**Other Comments Or Suggestions:**

Please see above.

**Other Strengths And Weaknesses:**

Strengthes:

The major advantage of the proposed method is that it is the first algorithm for minimizing the dynamic regret on Riemannian manifold that can **enforce in-manifold constraints**, which is a significant and important contribution to this area. In previous work, such as [Hu et al. (2023a)], the algorithm can only minimize "improper" dynamic regret, that is, the decision set is larger than the competitor. This is not very satisfying as it means that the learner can pick decisions outside of the constrained set. This paper successfully solved this problem with novel inexact updates.

Moreover, the authors applied their optimism methods for Riemannian min-max optimizaiton, which improve on prior works by either reducing the complexity or not requiring the knowledge of certain parameters.

Weaknesses:

My major concern about the proposed methods is about the computational complexity.

In [Hu et al. (2023a)] and [Wang et al. (2023b)], the algorithms are usually iterative, that is, to update the decision, the algorithm only need to query the Rimannian gradients **once** and commit the update. However, in this paper (i.e., lines 4 and 6 of Algorithm 1), to implement the proposed algorithm we need to solve two optimization problems at each round, which requires multiple queries of the gradients, and it makes the setting not truly "online" and also introduce extra dependence on L and \zeta. Finally, in Corollary 2, it says that the number of implements depend on \eta, so I am not sure how to set the  number of implements if eta is not known (since eta=1/sqrt{V_T} and V_T is not known).

Since in the sub-problems, the number of iterations depend on \zeta, I think the final "real" regret should also depend on zeta, as the total number of iterations is not T (but T times the  number of inner iterations).

**Questions For Authors:**

Please see above.

**Relation To Broader Scientific Literature:**

This paper did a good job on discussing and comparing with previous work. I particularly appreciate the two tables on page 4, which clearly present all the convergence results and conditions for existing methods.

**Theoretical Claims:**

All theoretical claims are accompanied by proofs.

---

> ### Author Rebuttal · Authors · 2025-03-28
>
> > this paper does not have experiments.
>
> We note that as we stated in the paper, we did provide experiments in Appendix E to validate our theory, with an implementation of our method in a constrained problem setting, for an application that only one other method (RAMMA, an impractical method) could tackle before.
>
> > to implement the proposed algorithm we need to solve two optimization problems at each round, which requires multiple queries of the gradients, and it makes the setting not truly "online"
>
> The action $\tilde{x}_t$ which RIOD chooses at time $t$ is computed based on an implicit step on the hint function $\tilde{\ell}_t$ and the loss from the previous round $\ell_{t-1}$ (this dependence is implicit via the iterate $x_{t}$). RIOD **does not** require the knowledge of the loss function $\ell_t$ before choosing its action. Therefore, our algorithm fits in the classical online optimization setting, which assumes that after choosing an action, the agent observes the **full** loss function $\ell_{t-1}$ and does not specify what the agent does with it: querying the gradient more than once is always permitted.
>
> Note that one of the previous optimistic methods in Riemannian optimization [Hu et al. (2023)](https://proceedings.mlr.press/v195/hu23a.html) also queries the gradient more than once per round.
>
> In fact, in many other online learning settings **exact minimization** of the previous loss plus possibly some regularizer is assumed, and some other online learning papers studied implicit updates where several gradients are taken [Kivinen & Warmuth (1997)](https://www.sciencedirect.com/science/article/pii/S0890540196926127), [Campolongo & Orabona (2020)](https://proceedings.neurips.cc/paper/2020/hash/9239be5f9dc4058ec647f14fd04b1290-Abstract.html) [Chen & Orabona (2023)](https://proceedings.mlr.press/v202/chen23t.html) with some advantages over explicit methods. Note however that no work obtained an optimistic online learning algorithm using implicit updates.
>
>
> > However, in this paper (i.e., lines 4 and 6 of Algorithm 1), to implement the proposed algorithm we need to solve two optimization problems at each round, which [...] [introduces] extra dependence on $L$ and $\zeta$
>
> The error criteria of the updates in lines 4 and 6 of Algorithm 1 depend on $L$ and $\zeta$, but this only introduces a logarithmic dependence when using CRGD. In any case, the regret metric is independent of the gradient complexity so this does not imply a dependence on $L$ and $\zeta$ in the regret bounds, not even a logarithmic one.
>
> > Finally, in Corollary 2, it says that the number of implements depend on $\eta$;, so I am not sure how to set the number of implements if eta is not known (since $eta=1/sqrt{V_T}$ and $V_T$ is not known).
>
> In general, RIOD works for any $\eta>0$ in contrast to the previous explicit optimistic methods, which require $\eta\lesssim 1/L$ (and possibly require some geometric terms in the Riemannian setting) in order to show the regret bound, that matches the OOMD [Rakhlin et al. (2013)](https://proceedings.neurips.cc/paper/2013/hash/f0dd4a99fba6075a9494772b58f95280-Abstract.html). As for [Rakhlin et al. (2013)](https://proceedings.neurips.cc/paper/2013/hash/f0dd4a99fba6075a9494772b58f95280-Abstract.html), in this work we do not focus on adapting for the best value of \eta. But we note that (A) we did not require this to obtain optimal convergence for smooth g-convex g-concave optimization and (B) in the general case, when $V_T$ is not known in advance, we can follow the method laid out in Theorem 3, [Zhao et al, (2020)](https://proceedings.neurips.cc/paper_files/paper/2020/hash/939314105ce8701e67489642ef4d49e8-Abstract.html), which is based on using an expert meta-algorithm, that is not dependent on the type of algorithm being run. This methods was shown to work in the Riemannian setting in [Hu et al. (2023)](https://proceedings.mlr.press/v195/hu23a.html).
>
> > Since in the sub-problems, the number of iterations depend on $\zeta$, I think the final "real" regret should also depend on zeta, as the total number of iterations is not $T$ (but $T$ times the number of inner iterations).
>
> In the online setting, the computational or gradient complexity is an independent metric from the regret, so there is no extra factor. But indeed in the minmax problem both metrics multiply in order to obtain the total gradient complexity, that yields our rates in Table 2, which contain a dependence on $\zeta$ for PRGD or RGD, but only a logarithmic factor for CRGD. This showcases the power of CRGD when it comes to reducing the subproblem complexity.
>
> We appreciate that the reviewer found the paper well-written and valued our results in Riemannian online learning and minmax problems, overcoming several difficulties in existing work. If you now have a better opinion of our work, we kindly ask you to consider increasing your score.

---

### Official Review · Reviewer_fAnn · 2025-03-20

**Overall Recommendation:** 1

**Summary:**

In this paper the authors consider Riemannian optimism on Hadamard manifolds.

**Claims And Evidence:**

Theoretical paper, no evidence presented.

**Essential References Not Discussed:**

NA

**Experimental Designs Or Analyses:**

None.

**Methods And Evaluation Criteria:**

None present.

**Other Comments Or Suggestions:**

NA

**Other Strengths And Weaknesses:**

The authors claim that their concentration bounds no longer rely on the curvature of the space is a strength, however, Hadamard manifolds have non-positive curvature. The issue I see with this is that positively curved spaces are the challenging manifolds, in a sense. For instance, in the estimation of the Karcher mean, there are strong limitations on positively curved spaces but on non-positively curved spaces there is no such issues. Generally speaking, non positively curved manifolds behave very similar to Euclidean spaces and thus I don't see the results of this paper as being surprising. Lastly, the main paper didn't have any experimental or simulation results which I see as a huge limitation.

**Questions For Authors:**

NA

**Relation To Broader Scientific Literature:**

NA

**Theoretical Claims:**

Yes, they were briefly checked.

---

> ### Author Rebuttal · Authors · 2025-03-28
>
> > Experimental results
>
> We note that as stated in the paper, we did provide experiments in Appendix E with an implementation of our method, in a constrained problem setting, an application that only one other method (RAMMA, an impractical method) could tackle before, showing that now it is possible to efficiently solve such problems.
>
> The Hadamard assumption is standard in the field and quite general (accelerated minimization algorithms [[1]](https://proceedings.mlr.press/v195/martinez-rubio23a.html), lower bounds, [[2]](https://proceedings.neurips.cc/paper_files/paper/2021/hash/201d546992726352471cfea6b0df0a48-Abstract.html), [[3]](https://proceedings.mlr.press/v178/criscitiello22a.html), online learning [[4]](https://www.jmlr.org/papers/v24/21-1308.html), [[5]](https://proceedings.neurips.cc/paper/2021/hash/ee389847678a3a9d1ce9e4ca69200d06-Abstract.html), [[6]](https://proceedings.mlr.press/v195/hu23a.html), among many others [[7]](https://www.tandfonline.com/doi/abs/10.1080/02331934.2020.1810248), [[8]](https://www.tandfonline.com/doi/full/10.1080/02331934.2022.2088369), [[9]](https://www.tandfonline.com/doi/full/10.1080/01630563.2018.1553887), [[10]](https://link.springer.com/article/10.1007/s10208-023-09628-5), [[11]](https://searchworks.stanford.edu/view/13531274), [[12]](https://www.sciencedirect.com/science/article/pii/S0362546X02002663)).
>
> The Hadamard case is challenging enough, note that besides other optimization problems listed above, **7 papers before ours** study our minmax problem and none of them could achieve our parameter freeness, or working with constraints or achieving the optimal convergence rates. Similarly, for the online learning case, prior work could not achieve the optimal regret, while at the same time they had unreasonable assumptions, like some circularity between the step sizes (requiring a specific bound on the iterates a prior) and where the iterates lied.

---

### Official Review · Reviewer_qPnh · 2025-03-21

**Overall Recommendation:** 3

**Summary:**

This paper proposes implicit Riemannian optimistic online methods for problems with g-convex and g-concave objectives, accommodating in-manifold constraints and matching state-of-the-art Euclidean rates. The analysis removes the dependence on geometric constants which closes the gap between the Riemannian problem and its Euclidean counterpart, in some sense. The principle is applied to the (Riemannian) minmax problem and obtains new results. Experiments on the Karcher mean problem are implemented to demonstrate the effectiveness of the proposed methods.

**Claims And Evidence:**

The paper is well-written, and the presentation of Table 1 and Table 2 is detailed, clarifying contributions to Riemannian optimization, online learning, and minmax problem.

**Essential References Not Discussed:**

It seems that all the highly related references are discussed.

**Experimental Designs Or Analyses:**

The experiments provided in this paper are limited exclusively to the minmax setting. However, since the original motivation of this work is to address the **Riemannian online learning problem**, the minmax experiments should serve as an illustrative application rather than the primary experimental validation. Expanding the experimental section to assess performance in online learning settings would significantly strengthen the paper and better align with its stated purpose.

**Methods And Evaluation Criteria:**

The extension to minmax problem is interesting, which broadens the work's applicability in machine learning community.

**Other Comments Or Suggestions:**

See questions.

**Other Strengths And Weaknesses:**

See above.

**Questions For Authors:**

1. One of the main contributions emphasized by the authors is the removal of dependence on geometric constants (e.g., $\zeta$) from complexity bounds. Could the authors provide intuitive explanations regarding why the proposed framework successfully circumvents this geometric hurdle, whereas previous methods have not?
2. Corollary 2 reveals that implementing the proposed framework with PRGD still introduces dependence on the geometric constant $\zeta$ while using CRGD does not. This raises a natural question: **is the removal of geometric constants primarily due to the choice of the optimization subroutine (CRGD) rather than the proposed framework (RIOD) itself?** Clarifying this point would help better highlight the true sources of improvement in the paper.
3. Adopting CRGD as a subroutine removes dependence on $\zeta$. However, implementing the CRGD update steps might introduce additional computational complexity in practice. Therefore, it would be valuable to experimentally implement and compare the CRGD-based version of the algorithm, rather than limiting the discussion solely to theoretical advantages.
4. In line 225 (right column), where the author explains OFTRL algorithms without linearizing the loss functions, the statement "minimizing $\sum_{i=1}^t\ell_i(x_i)+\tilde{\ell}_t(x_t)+\frac{1}{2\eta}d(x_t,x)^2$" seems confusing in that the variable $x$ only appears in the last term of the objective.
5. The comparison presented in Table 2 clearly outlines the superiority of this work over existing methods. However, in Appendix E, the implementation only includes the proposed approach. It would be insightful to experimentally compare the proposed methods with some of the algorithms listed in Table 2.

I would like to increase my grade if the concerns are addressed.

**Relation To Broader Scientific Literature:**

The extension to Riemannian minmax problem is interesting, which broadens the application of Riemannian optimisatoin.

**Theoretical Claims:**

The theory is solid, and the method (RIOD), with the min-max extension (RIODA) is shown to yield good regret and oracle-complexity bounds that overcome difficulties in existing work.

---

> ### Author Rebuttal · Authors · 2025-03-28
>
> > Experiments in online learning settings would strengthen the paper
>
> We note that the online learning setting assumes an adversarial agent generating losses against our algorithm. This means that in order to empirically validate the bound, we would need to craft a worst-case adversary to play against our algorithm. This is not only impractical, but it is also undesirable since online learning is a framework that says that we perform well with respect to the best fixed action, and so one also wants to see if it also performs well in favorable scenarios.
>
> In the literature, typical ways of validating online learning bounds in practice are settings where losses are unpredictable, just as the landscape of min max optimization losses.
>
> > One of the main contributions emphasized by the authors is the removal of dependence on geometric constants (e.g., $\zeta$ ) from complexity bounds. Could the authors provide intuitive explanations regarding why the proposed framework successfully circumvents this geometric hurdle, whereas previous methods have not?
>
> There are several independent factors that made our final result possible. We found that (A) implicit (proximal) approaches make the outer loop of our algorithms be free of these constants if used with optimism, as opposed to most of prior work that uses explicit methods and have to rely on the use of Riemannian Cosine-Law Inequalities introducing $\zeta$ factors (Appendix D.1). Then (B), achieving such implicit optimism requires a careful structure where losses are not linearized since the linearizations are not g-convex (unlike in the Euclidean space). Finally (C) the use of a subroutine such as CRGD as opposed to the most commonly used Riemannian Gradient Descent ones makes the inner problem be also free of these constants, up to a log factor.
>
> > The comparison presented in Table 2 clearly outlines the superiority of this work over existing methods. It would be insightful to experimentally compare the CRGD-based version and other proposed methods with some of the algorithms listed in Table 2.
>
> We focused on showcasing that we can effectively optimize problems with in-manifold constraints, which was not possible to optimize before, except with RAMMA in [Martínez-Rubio et al. (2023)](https://arxiv.org/abs/2305.16186). While this method is of theoretical interest, as it establishes new upper bounds, it is highly complex and consists of 5 loops, making it impractical.
>
> > In line 225 (right column)...
>
> Thanks for pointing this typo out, it should have been \( \sum_{i=1}^{t}\ell_i(x) +\tilde{\ell}_t(x) + \frac{1}{2\eta}d(x_t,x)^2 \).
>
> > Corollary 2 reveals that implementing the proposed framework with PRGD still introduces dependence on the geometric constant while using CRGD does not. Is the removal of geometric constants primarily due to the choice of the optimization subroutine (CRGD) rather than the proposed framework (RIOD) itself?
>
> Note that in the online setting, the regret depends only on the cumulative cost $\sum_{t=1}^T \ell_t(x_t)$ paid for the agent's actions, and is independent of the oracle complexity. Hence, the improved **regret** guarantees (table 1) are independent of the oracle complexity of the optimization subroutine used to compute the actions.
>
> The improvement in terms of the regret is achieved via our inexact implicit optimistic technique. Corollary 2 only discusses the oracle complexity of implementing the subproblems up to the specified precision, which is orthogonal to the regret. Removing the geometric constant in the subroutine is mostly due to CRGD. On the other hand, when applying our method to optimization, the complexities do multiply, making necessary to be free of these constants at both levels, but CRGD only adds a log factor.
>
> The design of the subroutine with our implicit approach has other advantages orthogonal to the use of CRGD or any other subroutine, such as removing the need to know the initial distance a priori or the Lipschitz constant (see table 2).
>
> We appreciate that the reviewer found the paper well-written and valued our results in Riemannian online learning and minmax problems, overcoming several difficulties in existing work. If you now have a better opinion of our work, we kindly ask you to consider increasing your score.

---

> > ### Comment · Reviewer_qPnh · 2025-04-03
> >
> > Thanks for the explanation and clarification. I would keep my original score.

---

### Decision · Program_Chairs · 2025-05-01

**Decision:**

Accept (poster)

**Comment:**

All the reviewers are appreciative of the work and believe that the paper presents interesting insights for enforcing the manifold constraint. The proposed updates are definitely novel. A concern which has been raised is the lack of experiments.